# ZERO-SHOT IMPUTATION WITH FOUNDATION INFERENCE MODELS FOR DYNAMICAL SYSTEMS

**Patrick Seifner[1,2], Kostadin Cvejoski[1,3], Antonia Körner[2] & Ramsés J. Sánchez[1,2,3]**
Lamarr Institute[1], University of Bonn[2] & Fraunhofer IAIS[3]
seifner@cs.uni-bonn.de, sanchez@cs.uni-bonn.de

## ABSTRACT

Dynamical systems governed by ordinary differential equations (ODEs) serve as models for a vast number of natural and social phenomena. In this work, we offer a fresh perspective on the classical problem of imputing missing time series data, whose underlying dynamics are assumed to be determined by ODEs. Specifically, we revisit ideas from amortized inference and neural operators, and propose a novel supervised learning framework for *zero-shot time series imputation*, through parametric functions satisfying some (hidden) ODEs. Our proposal consists of two components. First, a broad probability distribution over the space of ODE solutions, observation times and noise mechanisms, with which we generate a large, synthetic dataset of ODE solutions, along with their noisy and sparse observations. Second, a neural recognition model that is trained *offline*, to map the generated time series onto the spaces of initial conditions and time derivatives of the (hidden) ODE solutions, which we then integrate to impute the missing data. We empirically demonstrate that *one and the same* (pretrained) recognition model can perform zero-shot imputation across 63 distinct time series with missing values, each sampled from widely different dynamical systems. Likewise, we demonstrate that it can perform zero-shot imputation of missing high-dimensional data in 10 vastly different settings, spanning human motion, air quality, traffic and electricity studies, as well as Navier-Stokes simulations — *without requiring any fine-tuning*. What is more, our proposal often outperforms state-of-the-art methods which are trained on the target datasets.

Our pretrained model, repository and tutorials are available online[1].

## 1 INTRODUCTION

Dynamical systems are mathematical systems that change with time according to a fixed evolution rule, and serve as representational and analytical tools for phenomena which generate patterns that change over time. Very often, the recorded changes of these empirical patterns are such that they can be viewed as occurring continuously in time, and thus can be represented mathematically by systems whose evolution rule is defined through differential equations. Dynamical systems governed by ordinary differential equations (ODEs) correspond to an important subset of these models, and describe the rate of change of a single parametric function $\mathbf{x} : \mathbb{R}^+ \to \mathbb{R}^D$, which represents the state of the ($D$-dimensional) system, as time evolves, by means of a vector field $\mathbf{f} : \mathbb{R}^+ \times \mathbb{R}^D \to \mathbb{R}^D$. In equations, we write

$$\dot{\mathbf{x}}(t) = \mathbf{f}(t, \mathbf{x}(t)), \text{ where } \dot{\mathbf{x}}(t) = \frac{d\mathbf{x}(t)}{dt}. \tag{1}$$

These deceptively simple systems have had a fundamental role in our understanding of many natural processes across nearly every scientific discipline — from their very introduction and application to celestial mechanics in the late seventeenth century (Newton, 1687; Bernoulli, 1712), to their function as models of concentration changes in molecular reaction networks (Hoff, 1986); models of population oscillations in biology (Lotka, 1925; Volterra, 1927); of atmospheric convection and its chaotic features (Lorenz, 1963); and of the coherent, high energy modes within turbulent flows

---

[1]https://fim4science.github.io/OpenFIM/intro.html

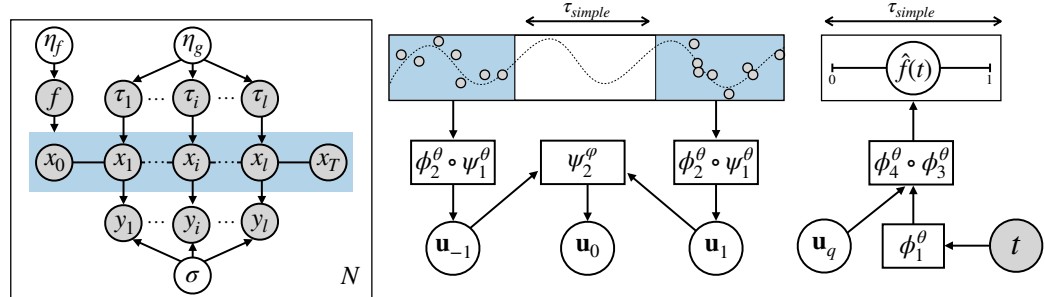

Figure 1: Foundation Inference Model (FIM). *Left*: Graphical model of the data generation model (Eq. 2). Filled (empty) circles represent observed (unobserved) random variables. *Center*: Schematic representation of imputation mechanism for temporal-wise missing patterns (Eq. 7). *Right*: Neural operator module processing the context vector $\mathbf{u}_q$ and query time $t$ (Eq. 4 and 8).

(Noack et al., 2003), just to name a few — and continue to be the go-to mathematical objects for the representation of dynamic phenomena today.

In this work, we consider the general problem of imputing missing values in time series data, recorded from some empirical process $(\mathbf{y}^* : \mathbb{R}^+ \to \mathbb{R}^D)$ whose dynamics are *assumed* to be governed by some unknown ODE. In other words, we assume that both available and missing values in the series $\mathbf{y}^*(\tau_1), \dots, \mathbf{y}^*(\tau_l)$ correspond to the values taken by *the solution* $\mathbf{x}(t)$ of some hidden ODE, at the observation times $\tau_1, \dots, \tau_l$, potentially corrupted by some noise signal of which only a few statistics are known. Therefore, the goal is to infer the ODE solution $\mathbf{x}(t)$ that best *interpolates* the noisy time series $\mathbf{y}^*(\tau_1), \dots, \mathbf{y}^*(\tau_l)$ *and hence imputes its missing values*.

The current machine learning paradigm tackles this problem by (implicitly) constraining models to handle a single process only. That is, practitioners typically encode their inductive biases into either the model architectures or the training objectives, and optimize the model parameters to fit a single empirical distribution (see *e.g.* Section 2). One disadvantage of this approach is that models trained to fit a single process tend to be overly specific to its distribution, and thus can rarely be reused to impute the missing values of a second one, even when both processes are assumed to be governed by *e.g.* similar ODEs. Another disadvantage is that, to succeed, the paradigm requires practitioners to have access to enough observations on the process they study, to train and test their models from scratch, and that they also have the experience and expertise to face the trials and tribulations of their intricate training procedures.

In this paper, we instead frame the imputation problem as an instance of *amortized inference*, in the sense introduced by Stuhlmüller et al. (2013). Indeed, in lieu of training one complex model on a single empirical process, we train a neural recognition model *offline* to infer a *large and varied set* of ODE solutions $\mathbf{x}(t)$, from a synthetic dataset that is composed of noisy series of observation on those solutions, displaying different missing value patterns. Somewhat more precisely, we train our model in a supervised fashion, to infer both the (latent) initial conditions $\mathbf{x}(0)$ and (latent) time derivatives $\dot{\mathbf{x}}(t)$ that determine the target set of ODE solutions. However, opposite to Stuhlmüller et al. (2013) and other follow-up works, like that by Paige & Wood (2016), who treat their recognition models as auxiliary to Monte Carlo methods, we employ our pretrained models to directly impute different synthetic, simulation and experimental datasets, *without any parameter fine-tuning*. We therefore adopt the "zero-shot" terminology introduced by Larochelle et al. (2008)[2]. Let us briefly note here that we have recently used this general amortized inference framework to train models that perform *zero-shot inference* of both Markov jump processes (Berghaus et al., 2024) and stochastic differential equations (Seifner et al., 2025).

In what follows, we first review previous work on time series imputation in Section 2. Section 3 introduces our main ideas, the synthetic dataset encoding our assumptions, and our recognition

---

[2]We invite the reader to check Appendix A, where we also comment on the differences between our methodology and other meta-learning methods.

model, which we name *Foundation Inference Model*[3] (FIM) for dynamical systems. In Section 4, we report our experimental findings and empirically demonstrate that: (i) the hierarchical structure underlying FIM — which treats $\mathbf{x}(0)$ and $\dot{\mathbf{x}}(t)$ as latent variables — allows us to reconstruct, in a zero-shot fashion, the phase portrait of complex dynamical systems; (ii) FIM is able to impute, in zero-shot mode, missing values in a set of 63 noisy time series, each of which is sampled from dynamical systems of different dimensionalities; and (iii) the same (pretrained) FIM can perform zero-shot imputation of vastly different, high-dimensional, experimental and simulation data, while often out-performing state-of-the-art models which are trained on the target datasets. Finally, Section 5 comments on the main limitations of our methodology, and closes the paper with some concluding remarks about future work.

## 2 RELATED WORK

In its most general form, the problem of imputing missing time series data with an ODE model involves inferring the vector field ($\mathbf{f} : \mathbb{R}^+ \times \mathbb{R}^D \to \mathbb{R}^D$) that defines the (hidden) ODE in question. In practice, however, one does not need to explicitly infer the vector field in order to impute the missing data — *and we will take this perspective in the present work*. Prominent examples are the variants by Che et al. (2018) and Cao et al. (2018) that model the dynamics interpolating the data in some latent, high-dimensional space via linear ODEs, or the neural ODE model of Chen et al. (2018) that learns latent albeit nonlinear ODEs. The latter has in fact been modified to suit very different imputation and "smoothing" scenarios (Rubanova et al., 2019; Yildiz et al., 2019; Norcliffe et al., 2021; Seifner & Sanchez, 2023). Other works depart from ODEs and assume the dynamics are stochastic. Examples thereof include the works by Fortuin et al. (2020) and Fang et al. (2024), which leverage Gaussian processes to represent the time evolution, and that by Tashiro et al. (2021), which instead utilizes conditional score-based diffusion models. Researcher have also recently dispensed with continuous-time models altogether, and deployed self-attention mechanisms to impute missing data (Du et al., 2023). Wang et al. (2024) provides a recent and comprehensive review on these and many other imputation methods, and we refer the reader to it for completeness.

Regardless of whether they rely on ODEs or not, all the models above find themselves under the umbrella of the classical paradigm, insofar as they are all optimized with respect to a single empirical process. There are, nevertheless, two recent exceptions. Similar to us, Becker et al. (2023) and d'Ascoli et al. (2024) generate large datasets of ODE systems and their (noisy) observations. Opposite to us, they attempt to explicitly infer the vector fields, and do so in symbolic form. The work of Becker et al. (2023) is however limited to one dimensional ODEs, whereas that of d'Ascoli et al. (2024) is limited to six-dimensional ones. To the best of our knowledge, we present *the first zero-shot solution that is applicable to real-world empirical processes of any dimensionality*.

## 3 FOUNDATION INFERENCE MODELS FOR DYNAMICAL SYSTEMS

In this section, we introduce a novel methodology for *zero-shot imputation* of missing time series data. Let the series $\mathbf{y}^*(\tau_1), \ldots, \mathbf{y}^*(\tau_l)$ correspond to a sequence of $l$ observations on some $D$-dimensional empirical process $\mathbf{y}^*(t)$, where each observation is represented by a vector $\mathbf{y}^*(\tau_j) \in \mathbb{R}^D$, with $j = 1, \ldots, l$. Suppose that some of the components of these observation vectors are missing, and the goal is to impute them back. Our proposal frames the problem of estimating these missing values as an inference task, in which one seeks to infer the ODE solution $\mathbf{x}(t)$ that best *interpolates* the series $\mathbf{y}^*(\tau_1), \ldots, \mathbf{y}^*(\tau_l)$, *and thus imputes its missing values*.

The classical formulation of the imputation problem typically involves different missing patterns and in this work we focus on two of them. The first one is the so-called *point-wise missing pattern*, where individual vectors in the series randomly lack some of their components. The second one is the *temporal missing pattern*, where certain components of the vectors in the series are missing over consecutive observation times. To handle them, we make the following two simple assumptions.

---

[3]We choose to name our model foundation model because it goes in line with the definition proposed by Bommasani et al. (2021). To wit: a foundation model is any model that is trained on broad data (generally using self-supervision at scale) that can be adapted to a wide range of downstream tasks.

First, we assume that for every time series of observations *featuring point-wise missing patterns*, one can always find a certain time scale $\tau_{\text{simple}}$, or some (sequential) subset of observations, for which the best interpolating ODE solution is "simple"[4]. Furthermore, we assume that the set of all such simple parametric functions can be well-represented by a heuristically constructed synthetic distribution. Second, we assume that time series *featuring temporal missing patterns* involve more complex interpolating functions, meaning that no such $\tau_{\text{simple}}$ is to be found in this case. Although more complex in nature, we assume that these functions are *locally* "simple", and that they often exhibit generic secular and seasonal structures, which encode important information about the missing values and can be well-represented by a second, synthetic distribution over parametric functions. Should these two general assumptions hold true, a model trained to infer both our synthetic set of "simple" parametric functions, and that of functions exhibiting generic, secular and seasonal structures, from noisy and sparse observations on them, will *automatically interpolate any unseen sequence of empirical observations and, consequently, impute all of its missing values*. In the experimental section below, we empirically demonstrate that this is indeed the case in a variety of scenarios.

Our methodology thus consists of two components. The first comprises a synthetic data generation model, and encodes our beliefs about both the "simple" ODE solutions that interpolate the data locally, and the more complex ones that feature global, generic structures. The second corresponds to a neural recognition model of minimal inductive biases, that maps sets of noisy observations onto the space of parametric functions. In what follows, we delve into the details of these two components and name our recognition model as Foundation Inference Model (FIM) for dynamical systems. Figure 1 illustrates the FIM framework.

## 3.1 Synthetic Data Generation Model

In this subsection, we describe the synthetic data generation model we use to sample a large and varied set of ODE solutions, together with their noisy and sparse observations. Given that every ODE solution is a parametric function of time, and that each component of any such $D$-dimensional function is itself a one-dimensional, parametric function of time, *we focus only on the space of $1D$ time series*. In other words, we opt for a channel independent strategy (Nie et al., 2023; Han et al., 2024). Let us then define the probability of observing the $1D$ noisy time series $y_1, y_2, \ldots, y_l$ at the observation times $0 \leq \tau_1 < \tau_2 < \cdots < \tau_l \leq 1$ — which might correspond to the values taken by any of the components of some $D$-dimensional process — as

$$\prod_{i=1}^{l} p_{\text{noise}}(y_i | x_i, \sigma) p(\sigma) \delta \left( x_i - x_0 - \int_0^{\tau_i} f(s) ds \right) p_{\text{grid}}(\tau_1, \ldots, \tau_l, \eta_g) p(f, \eta_f) p(x_0), \quad (2)$$

where $\delta(\cdot)$ represents the Dirac delta function, which identifies the ODE solution $x(t)$, evaluated at time $\tau_i$, with $x_0 + \int_0^{\tau_i} f(s) ds$. That is, we understand our ODE solutions as being determined by some initial condition $x_0$ and the parametric function $f(t)$, which represents the time derivative of $x(t)$. Note that we use the notation $x_i$ to denote $x(\tau_i)$, and that we denote both random variables and their values with the same symbol. Let us now specify each term in Eq. 2, starting from the right.

DISTRIBUTION OVER INITIAL CONDITIONS. We define the prior $p(x_0)$ over initial conditions as a standard Gaussian distribution. A standard Gaussian suffices, because *the values* of every time series we process are first normalized to lie on the unit interval (see Section 3.2.1).

DISTRIBUTION OVER PARAMETRIC FUNCTIONS OF TIME. The prior distribution $p(f, \eta_f)$ factorizes as $p(f|\eta_f)p(\eta_f)$, with $p(\eta_f)$ the prior over the hyperparameter set $\eta_f$. The conditional distribution $p(f|\eta_f)$ is a distribution over the space of parametric functions of time, *defined on the unit interval*. It encodes our beliefs about the class of interpolating functions we expect to find in practice. Indeed, as we briefly motivated earlier, we design two such distributions. One represents "simple" parametric functions that are assumed to be typical interpolating functions imputing point-wise missing patterns at the characteristic time scale $\tau_{\text{simple}}$. The other represents functions that are locally simple, but that exhibit secular and seasonal structures at longer time scales (*i.e.* $\tau \gg \tau_{\text{simple}}$). Structures that, in turn, are assumed to carry crucial information about temporal missing patterns.

---

[4]By simple we loosely mean functions that can be approximated by low-degree polynomials, and functions whose Fourier transform has support at low frequencies only.

We define these distributions by means of random Chebyshev expansions and Gaussian processes with different kernels, and refer the reader to Appendix B.1 for details.

DISTRIBUTION OVER OBSERVATION TIMES. We define the prior distribution $p_{\text{grid}}(\tau_1, \ldots, \tau_l, \eta_g)$, with hyperparameter set $\eta_g$, to represent missing data patterns that we expect to be relevant in real-world imputation tasks. Again, we consider two such distributions. The first one represents point-wise missing patterns and allows for regular and irregular observation grid instances with different observation count. The second one represents temporal missing patterns and combines point-wise patterns with randomly located observation gaps. We allow the latter to be as large as one-third of the unit interval. Appendix B.2 provides details regarding our implementations.

DISTRIBUTION OVER NOISE PROCESSES. When recording any empirical process, one typically only has access to the mean square error of those measurements. According to the maximum entropy principle, a Gaussian distribution is the best guess one can make about the noise distribution — actually, it is the most likely distribution — given the available information (that is, given those first two moments) (Jaynes, 2003). We choose our noise model $p_{\text{noise}}(y_i | x_i, \sigma)$ accordingly, and set $p(\sigma)$ to also be a Gaussian distribution of zero mean and variance $10^{-1}$.

We use the generative model, Eq. 2 above, to sample a large and varied set of noisy and sparse ODE solutions. We refer the reader to Appendix B.4 for details on the specifics of the sampling procedure.

## 3.2 FOUNDATION INFERENCE MODEL

In this subsection, we introduce a recognition model that exploits ideas from neural operators (Lu et al., 2021; Kovachki et al., 2023) to map time series data onto parametric functions. Indeed, given a set of noisy observations $(y_1, \tau_1), \ldots, (y_l, \tau_l)$ on some ODE solution $x(t)$ — sampled from the data generation model, Eq. 2 above — our goal is to infer both the $1D$ parametric function $f(t)$ and initial condition $x_0$ that specified $x(t)$ in the first place. In other words, we want to reverse the data generation process. Below, we first introduce a neural interpolation model that is trained to infer the distribution $p(f|\eta_f)$ over "simple" functions, from time series exhibiting point-wise missing patterns. Note that in this setting $\tau_{\text{simple}}$ is, by construction, of order one. Later, we introduce a second interpolation model that is trained to infer $p(f|\eta_f)$ over functions that are locally "simple" but display global structures at time scales $\tau \gg \tau_{\text{simple}}$, from time series characterized by temporal missing patterns. In what follows, we denote the first interpolation model with FIM$-\ell$, for it handles *local* dynamic features, and use FIM to refer to the the second one.

### 3.2.1 FIM FOR INTERPOLATING POINT-WISE MISSING PATTERNS

In order to interpolate time series featuring point-wise missing patterns, and values of every scale, we first need to normalize every input sequence and rescale their target parametric functions accordingly (see Appendix C.1 for details). Let us label the set of normalized, noisy observations with $\mathcal{Y}$, the space of rescaled $1D$ parametric functions with $\mathcal{F}$ and that of rescaled initial conditions with $\mathcal{X}_0$. Our interpolation problem can then be understood as the problem of mapping $\mathcal{Y}$ onto both $\mathcal{F}$ and $\mathcal{X}_0$. We begin with the first of these two maps.

Let us use $\phi^\theta$ and $\psi^\theta$ to denote feedforward (FFN) and sequence processing neural networks, respectively. Let us also denote the trainable network parameters with $\theta$. We now define the function

$$\mathbf{h}^\theta(t) = \phi_3^\theta(\mathbf{u}^\theta, \phi_1^\theta(t)), \ \text{with } \mathbf{u}^\theta = \phi_2^\theta(\psi_1^\theta(y_1, \tau_1, \ldots, y_l, \tau_l)), \tag{3}$$

where $\phi_1^\theta$ and the composition $\phi_2^\theta \circ \psi_1^\theta$ can be interpreted as the trunk and branch nets of DeepONets (Lu et al., 2021), while $\mathbf{u}^\theta$ is the **context vector** encoding the (context) points $(y_1, \tau_1, \ldots, y_l, \tau_l)$. Given the vector-valued, parametric function $\mathbf{h}^\theta(t)$, we now define the mean and variance of a Gaussian random variable *over the possible values* of the estimated time derivative as follows

$$\hat{f}(t) = \phi_4^\theta(\mathbf{h}^\theta(t)), \quad \log \text{Var}(\hat{f})(t) = \phi_5^\theta(\mathbf{h}^\theta(t)), \tag{4}$$

where, similar to the works of Lakshminarayanan et al. (2017) and Valdenegro-Toro & Mori (2022), the variance $\text{Var}(\hat{f})$ is used to represent the *model's uncertainty in the estimation of* $f(t)$ (see *e.g.* Figure 3 for an illustration). Next, to perform the map between $\mathcal{Y}$ and $\mathcal{X}_0$, we also model the initial condition $\hat{x}(0)$ as a Gaussian random variable, whose mean and variance are given by

$$\hat{x}_0 = \phi_6^\theta(\mathbf{u}^\theta), \quad \log \text{Var}(\hat{x}_0) = \phi_7^\theta(\mathbf{u}^\theta). \tag{5}$$

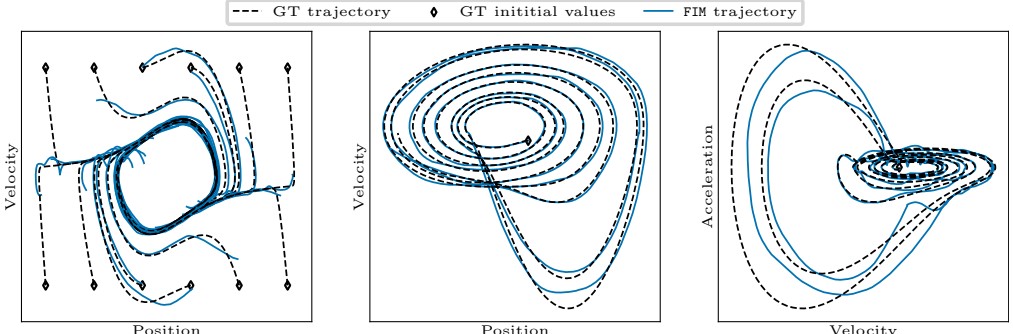

Figure 2: Zero-shot phase portrait reconstruction of two dynamical systems. *Left*: Van der Pol oscillator. *Center and Right*: Rössler attractor in the position-velocity and velocity-acceleration planes, respectively.

Equations 4 and 5 allow us to express the function interpolating the time series $(y_1, \tau_1), \ldots, (y_l, \tau_l)$ — *at any desired time* $\tau \in (0, 1)$ — as $\hat{x}_0 + \int_0^\tau \hat{f}(s)ds$. We train FIM$-\ell$ in a supervised fashion, to maximize the log-likelihood of Eqs. 4 and 5, and minimize the one-step reconstruction error of the integrated solution, both with respect to the distribution over "simple" functions. Thus, FIM$-\ell$ can be understood as an estimator of the *posterior distribution over the space of interpolating functions, given the context points* $(y_1, \tau_1, \ldots, y_l, \tau_l)$. We refer the reader to Appendices C.2 and C.3 for details regarding the model architecture and training objective.

### 3.2.2 BEYOND SIMPLE FUNCTIONS: PROCESSING DATA OF ANY LENGTH AND DIMENSIONALITY WITH FIM$-\ell$

Let us briefly comment on how to use our (pretrained) FIM$-\ell$ to process time series of any length and dimensionality. We provide further details (and limitations) of our strategies' implementation in Appendix C.6. During training, FIM$-\ell$ processes only $1D$ time series with at least $L_{\min}$ and at most $L_{\max}$ observations (see Appendix B.2). We define $L_{\min}$ as the minimum number of *context points* FIM generally needs to function, as shorter time series are considered *out-of-distribution*. Similarly, time series with more than $L_{\max}$ observations also fall outside the distribution, not just because of their length, but because their (hidden) interpolating function may not be well-represented by our distribution of "simple" functions. To address this limitation, we first split any target time series longer[5] than $L_{\max}$ into successive and overlapping time windows, ensuring that both their *observation count and dynamic features remain within distribution*. That is, we assume that each time window spans a time scale of order $\mathcal{O}(\tau_{\text{simple}})$. Then, we combine the local FIM$-\ell$ estimates obtained from each window into a global one for the entire target time series (see Appendix C.6). In an analogous manner, we adopt a channel independent strategy and process each component of any target, $D$-dimensional process independently with FIM$-\ell$.

We empirically demonstrate the efficacy of these approaches in Section 4 below. We also investigate the effect of changing the length (and number) of the overlapping windows in Appendix F.4.4.

### 3.2.3 FIM FOR INTERPOLATING TEMPORAL MISSING PATTERNS

In this subsection, we tackle the interpolation of time series displaying temporal missing patterns by decoupling trends and seasonality from local fluctuations. Suppose we are given a noisy time series with $l$ observations, whose values and observation times lie within the unit interval. Suppose that this time series has been split into $K$ sequential (*i.e.* ordered) sets

$$y_1, \ldots, y_{w_1} \cup y_{w_1+1}, \ldots, y_{w_1+w_2} \cup \cdots \cup y_{l-w_K+1}, \ldots, y_l, \qquad (6)$$

where $w_k$ is the number of observations within the $k$th set. Suppose now that the $q$th set is missing, and the goal is to impute it back. See *e.g.* the center image in Figure 1.

---

[5]Note that this approach can also be applied to time series of length shorter than $L_{\max}$, but whose dynamic features are believed to be out-of-distribution.

Table 1: MAE of the inferred time derivative $\dot{\mathbf{x}}(t)$ and ODE solution $\mathbf{x}(t)$ on ODEBench. The standard deviation is calculated across 10 samplings of the corruption schemes.

| Model | Inferred time derivative $\dot{\mathbf{x}}(t)$ | | Reconstructed ODE solution $\mathbf{x}(t)$ | |
|---|---|---|---|---|
| | $\gamma = 0$ | $\gamma = 0.05$ | $\gamma = 0$ | $\gamma = 0.05$ |
| ODEFormer | $8.00 \pm 0.40$ | $7.90 \pm 0.60$ | $1.18 \pm 0.05$ | $1.16 \pm 0.05$ |
| FIM-$\ell$ | $\mathbf{2.44 \pm 0.05}$ | $\mathbf{3.79 \pm 0.05}$ | $\mathbf{0.17 \pm 0.01}$ | $\mathbf{0.34 \pm 0.01}$ |

We assume that *locally*, within every set (even the missing one), the functions underlying the data are well represented by our synthetic distribution of simple functions. That is, we assume that each set spans a time scale of the order $\mathcal{O}(\tau_{\text{simple}})$ and thus, that its underlying function can be modelled well with our *pretrained* FIM-$\ell$. We also assume that beyond those time scales ($\tau \geq \tau_{\text{simple}}$), there exist *inter-set, global* structures and correlations that carry information about the missing (*i.e. q*th) set. Our task is to define a model that encodes precisely this information. In other words, we require a model that extends the context of FIM-$\ell$ to longer time scales.

Since FIM-$\ell$ is trained to deal with normalized data, we first normalize each of the available sets, and denote with $s_j$ the statistics containing information about the local scale (*i.e.* the norms) of the $j$th set (see Appendix D for details). Let us now process each (normalized and available) set with our *pretrained* encoding networks $\phi_2^\theta \circ \psi_1^\theta$, to obtain the sequence

$$(\mathbf{u}_1^\theta, s_1), \ldots, (\mathbf{u}_{q-1}^\theta, s_{q-1}), (\mathbf{u}_{q+1}^\theta, s_{q+1}), \ldots, (\mathbf{u}_K^\theta, s_K) \text{ with } \mathbf{u}_j^\theta = \phi_2^\theta(\psi_1^\theta(y_1, \tau_1, \ldots, y_{w_j}, \tau_{w_j})),$$

where, for simplicity, we relabelled the sub-indices (of the $j$th set) on the right hand side as $m \leftarrow \sum_{i=1}^{j-1} w_i + m$, with $m$ an integer between 1 and $w_j$. Our second interpolation model FIM consists of a sequence processing network $\psi_2^\varphi$, with trainable parameter set $\varphi$, that computes

$$\mathbf{u}_q^\varphi = \psi_2^\varphi((\mathbf{u}_1^\theta, s_1), \ldots, (\mathbf{u}_{q-1}^\theta, s_{q-1}), (\mathbf{u}_{q+1}^\theta, s_{q+1}), \ldots, (\mathbf{u}_K^\theta, s_K)), \quad (7)$$

*i.e.* the context vector for the missing (*i.e. q*th) set. At this point, we can use the *pretrained* $\phi_1^\theta, \phi_3^\theta, \phi_4^\theta$ and $\phi_5^\theta$ networks of FIM-$\ell$ to estimate the function $f(t)$ and its variance *along the gap*. That is

$$\hat{f}(t) = \phi_4^\theta(\mathbf{h}^\varphi(t)), \ \log \text{Var}(\hat{f})(t) = \phi_5^\theta(\mathbf{h}^\varphi(t)), \ \text{with } \mathbf{h}^\varphi(t) = \phi_3^\theta(\mathbf{u}_q^\varphi, \phi_1^\theta(t, \theta)). \quad (8)$$

We optimize $\varphi$ in a supervised manner — *while keeping $\theta$ fixed* — to maximize the likelihood of $\hat{f}(t)$ along the missing set, with respect to our synthetic dataset of complex functions that feature global, generic structures and temporal missing patterns. We refer the reader to Appendix D, where we provide additional details and discuss about how to integrate $\hat{f}(t)$ to infer $\hat{x}(t)$ along the gap.

## 4 EXPERIMENTS

In this section, we test our methodology on widely different imputation tasks, which involve datasets of varying complexity, different dimensionalities and noise signals of very varied nature. We use FIM-$\ell$ and FIM to impute — *in zero-shot mode* — missing data featuring point-wise and temporal missing patterns, respectively. To be precise, we apply our pretrained models directly to the test sets of the target datasets, *without any parameter fine-tuning*. FIM-$\ell$ was pretrained to infer 2M ODE solutions from time series with $(L_{\min}, L_{\max})$ set to $(4, 128)$. FIM was pretrained to infer 500K local ODE solutions from time series with observation gaps that amounted to one-third of the data. Additional information regarding model architecture and hyperparameters, training details and ablation studies can all be found in Appendices C and D.

METRICS. Below we evaluate the performance of our models wrt. the mean-absolute error (MAE). In the Appendix, we also report our results wrt. the root-mean-square error (RMSE) and, in some cases, the $R^2$ coefficient of determination and the mean-relative error (MRE). Formulas for these metrics can be found in Appendix E.

BASELINES. Depending on the task, we compare our findings against the (symbolic) ODE-Former (d'Ascoli et al., 2024) model; the LatentODE (Chen et al., 2018), NeuralODEProcesses (Norcliffe et al., 2021) and BRITS (Cao et al., 2018) models; the GP-VAE (Fortuin et al.,

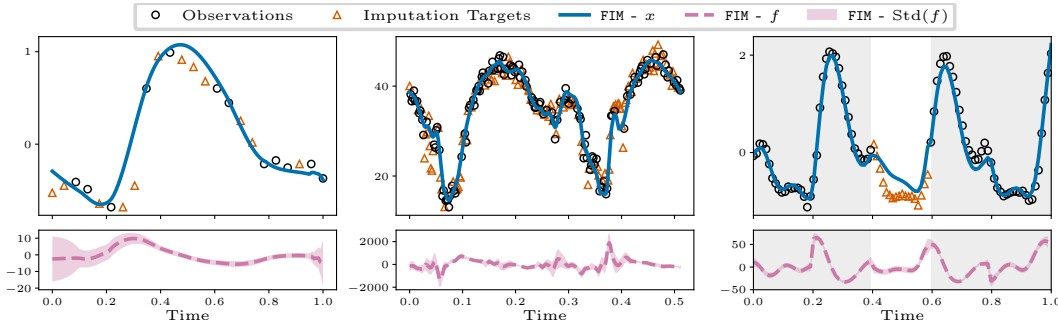

Figure 3: Zero-shot imputation with FIM. *Left and Center*: Point-wise missing imputation in (one dimension of) the *Beijing* and *GuangZhou* datasets, respectively. (*Right*): Temporal missing imputation in single PCA dimension of the *Motion Capture* dataset.

2020) and the score-based CSDI (Tashiro et al., 2021) models; the (self-attention) SAITS (Du et al., 2023) model; and the (Gaussian process) nPODE (Heinonen et al., 2018) model, among others. Besides ODEFormer, *all other baselines are trained on their target datasets*.

### 4.1 PHASE PORTRAIT RECONSTRUCTION

Before testing our methodology in real-world imputation scenarios proper, we explore its ability to accurately reconstruct the phase portrait of complex dynamical systems, using noisy and sparse observations on only one of their coordinates.

Phase portraits are geometric representations of the trajectories of dynamical systems in phase space. They help unveil attractor sets or limit circles and allow to determine, inter alia, how chaotic the systems in question might be (Benettin et al., 1976). In the seminal paper by Packard et al. (1980), the authors demonstrated that one can obtain a faithful phase-portrait representation of any $D$-dimensional dynamical system, through one of its coordinates, say $x_1(t)$, and all its time derivatives $(\dot{x}_1(t), \ddot{x}_1(t), \dots)$ up to order $D - 1$. Carrying out such a reconstruction from noisy and sparse observations on $x_1(t)$ alone, not only entails interpolating the data, but also numerically computing the time derivatives of the interpolating function. In this section, we empirically demonstrate that one can use the hierarchical structure underlying FIM — which treats the time derivatives $\dot{x}_1(t)$ ($f(t)$ in our notation) as a latent variable — to reconstruct, in a zero-shot fashion, the phase portrait of complex dynamical systems.

Suppose we simulate a dynamical system to obtain the solution $\mathbf{x}(t)$ over the interval $[0, T]$, represented on a fine-grid of $L$ points. We then introduce two types of data corruption. The first is multiplicative noise $y_{ij} = (1 + \epsilon)x_{ij}$, with $\epsilon \sim \mathcal{N}(0, \gamma)$, where $i = 1 \dots D$ and $j = 1 \dots L$. The second is random subsampling, where a fraction $\rho$ of the fine-grid is removed (and thus corresponds to a point-wise missing data pattern). For concreteness, we set $T = 10$ and $\gamma = 0.05$. The task is to reconstruct the phase portrait of the dynamical system from the 1D time series $y_{11}, \dots, y_{1l}$ alone.

Let us start with the (nonlinear) Van der Pol oscillator, whose dynamics are given by the second-order ODE $\ddot{x}_1 + \mu(x_1^2 - 1)\dot{x}_1 + x_1 = 0$. This system features a limit circle around $|x_1| = 1$, which gets distorted as one increases the strength of the nonlinear term $\mu$. We set $\mu = 0.5$, simulate the system with 12 different initial conditions and record only the noisy position of the oscillator 128 times per trajectory. The left panel of Figure 2 shows the 12 trajectories inferred by FIM−$\ell$ on $(\dot{x}_1, x_1)$ phase space, together with the ground-truth, on a *plotting grid* of 2048 points, which amounts to an imputation of 1920 missing points. The agreement is good and the visible deviations are due to small discrepancies in the estimation of the oscillator's velocity, for some initial conditions. We suspect that such rapidly changing functions are not well-represented by our synthetic distribution of "simple" functions. Next we consider the Rössler system, which is a $3D$ dynamical system that features a chaotic attractor, and record up to 2048 noisy observations on $x_1(t)$ in order to discern its main features. The center panel of Figure 2 displays the trajectory inferred by FIM−$\ell$ in $(\dot{x}_1, x_1)$ space, on a plotting grid of 8192 points, which is in very good agreement with the ground-truth. Similarly, the right panel of the same figure displays the inferred trajectory in $(\ddot{x}_1, \dot{x}_1)$ space,

Table 2: MAE (at missing values) on 8 datasets featuring $50\%$ *point-wise missing patterns*. Baselines scores for *GuangZhou* and *Solar* are extracted from Fang et al. (2024). The rest are extracted from Du et al. (2024). See Appendix H for standard deviations and additional baselines.

| | Air quality | | | Traffic | | | Electricity | |
| Method | Beijing | Italy | GuangZhou | PeMS | Pedestrian | Solar | ETT_h1 | Electricity |
| --- | --- | --- | --- | --- | --- | --- | --- | --- |
| BRITS | 0.169 | 0.321 | 3.335 | **0.287** | 0.259 | 1.985 | 0.238 | 1.124 |
| SAITS | 0.194 | 0.285 | 3.391 | 0.302 | **0.205** | 1.827 | **0.223** | 1.399 |
| GP-VAE | 0.258 | 0.453 | 3.419 | 0.346 | 0.451 | 1.810 | 0.414 | 1.099 |
| CSDI | **0.144** | 0.958 | 3.202 | 0.288 | 0.351 | 0.804 | 0.318 | 0.798 |
| BayOTIDE | - | - | 2.687 | - | - | 0.734 | - | - |
| FIM$-\ell$ | 0.166 | **0.215** | **2.427** | 0.365 | 0.273 | **0.595** | 0.279 | **0.083** |

*obtained by applying* FIM$-\ell$ *twice to the noisy observations on* $x_1(t)$. Overall, the agreement is still strong and FIM$-\ell$ only struggles to fit some very rapid changes in the acceleration function, which likely lie out-of-distribution. We provide details of these computations and explore the systems further in Appendices F.1 and F.2, respectively.

The discussion above yielded a qualitative picture of the inference capabilities of FIM$-\ell$. To obtain a more quantitative perspective, we now compare FIM$-\ell$ against the ODEFormer model of d'Ascoli et al. (2024). ODEFormer is trained *offline* to infer the vector field $\mathbf{f}(\mathbf{x})$ of dynamical systems in symbolic form. We estimate $\dot{\mathbf{x}}(t)$ with ODEFormer by first solving their inferred ODE, to obtain an estimate of the ODE solution $\mathbf{x}(t)$, and then evaluating their inferred vector field along their estimated $\hat{\mathbf{x}}(t)$. As target dataset we analyse ODEBench, which was also introduced by d'Ascoli et al. (2024). ODEBench consists of 63 autonomous ODEs of different dimensionalities (specifically, 23 $1D$, 28 $2D$, 10 $3D$ and 2 $4D$ equations) and their solutions. The latter are represented on a fine-grid of 512 points. We set $\rho = 0.5$ (which defines point-wise missing patterns) and let $\gamma$ be either 0 or 0.05. We then compute the MAE of ODEFormer and FIM$-\ell$ on the estimation of both $\dot{\mathbf{x}}(t)$ and $\mathbf{x}(t)$ with respect to the ground-truth trajectories across all 63 target ODEs. Table 1 reports our results averaged over all ODEs and shows that FIM$-\ell$ outperforms ODEFormer in every case.

Some final remarks are in order. First, d'Ascoli et al. (2024) originally report $R^2$ scores in their paper. We also report these per dimension in Table 9 of Appendix F.3. Our conclusions remain unchanged. Second, these results also reveal that, despite being trained on additive noise only, FIM$-\ell$ can handle multiplicative noise well. Third, we empirically demonstrate that FIM$-\ell$ is also superior to LatentODE trained on the iconic Lorenz systems in Appendix F.4. Fourth, we also demonstrate that FIM$-\ell$ outperforms the very recent NeuralODEProcesses model on low-data regimes in Appendix F.5. All together, these results reflect the capabilities of FIM$-\ell$ to not only impute point-wise missing data in dynamical systems, but also accurately reconstruct their phase portraits, *both in zero-shot mode*. Finally, Appendix I contains preliminary results on fine-tuning FIM.

## 4.2 IMPUTATION OF POINT-WISE MISSING PATTERNS

In this subsection we evaluate FIM$-\ell$ on 8 real-world, high-dimensional datasets featuring *point-wise missing patterns*. Specifically, we study the *Guangzhou* dataset, which contains traffic speed records with 214 channels and 500 observations, and the *Solar* dataset, which consists of solar-power generation records with 137 channels and 52560 observations. We obtained the (preprocessed) datasets from Fang et al. (2024). We also study two popular air quality datasets from *Beijing* and *Italy*, which have 132 channels with 1458 observations, and 13 channels with 774 observations, respectively; the *Electricity* and *ETT-h1* datasets of electricity consumption, common in forecasting studies, with 370 (1457) and 7 (358) channels (observations) each; and two additional traffic-related datasets, namely the *PeMS* dataset of road occupancy with 862 channels and 727 observations, and the single-channel *Pedestrian* dataset, which reports pedestrian activity in Australia and consists of 3633 observations. We obtained this second set of 6 (preprocessed) datasets from Du et al. (2024).

After being split into train, validation and test sets, fifty percent of these subsets is randomly removed, defined as missing and set aside for evaluation. We only make use of the (available 50% of the) test subsets with FIM$-\ell$. Figure 3 illustrates the type of *zero-shot* ODE solutions inferred by FIM$-\ell$ on the *Beijing* and *Guangzhou* datasets (left and center). The bottom panels portrait instead

Table 3: MAE (at missing values) on Motion Capture (MC) and Navier-Stokes datasets featuring *temporal missing patterns* of 20%. The large error bars in the MC dataset have been reported before (Heinonen et al., 2018). (Cubic) spline(F) includes a Savgol filter.

| Model | Motion Capture | | Navier Stokes | |
|---|---|---|---|---|
| | PCA | No PCA | PCA | No PCA |
| LatentODE | **1.658 ± 0.989** | - | 0.076 ± 0.030 | - |
| (Cubic) spline | 3.362 ± 1.175 | 4.209 ± 1.436 | 0.085 ± 0.003 | 0.083 ± 0.003 |
| (Cubic) spline(F) | 2.897 ± 0.871 | 2.998 ± 0.881 | 0.084 ± 0.000 | 0.075 ± 0.002 |
| FIM | **1.765 ± 0.627** | **1.611 ± 0.453** | **0.063 ± 0.003** | **0.051 ± 0.002** |

the inferred (time) derivatives of the interpolating functions, together with the confidence of the model, which increases in regions with high, local information. Table 2 reports the average MAE at the missing values for all models. Remarkably, $FIM-\ell$ outperforms all baselines in 4 out of 8 datasets, comes second in one, and third in two, which suggests that *there is indeed enough local information to perform the data imputation with "simple" functions in these use cases*. We close this subsection by referring the reader to Appendix G, where we report the scores of additional, albeit less common baselines, as well as our results on the data splits investigated by Du et al. (2023).

### 4.3 IMPUTATION OF TEMPORAL MISSING PATTERNS

In this subsection, we look into the harder problem of imputing time series data featuring *temporal missing patterns*. Indeed, we explore the problem setup proposed by Heinonen et al. (2018), in which (about) 20% of the data from the middle of the time series is missing completely. More precisely, we consider their human Motion Capture dataset, which consists of 43 trajectories, each with 50 channels and 100 observations. We also apply their setup to the Navier-Stokes simulation of Course & Nair (2023), which instead contains 596602 channels. To be able to handle the (high) dimensionality of the datasets, Heinonen et al. (2018) projects the data to latent space using PCA, and trains their models to perform the imputation there (see Appendix H for details). We follow their methodology and train a LatentODE model, to outperform the results reported by Heinonen et al. (2018). Additionally, we compare against a naive spline interpolation.

Having set the stage, we leverage our pretrained FIM to infer the ODE solution that best imputes the missing data in PCA space, and report the MAE at the missing values, after projecting back to data space, in Table 3. Since FIM (and spline) can be applied to each channel independently, we also report the MAE we obtain by imputing the data directly in high-dimensional (data) space. Again, *one and the same* FIM performs comparably to (or even better than) LatentODE in the Motion Capture dataset, and outperforms it on the Navier-Stokes case. Note that imputing the data in high-dimensional space helps FIM avoid the errors introduced by the PCA projections. The right panel of Figure 3 illustrates the class of *zero-shot* ODE solutions inferred by FIM in the Motion Capture dataset, and demonstrates how FIM leverages the global, seasonal structures outside the gap to impute the (locally simpler) missing data. The agreement is strong.

## 5 CONCLUSIONS

In this work, we introduced a novel methodology for *zero-shot imputation* of time series data, whose underlying dynamics are assumed to be governed by ordinary differential equations (ODEs). We empirically demonstrated that *one and the same* Foundation Inference Model (FIM) can impute datasets of any dimensionality, featuring noise signals of very different nature, even in cases which do not naturally admit a description in terms of ODEs. In fact, we showed that FIM often outperforms SOTA models that are trained to the target distributions.

*The main limitation* of our methodology is clearly imposed by our synthetic distributions. Evaluating FIM on empirical datasets whose distribution significantly deviates from our synthetic distribution will inevitably yield poor estimates. The left panel of Figure 2 provides such an example. Indeed, for some initial conditions, the velocity of the Van der Pol oscillator features rapid changes that are not well represented by our pretraining distribution. *Future work* shall extend our decoupling of local and global features of the imputation model to the problem of *zero-shot forecasting*.

## 6 REPRODUCIBILITY STATEMENT

The Appendix includes all information required to reproduce our presented methods and results. Let us therefore provide a short overview of its contents, focused on the parts related to reproducibility.

An important part of our methodology is our *synthetically generated training dataset*, which Appendix B is dedicated to. The generation and hyperparameter choices are described extensively in Appendix B.1 (distributions over functions), Appendix B.2 (observation grids) and Appendix B.3 (noise processes). Appendix B.4 schematizes the whole generation algorithm.

Appendix C includes all details related to FIM-$\ell$, our model for *point-wise missing pattern imputation*. In particular, Appendix C.2 describes the complete model architecture, including hyperparameters of our single trained FIM-$\ell$ model, and model inputs and outputs. Its training procedure is discussed in Appendix C.4, including hyperparameters for the optimizer and computing resources, whereas Appendix C.3 states the training objective.

The details for FIM, our model for *temporal missing pattern imputation*, are given in Appendix D. Appendix D.2 describes the complete model architecture, where we also include the hyperparameters used for our single trained FIM model. The training details are given in Appendix D.4, including hyperparameters of the optimizer and computing resources, and Appendix D.3 states the associated training objective.

Equations of the *evaluation metrics* used for comparisons in our experiments, and remarks about their usage in the different applications, are provided in Appendix E. Training details and computing resources for *LatentODE*, one of our baselines, are stated (for each dataset individually) in Appendices F.4.2, H.1.2 and H.2.2.

The *datasets in our experiments* are addressed individually in their own subsection in Appendices F, G and H. Each subsection contains either generation hyperparameters or links to their sources, and the applied pre-processing steps.

Finally, our pretrained model, repository and tutorials are available online[6].

## ACKNOWLEDGMENTS

This research has been funded by the Federal Ministry of Education and Research of Germany and the state of North-Rhine Westphalia as part of the Lamarr-Institute for Machine Learning and Artificial Intelligence.

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

## A  COMPARISON WITH OTHER META-LEARNING APPROACHES

[The discussion that follows took place during the rebuttal process].

Conditional neural network models such as the *Neural Statistician* (Edwards & Storkey, 2016; Hewitt et al., 2018), or members of the *Neural Process* family (Garnelo et al., 2018b;a; Kim et al., 2019) and their later extensions to dynamical process (Singh et al., 2019; Wang et al., 2022; Jiang et al., 2023), consider the problem of **(meta)training** a single model on a set of different, albeit related datasets $\mathcal{D}_1, \ldots, \mathcal{D}_m$, each of which is characterized by a corresponding (that is, shared) context latent variable $c_1, \ldots, c_m$.

Main difference with FIM:

1. The first and most important difference between all these approaches and ours is that they need to be trained on datasets from their target domains, which makes both their inferred representations and optimized weights "problem specific". Indeed, every one of these works focuses, by construction, on meta-learning among similar systems (or datasets) only. To illustrate, let us consider the work of Jiang et al. (2023). They first forecast bouncing ball simulations under different gravity conditions. For them, each gravity corresponds to a different dataset, and their inferred representations must encode their corresponding gravities (or at least some aspects of them). Later, they consider forecasting turbulent flow dynamics under different buoyant forces and train their model anew, in order to encode the forces characterizing each dataset. Thus, the parameters of the neural network building of their model are *not the same* for the bouncing ball experiment and the turbulent flow experiment. Similarly, the semantic content of the shared representations inferred by their model is *not the same* for the bouncing ball experiment and the turbulent flow experiment. In the former case it corresponds to gravity, whereas in the latter it corresponds to buoyant force.

   *In sharp contrast*, our proposal is not problem specific, because it entirely relies on the two general assumptions (inductive biases, or prior knowledge) of Section 3. These simple assumptions pertain to the nature of interpolating functions only, and are therefore independent of the actual data generation mechanisms underlying our target datasets. Given that our model is only trained on a synthetic dataset encoding these two assumptions, it can only recognize patterns that help it reconstruct the best interpolating function given the available (context) data, regardless of the underlying data generation mechanism. In other words, our method maintains the same network parameters and representation semantics throughout all experiments. The representations consistently correspond to the time derivative and initial conditions of the hidden interpolating function, regardless of the target dataset.

2. A second important difference lies in how these works leverage their prior knowledge for transfer. Specifically, they assume that their target domains allow for the collection of their different yet related datasets $\mathcal{D}_1, \ldots, \mathcal{D}_m$, on which to train their (meta)models.

   We instead fully rely on our two assumptions of Section equation 3, and therefore on our synthetic dataset which encodes them.

3. A third difference is that all these works assumed there is a shared latent variable $c_m$ encoding the main features of the $m$th dataset $\mathcal{D}_m$ in their collection. Most of these works rely on neural variational inference to optimize their encoder-decoder pairs, and infer their representations in an unsupervised manner (see however Wang et al. (2022)).

   In our proposal our latent variables are not shared, for each time series has associated to it a single interpolating function. Transfer learning takes place because of the general assumptions we encode into our synthetic dataset.

## B  SYNTHETIC DATA GENERATION MODEL: SPECIFICS

### B.1  ON THE DISTRIBUTION OVER PARAMETRIC FUNCTIONS OF TIME

In this section we define two distributions over the space of parametric functions. We use the first one to train our FIM for interpolation. We use the second one to train our extended FIM for imputation.

### B.1.1 Distribution Over "Simple" Interpolation Functions

To train our model for *point-wise missing patterns*, we define $p(f|\eta_f)$ either via Gaussian processes (GP) with Radial Basis Function (RBF) kernels (Williams & Rasmussen, 1995; Vert et al., 2004), or as truncated Chebyshev expansions, whose coefficients and degree are both randomly sampled. Each alternative has different $\eta_f$ hyperparameters.

*In the case of RBF kernels*, there is a single free hyperparmeter, the *lengthscale* $\eta_f$, which controls the scale at which variations take place. A small $\eta_f$ results in parametric functions with short-range fluctuations, whereas larger $\eta_f$ produce smoother functions that capture broader trends. We define $p(\eta_f)$ as a mixture of two Beta distributions, with equal mixing coefficients. To wit

$$p(\eta_f) = \frac{1}{2}\text{Beta}(\eta_f|2, 10) + \frac{1}{2}\text{Beta}(\eta_f|2, 5). \tag{9}$$

Here, the first component returns functions with faster change (*i.e.* higher frequency), whereas the second component returns smoother functions.

*In the case of* (the $M$-order) *truncated Chebyshev expansions*, we write

$$f(t) = \sum_{m=1}^{M} a_m T_m(t), \tag{10}$$

where $T_m(t)$ is the $m$th Chebyshev polynomial with (real) coefficient $a_m$. Now, in order to generate a random parametric function with Eq. 10, we sample both the degree $M$ and the set of coefficients $\{a_1, \ldots, a_M\}$ from the prior $p(\eta_f)$. Intuition says one would like high-order polynomials to occur rarely. It also says one would like the scale of their coefficients to be small (see also the work of *e.g.* Chan et al. (2022)). We therefore (implicitly) define the distribution $p(\eta_f)$ over hyperparameters as

$$(a_1, \ldots, a_M) \sim \mathcal{N}\left(0, \frac{1}{M}\right), \text{ with } M \sim \text{Zipf}(2). \tag{11}$$

In practice, we generate a synthetic dataset of 2M parametric functions (200K for the test set), each of which is evaluated on a *fine grid of $L_{max} = 128$ points*. This fine grid is defined regularly on the unit interval $[0, 1]$. Half of this dataset consists of random Chebyshev expansions. The other half consists of parametric functions sampled via GPs.

### B.1.2 Distribution Over Imputation Functions with Global Patterns

To train our model for *temporal missing patterns*, we require data with *trends* and *seasonality*, such that our model learns to use these (global) patterns to impute the missing data. We hence define $p(f|\eta_f)$ as again via GPs, but opt for Periodic kernels, which exhibit the required *seasonality*.

Periodic kernels are specified by two free hyperparameters: the *lengthscale*, denoted by $\eta_f^l$, and the *period*, denoted by $\eta_f^p$. The lengthscale determines the local fluctuations of the function, as described in Appendix B.1.1. The period determines the frequency of repetitions of the function, *i.e.* the seasonality.

We sample both hyperparameters independently. In other words, our distribution over Periodic kernel GPs $p(\eta_f)$ factorizes as $p(\eta_f) = p(\eta_f^l, \eta_f^p) = p(\eta_f^l)p(\eta_f^p)$. We define uniform distributions for both hyperparameters:

$$p(\eta_f^l) = \mathcal{U}(\eta_f^l|[0.75, 1])$$
$$p(\eta_f^p) = \mathcal{U}(\eta_f^p|[0.3, 0.5])$$

We generate a dataset of 500k parametric functions (50K for the test set). Each function is evaluated on a regular *fine grid of $L_{\max} = 256$ points* on the unit interval $[0, 1]$.

Note that we do not need to explicitly address *trends* in the parametric functions over time. A sample from the defined GPs will (*a.s.*) have a non-zero mean. During data generation (see Appendix B.4), these deviations will accumulate over time, such that the corresponding ODE solutions exhibit trends naturally. In combination with instance normalization during model input processing (see Appendix D.1), this setup covers a variety of trends.

## B.2 On the Distribution Over Observation Grids

In this section we define two distributions over observation grids. We use the first one to train our FIM for interpolation. We use the second one to train our extended FIM for imputation.

### B.2.1 Observation Grids Encoding Point-wise Missing Patterns

We define the prior distribution over observation grids $p_{\text{grid}}(\tau_1, \ldots, \tau_l, \eta_g)$ in a way that it allows for both regular and irregular observation grid instances, each with at most $L_{\max}$ and at least $L_{\min}$ observations. Note that the latter number defines the *minimum number of context points needed* for FIM to function.

Let us recall that our parametric functions $f(t)$ are evaluated on a fine grid of 128 points (see Appendix B.1 above) over the unit interval $[0, 1]$. This fine grid is subsequently subsampled randomly, to define the random observation times $\tau_1, \ldots, \tau_l$. We employ two subsampling schemes — *regular* and *irregular* — which occur with equal probability within our entire dataset.

**Regular scheme**. In this scheme the observation times are obtained from the fine grid by strides of regular length, where the stride length is sampled from the Uniform($[1, 2, \ldots, 16]$) distribution. Note that this distribution defines $p(\eta_g)$.

**Irregular scheme**. In this scheme the observation times are instead defined via Bernoulli masks, whose survival probabilities (here denoted by $\eta_g$) are sampled from a categorical distribution $p(\eta_g)$. The latter is defined over the set $\{0.0625, 0.25, 0.5\}$ with class probabilities $\{0.5, 0.25, 0.25\}$, respectively.

Note that we ensure that there are at least $L_{\min} = 8$ observations in each time series, by rejecting observation grid instances with less than eight points. In contrast, there are time series with at most $L_{\max} = 128$ observations.

### B.2.2 Observation Grids Encoding Temporal Missing Patterns

To train our model for *temporal missing patterns*, the observation grids of our training data must exhibit such patterns. We sample such grids in a two-step process.

In the *point-wise step*, as for the observation grids sampled in Appendix B.2.1, we only employ point-wise subsampling schemes. These are required for our model can handle irregular grid time series data of variable lengths.

In the *temporal step* we introduce the *temporal missing patterns*, by dropping a consecutive range of observations. We train our model to impute these dropped values.

More formally, let $\tau_i$ denote times in the *point-wise* observation grid and $\tilde{\tau}_i$ denote times in the *temporal missing pattern* observation grid. Then their joint distribution depends on two hyperparameters, $\eta_g^{\text{point}}$ and $\eta_g^{\text{temp}}$, and factorizes as

$$p_{\text{grid}}(\tilde{\tau}_1, \ldots, \tilde{\tau}_k, \tau_1, \ldots, \tau_l, \eta_g^{\text{temp}}, \eta_g^{\text{point}}) = \\ p_{\text{temp}}(\tilde{\tau}_1, \ldots, \tilde{\tau}_k | \tau_1, \ldots, \tau_l, \eta_g^{\text{temp}}) p(\eta_g^{\text{temp}}) p_{\text{point}}(\tau_1, \ldots, \tau_l | \eta_g^{\text{point}}) p(\eta_g^{\text{point}}).$$

where $\{\tilde{\tau}_i\}_{i=1}^k \subseteq \{\tau_i\}_{i=i}^l$ .

Let us now report the specifics for both steps:

**Point-wise step**. One half of the observation grids are generated with a *regular* subsampling scheme, sampling stride lengths from Uniform($[1, 2, 3, 4]$). The other half is generated with a *irregular* subsampling scheme, sampling from a Bernoulli distribution with survival probability $0.5$.

**Temporal step**. To generate a missing pattern in the time series, we first sample the position of the missing pattern from a uniform distribution, specifically Uniform($1, 3$), which determines the window that will be masked. The length of the missing pattern is randomly sampled from a uniform distribution in the range $[10, 30]$. The remaining part of the time series is then split into four equal windows.

### B.3 ON THE DISTRIBUTION OVER NOISE PROCESSES

Let $x : \mathbb{R}_+ \to \mathbb{R}$ be a time-dependent trajectory. We define the distribution over noise processes on observations $x(t) \in \mathbb{R}$ of such trajectory to factorize, such that the noise process is fixed for each trajectory. More formally, denoting the noise process by $\sigma$ and the noisy trajectory by $y$, we assume

$$p_{\text{noise}}(y(t), x(t), \sigma(t)) = p_{\text{noise}}(y(t) \mid x(t), \sigma)p(\sigma) \text{ for all } t \in \mathbb{R}.$$

As outlined in Section 3.1, we use additive Gaussian noise

$$p_{\text{noise}}(y(t) \mid x(t), \sigma) = \mathcal{N}(x(t), \sigma)$$

for our synthetic datasets. Here, the noise process is defined only by a single value: the standard deviation $\sigma$ of a Gaussian distribution. Hence, our choice of a distribution over noise processes reduces to a choice of a suitable distribution for $\sigma$. We use $p(\sigma) = \mathcal{N}(0, \lambda)$ in our synthetic datasets, interpreting negative samples for $\sigma$ as their absolute value[7].

The value of $\lambda$ differs between our two synthetic datasets. In our dataset for *point-wise missing patterns*, we use $\lambda = 0.1$ and in our dataset for *temporal missing patterns*, we use $\lambda = 0.05$. We reduce $\lambda$ for the temporal-missing patterns to retain more structure of the trajectory, such that the model can learn to impute based on the structure of the time series.

### B.4 ON THE GENERATION OF THE SYNTHETIC DATASET

In this subsection we schematize the data generation process. Underlying this process is a sampling procedure of noisy time series data, based on samples of random functions, observation grids and noise processes. The corresponding distributions are described in Appendices B.1-B.3.

To generate the $j$th instance of our synthetic datasets, we utilise these distributions in the following generation steps:

1. Sample a *function*
$$f_j \sim p(f|\eta_{f,j}), \quad \text{where} \quad \eta_{f,j} \sim p(\eta_f),$$
and record its values $\{f_j(t_i)\}_{i=1}^L$ on the *fine grid* $\{t_i\}_{i=1}^L$, the regular grid on the unit interval $[0, 1]$.

2. Sample a *initial value* $x_{0j} \sim \mathcal{N}(0, 1)$ that defines the ODE solution
$$x_j(t) = x_{0j} + \int_0^t f_j(s)ds$$
and record its values $\{x_j(t_i)\}_{i=1}^L$ on the *fine grid*.

3. Sample a *observation grid*
$$\tau_{1j}, \dots, \tau_{lj} \sim p_{\text{grid}}(\tau_1, \dots, \tau_l|\eta_{g,j}), \quad \text{where} \quad \eta_{g,j} \sim p(\eta_g),$$
that defines a time series with (clean) observations $\{(x_{ij} = x(\tau_{ij}), \tau_{ij})\}_{i=1}^l$ of the ODE solution.

4. Sample *noisy observations*
$$y_{ij} \sim p_{\text{noise}}(y \mid x_{ij}, \sigma_j), \quad \text{where} \quad \sigma_j \sim p(\sigma),$$
of the ODE solution that define the time series $\{(y_{ij}, \tau_{ij})\}_{i=1}^l$.

Note that each step consists of i.i.d. samples of their respective (hierarchical) distributions. Thus, each instance of our synthetic datasets is also i.i.d..

## C FIM FOR LOCAL INTERPOLATION: ADDITIONAL DETAILS

In this section, we begin with instance normalization – in section C.1 – as a pre-processing step for handling times series of varying scales. We then outline the training objective in section C.3, focusing on the loss function design. Next, we discuss methods for processing time series of any length and dimensionality (section C.6).

---

[7]This distribution is also called a folded Gaussian distribution.

### C.1 INPUT PRE-PROCESSING AND OUTPUT POST-PROCESSING

To accommodate time series of all scales, we employ min-max-normalization for the observation values and times *per time series* before processing them with $\texttt{FIM}-\ell$ and renormalize the model outputs accordingly. To express this instance normalization in equations, let $(y_1, \tau_1), \ldots, (y_l, \tau_l)$ be a set of noisy observations, following the notation of Section 3.2.1 and denote

$$\tau_{\min} = \min_{i=1,\ldots,l} \tau_i, \quad \tau_{\max} = \max_{i=1,\ldots,l} \tau_i, \quad y_{\min} = \min_{i=1,\ldots,l} y_i, \quad y_{\max} = \max_{i=1,\ldots,l} y_i \quad . \tag{12}$$

Before applying $\texttt{FIM}-\ell$ as described in Section 3.2.1, we replace the inputs by their normalized values:

$$y_i \leftarrow \frac{y_i - y_{\min}}{y_{\max} - y_{\min}}, \quad \tau_i \leftarrow \frac{\tau_i - \tau_{\min}}{\tau_{\max} - \tau_{\min}} \tag{13}$$

Let $\hat{f}(t)$, $\log \mathrm{Var}(\hat{f})(t)$, $\hat{x}_0$ and $\mathrm{Var}(\hat{x}_0)$ be the outputs of $\texttt{FIM}-\ell$, following Equations 4 and 5 of Section 3.2.1, given the normalized inputs. Under the renormalization maps

$$y \leftarrow (y_{\max} - y_{\min})y + y_{\min}, \quad t \leftarrow (\tau_{\max} - \tau_{\min})t + \tau_{\min} \tag{14}$$

the model outputs are transformed to the original time and value scale as follows:

$$\hat{f}(t) \leftarrow \hat{f}(t)\frac{y_{\max} - y_{\min}}{\tau_{\max} - \tau_{\min}} \tag{15}$$

$$\log \mathrm{Var}(\hat{f})(t) \leftarrow \log \mathrm{Var}(\hat{f})(t) + 2\log \frac{y_{\max} - y_{\min}}{\tau_{\max} - \tau_{\min}} \tag{16}$$

$$\hat{x}_0 \leftarrow \hat{x}_0(y_{\max} - y_{\min}) + y_{\min} \tag{17}$$

$$\log \mathrm{Var}(\hat{x}_0) \leftarrow \log \mathrm{Var}(\hat{x}_0) + 2\log(y_{\max} - y_{\min}) \tag{18}$$

While the transformations of $\hat{x}_0$ and $\log \mathrm{Var}(\hat{x}_0)$ follow immediately from the value renormalization map, the transformations of $\hat{f}(t)$ and $\mathrm{Var}(\hat{f})(t)$ follow from the linearity of the derivative (in case of the value renormalization map) and the chain rule (in case of the time renormalization map).

This described instance normalization approach enables $\texttt{FIM}-\ell$ to be used in a range of (real-world) applications, as we show in our experiments (Section 4). But there are also limitations, which we will now address.

Firstly, some *signal-to-noise ratio patterns* are not accurately resolved because of this normalization. For example, observations of (almost) constant functions with additive noise are not interpolated very accurately, because the normalization parameters are mainly determined by the noise, not the underlying signal.

Secondly, the internal embeddings of $\texttt{FIM}-\ell$ do not contain any *information about the scale* of the input. Downstream tasks, like *imputation of temporal missing patterns* could benefit from this information. In such scenarios, we opted for separate embeddings of statistics about the input data scales, to reintroduce them alongside the $\texttt{FIM}-\ell$ embeddings (see Appendix D.2).

### C.2 MODEL ARCHITECTURE

Let us provide more details about the interpolation model architecture, which was already outlined in Section 3.2.1, and hyperparameters of the single $\texttt{FIM}-\ell$ model all experimental results were derived with. Note that we use SeLU (Klambauer et al., 2017) as activation for all feed-forward neural networks and employ a dropout rate of 0.1.

In total, $\texttt{FIM}-\ell$ has roughly 20 million learnable parameters. The results of an ablation study on architecture and hyperparameter choices are described in Appendix C.5.

**Model inputs**:

(i) A noisy time series $(y_1, \tau_1), \ldots, (y_l, \tau_l)$ with observation values $y_i \in \mathbb{R}$ and ordered, but potentially irregular, observation times $\tau_i \in \mathbb{R}_+$ with $\tau_1 < \cdots < \tau_l$.

(ii) A time $t \in \mathbb{R}_+$ at which to evaluate the functions $\hat{f}$ and $\log \mathrm{Var}(\hat{f})$ at.

**Model architecture**:

(i) *Temporal embedding.* We use the learnable time embedding from Shukla & Marlin (2020) with output in $\mathbb{R}^{512}$. Its $i$th dimension is defined as

$$\phi_0^\theta(t)[i] = \begin{cases} w_0 t + b_0 & \text{if } i = 0 \\ \sin(w_i t + b_i) & \text{otherwise} \end{cases}$$

where $w_i$ and $b_i$ are learnable parameters.

(ii) *Trunk net equivalent*: We use the time embedding $\phi_0^\theta$ in combination with a feed-forward neural network $\phi_1^\theta$, with 4x1024 hidden layers and output in $\mathbb{R}^{512}$, to encode the evaluation time $t$. The composition $\phi_1^\theta \circ \phi_0^\theta$ can thus be understood as the Trunk net of DeepOnet. Let us denote

$$\mathbf{t}^\theta = (\phi_1^\theta \circ \phi_0^\theta)(t)$$

(iii) *Individual observations embedding.* Let us denote the $i$th element of the time series in our input with

$$\mathbf{y}_i^\theta = \text{Concat}(y_i, \phi_0^\theta(\tau_i)).$$

(iv) *Branch net equivalent.* The embedded time series processing network, which can be understood as the branch net of DeepONets, is a composition of a sequence processing network $\psi_1^\theta$ and a feed-forward neural network $\phi_2^\theta$. The sequence processing network $\psi_1^\theta$ is defined as a bi-directional LSTM with hidden states in $\mathbb{R}^{512}$. The feed-forward neural network $\phi_2^\theta$ with 4x1024 hidden layers and outputs in $\mathbb{R}^{512}$. Let us denote

$$\mathbf{u}^\theta = \phi_2^\theta(\psi_1^\theta(\mathbf{y}_1^\theta, \dots, \mathbf{y}_l^\theta)).$$

(v) *Time derivative projection.* We combine the output of the trunk net and branch net equivalent networks via a feed-forward neural network $\phi_3^\theta$ with 4x1024 hidden layers and outputs in $\mathbb{R}^{512}$. Let us denoted its output by

$$\mathbf{h}^\theta(t) = \phi_3^\theta(\text{Concat}(\mathbf{t}^\theta, \mathbf{u}^\theta)).$$

The Gaussian distribution over the values of the estimated time derivative is parameterized by two linear projections from $h^\theta(t)$ to $\mathbb{R}$, denoted by

$$\hat{f}(t) = \phi_4^\theta(\mathbf{h}^\theta(t)) \quad \text{and} \quad \log \text{Var}(\hat{f})(t) = \phi_5^\theta(\mathbf{h}^\theta(t)).$$

(vi) *Initial condition projection*: Parameters of the Gaussian distribution for the initial condition

$$\hat{x}_0 = \phi_6^\theta(\mathbf{u}^\theta) \quad \text{and} \quad \log \text{Var}(\hat{x}_0) = \phi_7^\theta(\mathbf{u}^\theta)$$

are modeled with feed-forward neural networks $\phi_6^\theta$ and $\phi_7^\theta$ each with 4x1024 hidden layers and outputs in $\mathbb{R}$.

**Model outputs**: To summarize, our model returns

(i) a *time derivative* $\hat{f}(t)$, with uncertainty estimate $\log \text{Var}(\hat{f})(t)$, for a time $t \in \mathbb{R}_+$ and

(ii) a *initial value* $\hat{x}_0$, with uncertainty estimate $\log \text{Var}(\hat{x}_0)$.

Combined, these outputs define a function $\hat{x}(\tau)$ interpolating the noisy (input) time series $(y_1, \tau_1), \dots, (y_l, \tau_l)$:

$$\hat{x}(\tau) = \hat{x}_0 + \int_0^\tau \hat{f}(s) ds$$

## C.3   TRAINING OBJECTIVE

The parameter set $\theta$ of `FIM-ℓ` is trained in a supervised fashion, utilising the synthetic dataset for *point-wise missing patterns* introduced in Appendix B.

Following the notation of Appendix B.4, let $f$ be the sampled function from one instance in the synthetic dataset, $\{f(t_i)\}_{i=1}^L$ its values on the fine grid, $x_0$ the initial value of the ODE solution, recorded on the fine grid $\{x(t_i)\}_{i=1}^L$, and $\{(y_i, \tau_i)\}_{i=1}^l$ the associated noisy time series. Using $\{(y_i, \tau_i)\}_{i=1}^l$ as inputs for `FIM-ℓ`, we denote its outputs by $\hat{f}$, $\log \text{Var}(\hat{f})$, $\hat{x}_0$ and $\log \text{Var}(\hat{x}_0)$ as in Appendix C.2.

The training objective of `FIM-ℓ` consists of three parts:

(i) Maximizing the Gaussian log-likelihood $\log \mathcal{L}(f(t_i) \mid \hat{f}(t_i), \text{Var}(\hat{f})(t_i))$ of the *time derivative* at all $L$ points $t_i$ of the fine grid.

(ii) Minimizing the one-step-ahead reconstruction error $|x(t_{i+1}) - (x(t_i) + \hat{f}(t_i)(t_{i+1} - t_i))|$ of the *ODE solution* at $L - 1$ points on the fine grid.

(iii) Maximizing the Gaussian log-likelihood $\log \mathcal{L}(x_0 \mid \hat{x}_0, \text{Var}(\hat{x}_0))$ of the *initial value*.

Expressed as an equation, we train $\mathtt{FIM}-\ell$ to minimize

$$\mathcal{L} = \underset{f \sim p(f, \eta_f)}{\mathbb{E}} \left\{ \sum_{i=1}^{L} \frac{(f(t_i) - \hat{f}(t_i))^2}{2\text{Var}(\hat{f})(t_i)} + \frac{1}{2} \log(\text{Var}(\hat{f})(t_i)) \right\}$$

$$+ \underset{\substack{f \sim p(f, \eta_f) \\ x_0 \sim p(x_0)}}{\mathbb{E}} \left\{ \sum_{i=1}^{L-1} |x(t_{i+1}) - (x(t_i) + \hat{f}(t_i)(t_{i+1} - t_i))| \right\}$$

$$+ \underset{x_0 \sim p(x_0)}{\mathbb{E}} \left\{ \frac{(x_0 - \hat{x}_0)^2}{2\text{Var}(\hat{x}_0)} + \frac{1}{2} \log(\text{Var}(\hat{x}_0)) \right\}.$$

Note that the one-step-ahead reconstruction error is simply the absolute error of a single step of the Euler method, when using the *inferred* time derivative $\hat{f}(t_i)$, but the *ground-truth* starting point $x(t_i)$. Such term is only viable in our supervised training regime on synthetically generated data, where we have access to the ground-truth ODE solution $x$.

## C.4    TRAINING PROCEDURE AND IMPLEMENTATION

The parameters $\theta$ of $\mathtt{FIM} - \ell$ were optimized with AdamW (Loshchilov & Hutter, 2017), using a learning rate of $1e^{-6}$ and weight decay $1e^{-4}$. Using a batch size of 1024, the loss on a validation set converged after approximately 500 epochs.

We used four A100 80GB GPUs to train $\mathtt{FIM}-\ell$. The implementation is done in Jax[8]. Its code and the trained model weights are provided in the supplementary material.

## C.5    ABLATION STUDIES: ARCHITECTURE AND DATASET DESIGN

In this section, we present ablation studies of the design of the synthetic training dataset dataset for and the architecture of $\mathtt{FIM}-\ell$. We evaluate the ablations by the $R^2$-accuracy on the ODEBench dataset, corrupted with $\rho = 0.5$, $\sigma = 0.05$ (see Appendix F.3 for more details). It is important to note that these models are trained *only* on synthetic data, and *not* on the ODEBench dataset, in accordance with our zero-shot approach.

First, we experiment with the size of our model (2M, 20M, 50M parameters) and the number of trajectories in the the training dataset (2M, 8M trajectories) and summarize the results in Table 4. For 2M trajectories, increasing the number of parameters from 2M to 50M does increase the performance by around 4%. For 8M trajectories, the trend persists with the 20M parameter model, but reverts with the 50M parameter model, resulting in no significant improvement. The architecture specifics for the models of different sizes follows :

(i) 2M parameters: MLP: 4x256, hidden-dim: 256;

(ii) 20M parameters: MLP: 4x1024, hidden-dim: 512;

(iii) 50M parameters: MLP: 3x2048, hidden-dim: 1024.

Next, we experiment with the architecture of the sequence processing network $\psi_1^\theta$, considering BiLSTM and transformer networks. The architecture specifics in this case are:

(i) Transformer: MLP: 4x256, N-of-layers: 4, QKV-dim: 256, N-of-heads 8, output dim: 256, total-parameter-count: 20M;

---

[8]https://jax.readthedocs.io/en/latest/index.html

(ii) BiLSTM: MLP: 4x1024, hidden-dim: 512 (i.e. 256 per each directions), total-parameter-count: 20M.

The results are presented in Table 5 and show that there is no significant difference in performance when evaluated on the ODEBench dataset.

Table 4: $R^2$ accuracy with standard deviation, ablating parameter count and number of time series in train set.

| Parameters | Time Series in Train Set | |
| | 2M | 8M |
| --- | --- | --- |
| 2M | $82.6 \pm 0.7$ | $82.3 \pm 0.9$ |
| 20M | $87.3 \pm 1.3$ | $86.2 \pm 1.0$ |
| 50M | $87.7 \pm 2.0$ | $84.1 \pm 1.3$ |

Table 5: $R^2$ Accuracy with standard deviation, ablating the architecture of the sequence processing network $\psi_1^\theta$.

| $\psi_1^\theta$ Architecture | $R^2$ Accuracy (%) |
| --- | --- |
| BiLSTM | $87.30 \pm 1.35$ |
| Transformer | $87.38 \pm 1.42$ |

An additional study we conducted involves varying the size of the maximum and minimum training contexts points in a time series of the synthetic train set, which we denote by $L_{\max}$ and $L_{\min}$. The results for ablating $L_{\max}$, presented in Table 6, indicate that, at least on the ODEBench dataset, the model performs better with more context points. However, the results are within error bars. Similarly, the results for ablating $L_{\min}$, shown in Table 7, reveal that the model performs better when only trained on time series with more observations. Again, this result is only valid for the ODEBench dataset, as other applications *require small context windows*, e.g. because there are too few observations in the time series.

Table 6: $R^2$ Accuracy with standard deviation, ablating $L_{\max}$, the maximum number of observations in a time series of the synthetic train dataset.

| $L_{\max}$ | $R^2$ Accuracy (%) |
| --- | --- |
| 128 | $87.3 \pm 1.3$ |
| 64 | $86.7 \pm 1.2$ |

Table 7: $R^2$ accuracy with standard deviation, ablating $L_{\min}$, the minimum number of observations in a time series of the synthetic train dataset.

| $L_{\min}$ | $R^2$ Accuracy (%) |
| --- | --- |
| 8 | $87.3 \pm 1.3$ |
| 32 | $89.0 \pm 1.0$ |

## C.6 Processing Time Series of Any Length and Dimensionality

As a *zero-shot* inference model, FIM$-\ell$ should be able to cope with dynamic phenomena of very diverse nature. In particular, it should handle observations with different dimensionalities, lengths and scales. Our approach to handle different scales with instance normalization was discussed in Appendix C.1. Now we want to address our approach to the other two phenomena and offer some limitation of these approaches.

### C.6.1 Composition across Dimensions

To handle different (in fact, arbitrary) dimensional observations, we process each feature of a time series separately with our pretrained model FIM$-\ell$. The synthetic training dataset only contains 1D time series (see Appendix B.1) and the model architecture only returns 1D time derivatives (see Appendix C.2). Such channel independent strategy has been used previously, e.g. by Nie et al. (2023) and Han et al. (2024).

Regarding the inference problem of the *time derivative on a ODE solution path*, this reduction to 1D systems is *exact*, not an approximation. Indeed, consider a vector field $\mathbf{f} : \mathbb{R}^+ \times \mathbb{R}^D \to \mathbb{R}^D$ and the solution $\mathbf{x} : \mathbb{R}^+ \to \mathbb{R}^D$ to some initial value problem $(x_0, \mathbf{f})$. Then its time derivative, i.e. the vector field along the solution path, $\mathbf{f}(t, \mathbf{x}(t))$ is a purely time-dependent function, which naturally splits into $D$ *independent* coordinate functions $(\mathbf{f}_1(t, \mathbf{x}(t)), \ldots, \mathbf{f}_D(t, \mathbf{x}(t)))$.

Coordinates of datasets generated from *real-world scenarios*, which we consider in our experiments, might not be independent or uncorrelated. Still, we show that our approach can perform well in a va-

riety of such scenarios. However, extreme cases like the PHYSIONET2012 dataset (Du et al., 2024), with very sparse observations in some coordinates, are a natural limitation of channel independent approaches like ours.

### C.6.2 COMPOSITION ALONG TIME

Time series in the training set of $\mathtt{FIM}{-}\ell$ are bounded in length by a fixed upper bound $L_{max}$, as described in Section 3.1. Therefore, $\mathtt{FIM}{-}\ell$ can only (reasonably) process time series which lengths do not exceed this upper bound. Similarly, $\mathtt{FIM}{-}\ell$ can only (reasonably) infer time derivatives approximately contained in the broad distributions over parametric functions of time, specified in Appendix B.1.

Datasets from *real-world* scenarios will likely not adhere to these limitations derived from our synthetic training dataset.

To adapt the *pretrained* $\mathtt{FIM}{-}\ell$ to handle such scenarios, we employ a *windowing scheme*. Concretely, we split the time series into successive, overlapping windows that are still processable by the $\mathtt{FIM}{-}\ell$. Applying (instances of) $\mathtt{FIM}{-}\ell$ to each window individually yields local estimates of the solution $\hat{x}(t)$.

As windows are processed separately, instance normalization (see Appendix C.1) and corresponding renormalization occurs individually at their respective scales.

To get a global estimate out of the local estimates, one can combine them on the overlaps with two approaches.

1. *Interpolating the ODE solutions.* Consider two successive, overlapping time windows $A$ and $B$, defined on the intervals $[t_0^A, t_1^A]$ and $[t_0^B, t_1^B]$, respectively, so that

$$0 \leq t_0^A < t_0^B < t_1^A < t_1^B.$$

   Applying an instance of $\mathtt{FIM}{-}\ell$ to each window yields local solutions $\hat{x}_A(t)$ and $\hat{x}_B(t)$ in the respective windows. By defining

$$\hat{x}(t) = \begin{cases} \hat{x}_A(t), & \text{if } t < t_0^B \\ \frac{t_1^A - t}{t_1^A - t_0^B}\hat{x}_A(t) + \frac{t - t_0^B}{t_1^A - t_0^B}\hat{x}_B(t), & \text{if } t_0^B \leq t \leq t_1^A \\ \hat{x}_B(t), & \text{if } t > t_1^A \end{cases} \quad (19)$$

   for $t \in [t_0^A, t_1^B]$, we combine $\hat{x}_A(t)$ and $\hat{x}_B(t)$ to a global solution $\hat{x}(t)$ with linear interpolation on the overlap. The time derivative $\hat{f}$ corresponding to $\hat{x}$ can be found (*a.e.*) by taking the derivative of equation 19.

2. *Interpolating the time derivatives and solving the ODE.* Alternatively we could interpolate the estimated time derivatives $\hat{f}_A(t)$ and $\hat{f}_B(t)$ of the time windows $A$ and $B$, just as done in Eq. 19 but with $\hat{x}$ replaced with $\hat{f}$. Once such an interpolated time derivative is available one can integrate the ODE over the complete interval $[t_0^A, t_1^B]$. However, this second approach accumulates the errors of the ODE solutions of each interval.

In practice, and for all experiments described in Section 4 and the Appendix, we used the first approach.

**Notation**. There are two methods for specifying the overlapping windows:

1. Windows specified by the *number of windows* that cover the observations. For $m$ windows we write $\mathtt{FIM}{-}\ell(w.n. = m)$, identifying this method.

2. Windows specified by the *number of observations* they contain. For $m$ observations per windows, we write $\mathtt{FIM}{-}\ell(o.n. = m)$, identifying this method.

**Limitations**. Our experiments in Section 4 show that our compositional approach along time can be quite effective. Still, there are natural limitations, which we want address.

Firstly, although our distributions over parametric functions of time are broad, they do not cover all local patterns exhibited by real-world time series. For example, $\mathtt{FIM}{-}\ell$ struggles with high-frequency time series with sparse observations. Many windows are required to accurately interpolate

such patterns. Future work will explore the design of the synthetic training data distributions, to cover more of these patterns with $\text{FIM}-\ell$

Secondly, (subsequent) temporal missing patterns can not be accurately imputed with the windowing approach. As windows are processed individually, and their outputs merged manually on the overlaps, no information is passed between them. Therefore, imputation that requires global, long range patterns can not be resolved. We handle the *temporal missing pattern imputation* with an additional imputation module in $\text{FIM}$ (see Appendix D).

# D  FIM FOR INTERPOLATING TEMPORAL MISSING PATTERNS: ADDITIONAL DETAILS

## D.1  INPUT PREP-PROCESSING AND OUTPUT POST-PROCESSING

Analogous to the local interpolation model discussed in Appendix C, our zero-shot model for interpolating temporal missing patterns $\text{FIM}$ should accommodate time series of all scales. Hence, we again employ min-max-normalization for the observation values *per time series*, as described for $\text{FIM}-\ell$ in Appendix C.1.

Our approach to the *temporal missing pattern* imputation task involves one more step of instance normalization, which we want to address now. For this sake, let us briefly recall the notation introduced in Section 3.2.3. A (ordered) time series with observations $\{(y_i, \tau_i)\}_{i=0}^{l}$ is split into $K$ sequentially ordered sets

$$y_1, \ldots, y_{w_1} \cup y_{w_1+1}, \ldots, y_{w_1+w_2} \cup \cdots \cup y_{l-w_l+1}, \ldots, y_l,$$

where the $q$th set is empty, meaning it contains no observations, but defines a consecutive time range, which we call the *time gap*. Our goal is to impute values in this gap.

Recall that one part of our model (described in Section 3.2.3 and Appendix D.2) applies a *pretrained* $\text{FIM}-\ell$ to all sets, except the $q$th. Internally, $\text{FIM}-\ell$ normalizes each set *locally*, processes it and renormalizes its outputs accordingly. In the crucial part of our approach, the model predicts an embedding $\mathbf{u}_q$ for the $q$th set, which is processe with $\text{FIM}-\ell$, to return $\hat{f}$, $\log \text{Var}(\hat{f})$ and $\hat{x}_0$ *inside the gap*.

It is only possible to *locally renormalize* these outputs wrt. the time range of the gap, because there are no observation values. The *global* renormalization is not effected by this deficit, because on this scale, the predicted output at the gap is renormalized with the *global normalization parameters*, equivalent to the outputs at all other sets.

Because the model is subjected to this deficit during training, and inputs are instance normalized globally, the model learns to adjust to these conditions.

## D.2  MODEL ARCHITECTURE

Let us provide more details about the architecture of $\text{FIM}$, the imputation model for *temporal missing patterns*, which was already outlined in Section 3.2.3 and hyperparameters of the single $\text{FIM}$ model all experimental results were derived with. Following the model architecture of $\text{FIM}-\ell$ (see Appendix C.2), we use SeLU (Klambauer et al., 2017) as activation for all feed-forward neural networks and employ a dropout rate of 0.1.

$\text{FIM}$ contains a pretrained $\text{FIM}-\ell$ model, which parameters $\theta$ are frozen. Additionally, $\text{FIM}$ has roughly 6M trainable pararameters, which we denote by $\varphi$.

**Model inputs**:

(i) A noisy time series $(y_1, \tau_1), \ldots, (y_l, \tau_l)$ with observation values $y_i \in \mathbb{R}$ and ordered, but potentially irregular, observation times $\tau_i \in \mathbb{R}_+$ with $\tau_1 < \cdots < \tau_l$.

(ii) An ordered splitting of the interval $[\tau_1, \tau_l]$ into $K = 5$ sequentially ordered sets, which induces a splitting of the observations into $K$ sets

$$y_1, \ldots, y_{w_1} \cup y_{w_1+1}, \ldots, y_{w_1+w_2} \cup \cdots \cup y_{l-w_l+1}, \ldots, y_l,$$

where the $q$th set only defines a consecutive time range (or *imputation gap*) and does not contain any actual observations.

(iii) A time $t \in \mathbb{R}_+$ in the imputation gap at which to evaluate the functions $\hat{f}$ and $\log \mathrm{Var}(\hat{f})$ at.

**Model architecture**:

(i) *Local scale statistics.* Defining $w_0 = 0$ and $w_j^{\mathrm{prev}} = \sum_{k=0}^{j-1} w_k$ for convenience, we set

$$y_j^{\mathrm{min}} = \min_{i=1,\ldots,w_j} y_{w_j^{\mathrm{prev}}+i} \qquad y_j^{\mathrm{max}} = \max_{i=1,\ldots,w_j} y_{w_j^{\mathrm{prev}}+i} \qquad y_j^{\mathrm{range}} = y_j^{\mathrm{max}} - y_j^{\mathrm{min}}$$

$$y_j^{\mathrm{first}} = y_{w_j^{\mathrm{prev}}+1} \qquad y_j^{\mathrm{last}} = y_{w_j^{\mathrm{prev}}+w_j} \qquad y_j^{\mathrm{diff}} = y_j^{\mathrm{last}} - y_j^{\mathrm{first}}$$

$$\tau_j^{\mathrm{first}} = \tau_{w_j^{\mathrm{prev}}+1} \qquad \tau_j^{\mathrm{last}} = \tau_{w_j^{\mathrm{prev}}+w_j} \qquad \tau^{\mathrm{diff}} = \tau_j^{\mathrm{last}} - \tau_j^{\mathrm{first}}$$

and denote by

$$s_j = [y_j^{\mathrm{min}}, y_j^{\mathrm{max}}, y_j^{\mathrm{range}}, y_j^{\mathrm{first}}, y_j^{\mathrm{last}}, y_j^{\mathrm{diff}}, \tau_j^{\mathrm{first}}, \tau_j^{\mathrm{last}}, \tau_j^{\mathrm{diff}}] \in \mathbb{R}^9$$

the statistics about the *position* and *local scale* for the $j$th set, where $j \neq q$.

(ii) *Local scale embedding.* A learnable linear layer $\phi_8^\varphi$ embedds the local scale statistics $s_j$ for all $j \neq q$. Let us denote

$$\mathbf{s}_j^\varphi = \phi_8^\varphi(s_j) \in \mathbb{R}^{512}.$$

As mentioned in Appendix D.1, we also have access to the boundary times of the imputation gap: $\tau_q^{\mathrm{first}}, \tau_q^{\mathrm{last}}$ and $\tau_q^{\mathrm{diff}}$. We can thus *locally* time instance normalize $t$ as $t \leftarrow (t - \tau_q^{\mathrm{first}})/\tau_q^{\mathrm{diff}}$, such that it is on the correct scale to apply the Trunk net of the pretrained $\mathrm{FIM}-\ell$. Let us again denote the output of the Trunk net as $\mathbf{t}^\theta = (\phi_1^\theta \circ \phi_0^\theta)(t)$.

(iii) $\mathrm{FIM}-\ell$ *embedding.* We apply the temporal encoder $\phi_0^\theta$ and the Branch net equivalent network $\phi_2^\theta \circ \psi_1^\theta$ of the underlying, *pretrained* $\mathrm{FIM}-\ell$ model to all sets $j \neq q$. Concretely, $\phi_0^\theta$ yields the individual observation embeddings

$$\mathbf{y}_i^\theta = \mathrm{Concat}(y_i, \phi_0^\theta(\tau_i)) \in \mathbb{R}^{513} \quad \text{for all } i = 1, \ldots, l,$$

and $\phi_2^\theta \circ \psi_1^\theta$ yields

$$\mathbf{u}_j^\theta = \phi_2^\theta(\psi_1^\theta(\mathbf{y}_{w_j^{\mathrm{prev}}+1}^\theta, \ldots, \mathbf{y}_{w_j^{\mathrm{prev}}+w_j}^\theta)) \in \mathbb{R}^{512} \quad \text{for all } j \neq q.$$

(iv) *Gap embedding estimation.* A sequence processing network $\psi_2^\varphi$, a transformer with 4 layers, 8 heads and (self) attention dimension 512, processes the sequence

$$(\mathbf{u}_1^\theta, \mathbf{s}_1^\varphi), \ldots, (\mathbf{u}_{q-1}^\theta, \mathbf{s}_{q-1}^\varphi), (\mathbf{u}_{q+1}^\theta, \mathbf{s}_{q+1}^\varphi), \ldots, (\mathbf{u}_K^\theta, \mathbf{s}_K^\varphi)$$

and returns an embedding for the $q$th set, that is

$$\mathbf{u}_q^\varphi = \psi_2^\varphi((\mathbf{u}_1^\theta, \mathbf{s}_1^\varphi), \ldots, (\mathbf{u}_{q-1}^\theta, \mathbf{s}_{q-1}^\varphi), (\mathbf{u}_{q+1}^\theta, \mathbf{s}_{q+1}^\varphi), \ldots, (\mathbf{u}_K^\theta, \mathbf{s}_K^\varphi)) \in \mathbb{R}^{512}.$$

(v) *FIM-$\ell$ time derivative projection.* We apply feed-forward neural networks $\phi_3^\theta$, $\phi_4^\theta$ and $\phi_5^\theta$ of the underlying, *pretrained* $\mathrm{FIM}-\ell$ model to the estimated embedding $\mathbf{u}_q^\varphi$ and the time $t$ in the imputation gap. We write

$$\hat{f}_q(t) = \phi_4^\theta(\mathbf{h}_q^\varphi(t)), \ \log \mathrm{Var}(\hat{f}_q)(t) = \phi_5^\theta(\mathbf{h}_q^\varphi(t)), \ \text{with} \ \mathbf{h}_q^\varphi(t) = \phi_3(\mathbf{t}^\theta, \mathbf{u}_q^\varphi).$$

Finally, we revert the *local* time instance normalization from *(ii)* inside the imputation gap, with the formulas presented in Appendix C.1.

**Model outputs**: In conclusion, our model returns a *time derivative* $\hat{f}_q(t)$, with uncertainty estimate $\log \mathrm{Var}(\hat{f}_q)(t)$, for a time $t \in \mathbb{R}_+$ in the evaluation gap.

Let us note in passing that the intermediate embeddings $\mathbf{u}_j^\theta$ for $j \neq q$ can be processed as usual with the underlying $\mathrm{FIM}-\ell$. Therefore, the *temporal missing pattern imputation model* can also return $\hat{f}_j(t), \log \mathrm{Var}(\hat{f}_j)(t)$ and initial conditions for sets $j \neq q$, via applications of the underlying $\mathrm{FIM}-\ell$. In Appendix D.6 we discuss how to combine these outputs from all $K$ sets to infer a continuous interpolating path $\hat{x}(t)$ defined on $[0, 1]$.

### D.3 TRAINING OBJECTIVE

The parameters $\varphi$ of FIM are trained in a supervised fashion, utilising the synthetic dataset for *temporal missing patterns* introduced in Appendix B.

We briefly recall the notation of Appendix B.4 for the data generated by the synthetic dataset. Let $f$ be the sampled function from one instance in the synthetic dataset, $\{f(t_i)\}_{i=1}^{L}$ its values on the fine grid, $x_0$ the initial value of the ODE solution, recorded on the fine grid $\{x(t_i)\}_{i=1}^{L}$, and $\{(y_i, \tau_i)\}_{i=1}^{l}$ the associated noisy time series.

The noisy time series is split into $K$ sequentially ordered sets, as outlined in Appendix D.2. Following the notation therein, we can identify the subset

$$\{t_i^q\}_{i=1}^{M} = \{t_i \mid \tau_q^{\text{first}} \leq t_i \leq \tau_q^{\text{last}}\} \subseteq \{t_i\}_{i=1}^{L}$$

of points on the fine grid contained in the imputation window.

Using $\{(y_i, \tau_i)\}_{i=1}^{l}$ as inputs for FIM, we denote its outputs evaluated on fine grid points contained in the imputation window by $\hat{f}(t_i^q)$ and $\log \text{Var}(\hat{f})(t_i^q)$, as in Appendix D.2.

The training objective of FIM is then to maximize the Gaussian log-likelihood of the *time derivative* $\log \mathcal{L}(f(t_i^q) \mid \hat{f}(t_i^q), \text{Var}(\hat{f})(t_i^q))$ at all $M$ points $t_i^q$ of the fine grid contained in the imputation gap.

Expressed as an equation, we train FIM to minimize

$$\mathcal{L} = \mathop{\mathbb{E}}_{f \sim p(f, \eta_f)} \left\{ \sum_{i=1}^{M} \frac{(f(t_i^q) - \hat{f}(t_i^q))^2}{2\text{Var}(\hat{f})(t_i^q)} + \frac{1}{2} \log(\text{Var}(\hat{f})(t_i^q)) \right\}.$$

### D.4 TRAINING PROCEDURE

The parameters $\varphi$ of FIM were optimized with AdamW (Loshchilov & Hutter, 2017), using a weight decay of $1e^{-3}$. We use a cosine annealing schedule as introduced by Loshchilov & Hutter (2016), where the learning rate decays from $1e^{-4}$ to $1e^{-7}$ over 400 epochs. Note that we do not deploy warm restarts. With a batch size of 1024, we trained FIM on a single A100 80GP GPU.

### D.5 ABLATION STUDIES: DATASET DESIGN

The dataset is generated using a GP model with periodic kernels. The grid spans from 0 to 1 with a resolution of 512 points. Each GP is created by combining kernels, and the parameters are sampled from uniform distributions. The lengthscale $\ell$ is sampled from $\mathcal{U}(0.75, 1)$, while the period is sampled from $\mathcal{U}(0.3, 0.5)$. The training dataset contains 500,000 samples and the test dataset contains 50,000 samples.

Our experimental datasets for the temporal missing pattern imputation (see Appendix H) contain no significant trends. Moreover, the temporal size of the imputation gap is $20\%$ for all samples in the time series Yet our synthetic generated training data includes both: trends and variable window sizes.

To assess the impact of the trend in our synthetic generated training data on the evaluation performance, we train two sets of models: one with a trend in the training data and the other without a trend. For both datasets, we train two models, varying the size of the imputation window. Specifically, the window size is sampled from either $\mathcal{U}(10, 30)$ or $\mathcal{U}(5, 50)$. In Table 8 we present the results of the four different FIM models trained on the four different dataset. One can see that the model trained on the dataset with smaller imputation window performs the best. Increasing the imputation window size leads to worse performance on the Motion Capture dataset for both case (PCA and no PCA). The performance of the FIM model also decreases when trend is included in the training dataset.

### D.6 CONTINUITY ALONG THE GAP

Similar to the windowing scheme of FIM$-\ell$ (see Appendix C.6.2), FIM processes a time series split into $K$ sets of sequentially ordered observations. Continuing the notation from Appendix D.2, we

Table 8: Ablation study on the size of the imputation window size and the occurrence of trends in the synthetic training data of $\texttt{FIM}$. We report RMSE in the imputation gap of the Motion Capture dataset.

| | Motion Capture | |
|---|---|---|
| Dataset | PCA | No PCA |
| No Trend - $\mathcal{U}(10, 30)$ | $3.27 \pm 1.12$ | $2.97 \pm 0.96$ |
| Trend - $\mathcal{U}(10, 30)$ | $3.41 \pm 1.17$ | $3.16 \pm 1.07$ |
| No Trend - $\mathcal{U}(5, 50)$ | $3.59 \pm 1.14$ | $3.52 \pm 1.18$ |
| Trend - $\mathcal{U}(5, 50)$ | $3.67 \pm 1.13$ | $3.59 \pm 1.19$ |

now detail our approach to connecting these outputs, such that the combined interpolating path $\hat{x}(t)$ is continuous in $[0, 1]$.

We can group sets containing observations, i.e. $j \neq q$ outside of the imputation gap, into two disjoint groups: sets $1 \leq j < q$ to the *left* of the imputation window and sets $q < j \leq K$ to the *right* of the imputation window. Model outputs for each group can be combined with the ideas from $\texttt{FIM}-\ell$, presented in Appendix C.6.2. Indeed, these sets are just windows that can be combined on some predefined overlap.

Let $\hat{x}^{\text{left}}$ and $\hat{x}^{\text{right}}$ denote the inferred trajectories to the left and right of the imputation gap $[\tau_q^{\text{first}}, \tau_q^{\text{last}}]$, which are designed to be *continuous* by the efforts described in Appendix C.6.2. To connect the two trajectories to a continuous path inside the imputation gap, we first extend both of them individually by integrating the time derivative $\hat{f}_q$ inferred from $\texttt{FIM}$ with initial values $\hat{x}^{\text{left}}$ and $\hat{x}^{\text{right}}$ respectively.

Expressed in equations, we define for $t \in [\tau_q^{\text{first}}, \tau_q^{\text{last}}]$ the following extensions:

$$\hat{x}^{\text{left}}(t) = \hat{x}^{\text{left}}(\tau_q^{\text{first}}) + \int_{\tau_q^{\text{first}}}^{t} \hat{f}_q(s)ds$$

$$\hat{x}^{\text{right}}(t) = \hat{x}^{\text{right}}(\tau_q^{\text{last}}) - \int_{t}^{\tau_q^{\text{last}}} \hat{f}_q(s)ds$$

These extensions are then combined via weighted temporal interpolation, where we define the combined solution at a time $t \in [0, 1]$ by

$$\hat{x}(t) = \begin{cases} \hat{x}^{\text{left}}(t) & \text{for } t < \tau_q^{\text{first}} \\ \frac{\tau_q^{\text{last}} - t}{\tau_q^{\text{last}} - \tau_q^{\text{first}}} \hat{x}^{\text{left}}(t) + \frac{t - \tau_q^{\text{first}}}{\tau_q^{\text{last}} - \tau_q^{\text{first}}} \hat{x}^{\text{right}}(t) & \text{for } t \in [\tau_q^{\text{first}}, \tau_q^{\text{last}}] \\ \hat{x}^{\text{right}}(t) & \text{for } t > \tau_q^{\text{last}} . \end{cases} \quad (20)$$

Note that $\hat{x}$ is *continuous by design* and its corresponding combined time derivative $\hat{f}$ can be found (*a.e.*) by taking the derivative of equation 20.

## D.7 PROCESSING TIME SERIES OF ANY DIMENSIONALITY

$\texttt{FIM}$ contains a pretrained $\texttt{FIM}-\ell$ model. In fact, $\texttt{FIM}$ just extends that (frozen) model by two networks: a linear layer $\phi_8$ to embed local scale statistics, and a sequence processing network $\psi_2$, processing a sequence of $\texttt{FIM}-\ell$ embeddings of sets of (consecutive) observations. Therefore, it also only returns 1D time derivatives. To impute arbitrary dimensional time series data with $\texttt{FIM}$, we process each component of a time series separately. The same channel independent strategy is used for $\texttt{FIM}-\ell$. In Appendix C.6.1 we discuss limitations of this approach, which are also valid for the temporal missing pattern imputation model $\texttt{FIM}$.

## E EVALUATION METRICS

Depending on the experiment and the baselines, we use one or more of the following evaluation metrics for multi-dimensional time series.

Following the notation of Du et al. (2024), consider a multi-dimensional target time series $x_1, \ldots, x_L$, with $x_i \in \mathbb{R}^D$, and corresponding estimated time series $\hat{x}_1, \ldots, \hat{x}_L$. Some baselines only compute the metrics below on certain elements and components of each time series. Let $m_1, \ldots, m_L$ be an associated sequence of masks $m_i \in \{0, 1\}^D$, indicating which elements and components are part of the metric calculation. Finally, denote by $x_i^d$, $\hat{x}_i^d$ and $m_i^d$ the $d$th component of each vector.

We compute per trajectory the Mean Absolute Error (MAE), (Root) Mean Squared Error ((R)MSE), Mean Relative Error (MRE) and the $R^2$ score , defined as follows:

$$\text{MAE}(x, \hat{x}, m) = \frac{\sum_{i=1}^{L} \sum_{d=1}^{D} \left| x_i^d - \hat{x}_i^d \right| \cdot m_i^d}{\sum_{i=1}^{L} \sum_{d=1}^{D} m_i^d}$$

$$\text{MSE}(x, \hat{x}, m) = \frac{\sum_{i=1}^{L} \sum_{d=1}^{D} (x_i^d - \hat{x}_i^d)^2 \cdot m_i^d}{\sum_{i=1}^{L} \sum_{d=1}^{D} m_i^d}$$

$$\text{RMSE}(x, \hat{x}, m) = \sqrt{\text{MSE}(x, \hat{x}, m)}$$

$$\text{MRE}(x, \hat{x}, m) = \frac{\sum_{i=1}^{L} \sum_{d=1}^{D} \left| x_i^d - \hat{x}_i^d \right| \cdot m_i^d}{\sum_{i=1}^{L} \sum_{d=1}^{D} \left| x_i^d \right| \cdot m_i^d}$$

$$R^2(x, \hat{x}) = \frac{1}{D} \sum_{d=1}^{D} \left[ 1 - \frac{\sum_{i=1}^{L} (x_i^d - \hat{x}_i^d)^2}{\sum_{i=1}^{L} (x_i^d - \bar{x}^d)^2} \right]$$

where $\bar{x}^d = \frac{1}{L} \sum_{i=1}^{L} x_i^d$. Note that we only require the $R^2$ score for the comparison on ODEBench, where no targets are masked, so we do not add the mask in the formula above.

For the imputation task of *point-wise missing patterns*, we usually compute the metrics only at the missing points, following the methodology of our baselines, e.g. (Du et al., 2024). One exception is the ODEBench evaluation, where the baseline values are computed against the complete target trajectory, including observed and missing time points. We adapt our computation to the baseline methodology for a valid comparison.

For the imputation task of *temporal missing patterns*, metrics are only computed *in the imputation gap*, in accordance with our baselines.

Usually, we report mean and standard deviation of the metrics calculated over all trajectories in a dataset. The only exception is the $R^2$-Accuracy, defined by d'Ascoli et al. (2024) as the percentage of time series with $R^2$ score larger than 0.9. Note that Du et al. (2024) and Fang et al. (2024) only report metrics averaged over all missing values over all time series, not a standard deviation over time series.

## F  PHASE PORTRAIT RECONSTRUCTION: ADDITIONAL RESULTS

### F.1  VAN DER POLL OSCILLATOR

#### F.1.1  DATA DESCRIPTION AND PRE-PROCESSING

One particular (second-order) differential equation from the ODEBench dataset is the Van der Pol oscillator:

$$dx = v$$
$$dv = \mu(1 - x^2)v - x$$

We simulate the system in the time interval $[0, 10]$ for a set of initial positions $x(0)$, initial velocities $v(0)$ and parameters $\mu$:

$$x(0) \in \{-5, -3, -1, 1, 3, 5\}$$
$$v(0) \in \{-4.5, 4.5\}$$
$$\mu \in \{0.1, 0.5, 1.5\}$$

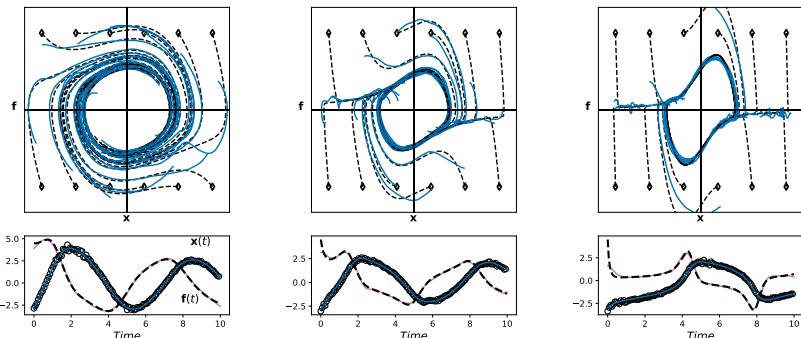

Figure 4: Phase portraits of Van der Pol systems with parameters $\mu = 0.1, 0.5, 1.5$. 128 noisy observation (black circles) get interpolated by $\mathtt{FIM}-\ell(w.n. = 4)$ (blue line). The inferred time derivative matches (magenta line) matches the ground truth vector field at the solution path (black dashed lines).

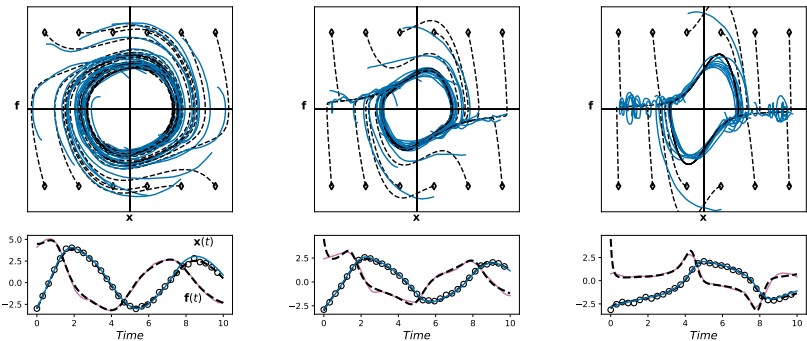

Figure 5: Phase portraits of Van der Pol systems with parameters $\mu = 0.1, 0.5, 1.5$. 32 noisy observation (black circles) get interpolated by$\mathtt{FIM}-\ell(w.n. = 4)$ (blue line). The inferred time derivative matches (magenta line) matches the ground truth vector field at the solution path (black dashed lines).

We generate $1D$ time series by observing *the position $x(t)$ only* at a regular grid of either 128 or 32 points in the interval $[0, 10]$ and corrupt them with multiplicative noise of level $\gamma = 0.05$ inspired by the ODEBench setup (see Appendix F.3).

### F.1.2 Modelling and Results

We apply $\mathtt{FIM}-\ell(w.n. = 4)$ and visualise the inferred path and the inferred time derivative by means of phase portraits. Figure 4 contains the visualisation for 128 observations and Figure 5 for 32 observations. In each figure, the parameter $\mu$ ranges from $\mu = 0.1$ in the left, to $\mu = 0.5$ in the middle and $\mu = 1.5$ in the right subplot.

$\mathtt{FIM}-\ell$ interpolates the observed path and the inferred time derivative $f(t)$ matches the ground truth vector field $v(t)$ at the path closely. The accuracy of the (implied) estimation $f(0)$ of the initial velocity $v(0)$ is not perfect and degrades with increasing $\mu$. In other words, based purely on the (limited) observations, the model can not infer the sharp change in velocity at the beginning of a trajectory. The bottom, $1D$ subplots reveal that $\mathtt{FIM}-\ell$ adjusts to a more accurate estimation quickly.

Even with only 32 observations (considerably less than the roughy 256 observations in the ODEBench), $\mathtt{FIM}-\ell$ recovers the ground truth time derivative quite well, showcasing that $\mathtt{FIM}-\ell$ does not require many observations to approximate simpler regions of the dynamics.

## F.2 RÖSSLER ATTRACTOR

### F.2.1 DATA DESCRIPTION AND PRE-PROCESSING

In Figure 2 we consider the Rössler attractor differential equation from the ODEBench in its chaotic variation:

$$dx = -5(y + z)$$
$$dy = 5(0.2y + x)$$
$$dz = 5(0.2 + z(-5.7 + x))$$

Also following the ODEBench, we simulate the system in the time interval $[0, 10]$ with initial value $(2.3, 1.1, 0.8)$. From $4096$ observations on a regular time grid in $[0, 10]$, we subsample a irregular observation grid by dropping each observation (independently) with probability $0.5$. The subsampled data is then corrupted with multiplicative noise of level $0.05$. This corruption scheme follows one corruption scheme of ODEBench, which is described in see Appendix F.3.

### F.2.2 MODELLING AND RESULTS

We apply $\texttt{FIM-}\ell(w.n. = 32)$ to recover the trajectory from the corrupted data. Our model also returns the time derivative along the recovered trajectory, which is the velocity of the particle in the Rössler attractor system. The centered plot in Figure 2 shows the ground-truth and model inference by means of a phase portrait. It contains position and velocity for the first component, i.e. $x$ and $dx$, along the trajectory.

Our model can also approximate the acceleration of the particle. First, we discretize the *inferred velocity* via a regular grid of size $8192$ in the time interval $[0, 10]$. To this discretized function, we apply $\texttt{FIM-}\ell(w.n. = 64)$ and recover an approximation of the acceleration of the particle. The rightmost plot in Figure 2 shows the velocity and acceleration of the first component along the observed trajectory, again by means of a phase portrait.

## F.3 ODE BENCH

### F.3.1 DATA DESCRIPTION AND PRE-PROCESSING

The ODEBench dataset (d'Ascoli et al., 2024) contains the solution of $63$ ODEs with $2$ initial conditions each. The solutions are available on a regular grid of length $512$ in the time interval $[0, 10]$.

We follow the pre-processing of d'Ascoli et al. (2024), who corrupt both the observations values and the observation grid. Let us recall their corruption scheme here for completeness.

The *observation values* are corrupted by multiplicative noise, yielding noisy observations $y_i$ of the ground truth solution $x_i = x(t_i)$ at some time $t_i$ via $y_i = (1 + \epsilon)x_i$, where $\epsilon \sim \mathcal{N}(0, \gamma)$. The *observation grid* is corrupted by dropping each observation (independently) with probability $\rho$.

All combinations of $\gamma \in \{0, 0.05\}$ and $\rho \in \{0, 0.5\}$ are considered.

### F.3.2 BASELINES

We are interested in two tasks on this dataset:

(i) reconstruction of the underlying *ODE solution* (which is the task considered by d'Ascoli et al. (2024)) and

(ii) inference of the vector field on the solution path, i.e. the *time derivative*.

Note that in the case of $\rho = 0.5$, this setup is very similar to the *missing point imputation* task considered in other works.

While d'Ascoli et al. (2024) consider the first task (and we compare against their results below), they do not report any results related to the second task. We therefore reevaluate ODEFormer on the ODEBench, to gain access to the time derivative information needed for our comparison. We use

the implementation and weights provided by the authors[9], apply it to 10 samplings of the corruption schemes (as we do for our own model) and average the results.

On the solution reconstruction task, our re-evaluation yields different results than reported by the authors (see e.g. Table 9). We report both results, if they are available, and name the results from our re-evaluation *ODEFormer Re-ev.*.

Results of other well performing models on the ODEBench have been extracted from Figure 4 of d'Ascoli et al. (2024).

### F.3.3 MODELLING AND RESULTS

We apply `FIM-`$\ell$ to all equations of ODEBench and compare our results to the best performing models from d'Ascoli et al. (2024), as well as our re-evaluation of ODEFormer outlined in Appendix F.3.2.

The authors of ODEBench compare several methods based on the reconstruction of the ground-truth ODE solution at all 512 available points of each trajectory. This comparison is based on the $R^2$ accuracy - the percentage of predictions of which the $R^2$ score exceeds 0.9. We compare our zero-shot model `FIM-`$\ell$ with different numbers of windows to the best performing models from d'Ascoli et al. (2024). We report our result wrt. the $R^2$-accuracy (see Table 9), the RMSE (see Table 10) and the MAE (see Table 11. For our models, and if available for ODEFormer, we provide the metrics for equations grouped by their dimensionality, and average our results over 10 samplings of the corruption schemes detailed above.

Averaged over all equations, `FIM-`$\ell$ improves the baselines on all corruption schemes, in particular with larger number of windows. The optimal number of windows depends on the difficulty of the underlying solution. More complex dynamics require a higher local resulution, thus more windows, than easier dynamics, which benefit from the smooting effect of a low number of windows. Generally speaking, equations of the same dimensionality in the ODEBench are of similar complexity. We therefore denote by `FIM-`$\ell$(Weighted Sum) the average of the best performing number of windows per equation, weighted by the number of equations of that dimension.

Noticeably, the performance drops for equations of dimension 3. Figure 6 shows the path reconstruction for one initial value problem of each equation. Trajectories of $3D$ systems included in ODEBench are inherently more complex than for all other dimensions. Still, $FIM-\ell(w.n.=8)$, used for Figure 6, infers suitable initial conditions and vector fields that approximate the ground-truth solution well.

Let us now consider the accuracy of the inferred time derivative along the solution path. We provide the results for our model and for our re-evaluation of ODEFormer on ODEBench. Similar to the original ODEBench task, we compute the metrics against the time derivative along the solution path of the ground-truth solution. To extract an estimation of the this time derivative from ODEFormer, we can evaluate the inferred equation along two possible paths:

  (i) the *ground-truth* solution of the *ground-truth equation*
 (ii) the *inferred trajectory* from ODEFormer

While we report both evaluation approaches, we find that the second performs consistently better than the first. We mark the results with *ODEFormer (g.t. traj)* and *ODEFormer (inf. traj.)* respectively.

We report the results wrt. the RMSE (see Table 12) and the MAE (see Table 13). Our model performs better than ODEFormer on both evaluation approaches. The inferred time derivative is much closer to the ground-truth vector field along the solution path, especially when we use more than 2 windows.

---

[9]https://github.com/sdascoli/odeformer

Table 9: ODEBench $R^2$ accuracy of $\texttt{FIM-}\ell$ for different numbers of window, split by dimensions. The standard deviation is calculated across 10 samplings of the corruption schemes.

| Dim. | Model | $\rho = 0$ $\gamma = 0$ | $\rho = 0$ $\gamma = 0.05$ | $\rho = 0.5$ $\gamma = 0$ | $\rho = 0.5$ $\gamma = 0.05$ |
|---|---|---|---|---|---|
| All | ODEFormer | 71.2 | 52.2 | 69.9 | 60.3 |
| | ODEFormer-opt | 75.9 | 55.7 | 74.7 | 66.6 |
| | ODEFormer Re-ev. | $64.3 \pm 0.0$ | $64.7 \pm 2.1$ | $62.9 \pm 1.9$ | $61.6 \pm 3.0$ |
| | PySR | 82.3 | 63.2 | 77.0 | 38.2 |
| | $\texttt{FIM-}\ell(w.n. = 2)$ | $69.0 \pm 0.0$ | $67.0 \pm 0.8$ | $77.5 \pm 0.9$ | $76.1 \pm 1.3$ |
| | $\texttt{FIM-}\ell(w.n. = 4)$ | $86.5 \pm 0.0$ | $84.0 \pm 0.7$ | $86.6 \pm 0.5$ | $83.7 \pm 0.7$ |
| | $\texttt{FIM-}\ell(w.n. = 8)$ | $91.3 \pm 0.0$ | $88.3 \pm 0.5$ | $91.6 \pm 1.0$ | $87.5 \pm 1.0$ |
| | $\texttt{FIM-}\ell(w.n. = 16)$ | $\mathbf{100.0 \pm 0.0}$ | $96.2 \pm 0.5$ | $97.2 \pm 1.2$ | $91.6 \pm 1.9$ |
| | $\texttt{FIM-}\ell$ (Weighted Sum) | $\mathbf{100.0 \pm 0.0}$ | $\mathbf{96.9 \pm 0.4}$ | $97.2 \pm 1.2$ | $93.1 \pm 1.2$ |
| 1 | ODEFormer Re-ev. | $89.1 \pm 0.0$ | $87.8 \pm 2.0$ | $87.5 \pm 2.5$ | $84.5 \pm 3.8$ |
| | $\texttt{FIM-}\ell(w.n. = 2)$ | $97.8 \pm 0.0$ | $93.5 \pm 0.0$ | $98.7 \pm 1.1$ | $93.9 \pm 1.7$ |
| | $\texttt{FIM-}\ell(w.n. = 4)$ | $\mathbf{100.0 \pm 0.0}$ | $\mathbf{95.4 \pm 0.7}$ | $99.8 \pm 0.7$ | $\mathbf{95.2 \pm 0.9}$ |
| | $\texttt{FIM-}\ell(w.n. = 8)$ | $\mathbf{100.0 \pm 0.0}$ | $\mathbf{95.4 \pm 0.7}$ | $99.8 \pm 0.7$ | $93.5 \pm 1.0$ |
| | $\texttt{FIM-}\ell(w.n. = 16)$ | $\mathbf{100.0 \pm 0.0}$ | $93.7 \pm 0.7$ | $\mathbf{100.0 \pm 0.0}$ | $91.7 \pm 1.4$ |
| 2 | ODEFormer Re-ev. | $62.5 \pm 0.0$ | $63.8 \pm 3.5$ | $61.4 \pm 3.6$ | $60.7 \pm 4.6$ |
| | $\texttt{FIM-}\ell(w.n. = 2)$ | $62.5 \pm 0.0$ | $61.4 \pm 1.2$ | $80.0 \pm 2.0$ | $80.5 \pm 2.7$ |
| | $\texttt{FIM-}\ell(w.n. = 4)$ | $94.6 \pm 0.0$ | $93.8 \pm 1.3$ | $95.0 \pm 1.1$ | $93.6 \pm 0.9$ |
| | $\texttt{FIM-}\ell(w.n. = 8)$ | $\mathbf{100.0 \pm 0.0}$ | $\mathbf{98.6 \pm 0.8}$ | $99.3 \pm 1.2$ | $97.5 \pm 0.9$ |
| | $\texttt{FIM-}\ell(w.n. = 16)$ | $\mathbf{100.0 \pm 0.0}$ | $98.4 \pm 0.6$ | $99.5 \pm 0.9$ | $97.0 \pm 1.5$ |
| 3 | ODEFormer Re-ev. | $10.0 \pm 0.0$ | $13.1 \pm 3.7$ | $9.4 \pm 3.2$ | $12.5 \pm 4.7$ |
| | $\texttt{FIM-}\ell(w.n. = 2)$ | $15.0 \pm 0.0$ | $15.0 \pm 4.1$ | $17.5 \pm 2.6$ | $18.0 \pm 2.6$ |
| | $\texttt{FIM-}\ell(w.n. = 4)$ | $30.0 \pm 0.0$ | $27.0 \pm 2.6$ | $30.0 \pm 0.0$ | $26.5 \pm 2.4$ |
| | $\texttt{FIM-}\ell(w.n. = 8)$ | $45.0 \pm 0.0$ | $40.5 \pm 1.6$ | $49.5 \pm 6.0$ | $43.0 \pm 6.3$ |
| | $\texttt{FIM-}\ell(w.n. = 16)$ | $\mathbf{100.0 \pm 0.0}$ | $\mathbf{95.0 \pm 0.0}$ | $84.0 \pm 6.1$ | $74.5 \pm 6.9$ |
| 4 | ODEFormer Re-ev. | $75.0 \pm 0.0$ | $68.8 \pm 11.6$ | $68.8 \pm 11.6$ | $56.3 \pm 17.7$ |
| | $\texttt{FIM-}\ell(w.n. = 2)$ | $\mathbf{100.0 \pm 0.0}$ | $\mathbf{100.0 \pm 0.0}$ | $\mathbf{100.0 \pm 0.0}$ | $\mathbf{100.0 \pm 0.0}$ |
| | $\texttt{FIM-}\ell(w.n. = 4)$ | $\mathbf{100.0 \pm 0.0}$ | $\mathbf{100.0 \pm 0.0}$ | $\mathbf{100.0 \pm 0.0}$ | $\mathbf{100.0 \pm 0.0}$ |
| | $\texttt{FIM-}\ell(w.n. = 8)$ | $\mathbf{100.0 \pm 0.0}$ | $\mathbf{100.0 \pm 0.0}$ | $\mathbf{100.0 \pm 0.0}$ | $\mathbf{100.0 \pm 0.0}$ |
| | $\texttt{FIM-}\ell(w.n. = 16)$ | $\mathbf{100.0 \pm 0.0}$ | $\mathbf{100.0 \pm 0.0}$ | $\mathbf{100.0 \pm 0.0}$ | $\mathbf{100.0 \pm 0.0}$ |

Table 10: ODEBench RMSE of $\texttt{FIM-}\ell$ for different numbers of windows, split by dimensions. The standard deviation is calculated across 10 samplings of the corruption schemes.

| Dim. | Model | $\rho = 0$ $\gamma = 0$ | $\rho = 0$ $\gamma = 0.05$ | $\rho = 0.5$ $\gamma = 0$ | $\rho = 0.5$ $\gamma = 0.05$ |
|---|---|---|---|---|---|
| All | ODEFormer Re-ev. | $1.45440 \pm 0.0$ | $1.49561 \pm 0.28744$ | $1.61460 \pm 0.07631$ | $1.55996 \pm 0.06814$ |
| | $\texttt{FIM-}\ell(w.n. = 2)$ | $1.49276 \pm 0.0$ | $1.53191 \pm 0.01379$ | $1.11169 \pm 0.02296$ | $1.17046 \pm 0.02488$ |
| | $\texttt{FIM-}\ell(w.n. = 4)$ | $0.89672 \pm 0.0$ | $0.97397 \pm 0.00758$ | $0.7747 \pm 0.01644$ | $0.86858 \pm 0.01461$ |
| | $\texttt{FIM-}\ell(w.n. = 8)$ | $0.47344 \pm 0.0$ | $0.57404 \pm 0.0068$ | $0.4351 \pm 0.00907$ | $0.58163 \pm 0.01079$ |
| | $\texttt{FIM-}\ell(w.n. = 16)$ | $0.14572 \pm 0.0$ | $0.30887 \pm 0.0059$ | $0.28492 \pm 0.020$ | $0.52077 \pm 0.01884$ |
| | $\texttt{FIM-}\ell$ (Weighted Sum) | $0.14572 \pm 0.0$ | $\mathbf{0.25858 \pm 0.0059}$ | $0.28492 \pm 0.020$ | $0.41192 \pm 0.0205$ |
| 1 | ODEFormer Re-ev. | $0.50954 \pm 0.0$ | $0.52900 \pm 0.13294$ | $0.58829 \pm 0.04373$ | $0.66961 \pm 0.09289$ |
| | $\texttt{FIM-}\ell(w.n. = 2)$ | $0.47952 \pm 0.0$ | $0.54023 \pm 0.01949$ | $0.12225 \pm 0.00566$ | $\mathbf{0.26812 \pm 0.01323}$ |
| | $\texttt{FIM-}\ell(w.n. = 4)$ | $0.03913 \pm 0.0$ | $\mathbf{0.22905 \pm 0.01549}$ | $0.04358 \pm 0.00269$ | $0.27359 \pm 0.01662$ |
| | $\texttt{FIM-}\ell(w.n. = 8)$ | $0.02668 \pm 0.0$ | $0.27605 \pm 0.01472$ | $0.0215 \pm 0.00217$ | $0.36447 \pm 0.01162$ |
| | $\texttt{FIM-}\ell(w.n. = 16)$ | $0.01135 \pm 0.0$ | $0.36661 \pm 0.01394$ | $0.01218 \pm 0.00091$ | $0.54251 \pm 0.01067$ |
| 2 | ODEFormer Re-ev. | $0.58488 \pm 0.0$ | $0.55307 \pm 0.05853$ | $0.87805 \pm 0.12628$ | $0.67336 \pm 0.17509$ |
| | $\texttt{FIM-}\ell(w.n. = 2)$ | $0.48285 \pm 0.0$ | $0.4882 \pm 0.003$ | $0.33043 \pm 0.019$ | $0.3402 \pm 0.02128$ |
| | $\texttt{FIM-}\ell(w.n. = 4)$ | $0.14509 \pm 0.0$ | $0.15864 \pm 0.00224$ | $0.13071 \pm 0.01099$ | $0.15387 \pm 0.01138$ |
| | $\texttt{FIM-}\ell(w.n. = 8)$ | $0.04052 \pm 0.0$ | $0.06608 \pm 0.00199$ | $0.04881 \pm 0.00897$ | $0.09019 \pm 0.00698$ |
| | $\texttt{FIM-}\ell(w.n. = 16)$ | $0.00857 \pm 0.0$ | $\mathbf{0.06297 \pm 0.00327}$ | $0.03812 \pm 0.01606$ | $0.10932 \pm 0.00977$ |
| 3 | ODEFormer Re-ev. | $6.34598 \pm 0.0$ | $6.65072 \pm 1.60730$ | $6.35373 \pm 0.32882$ | $6.39117 \pm 0.38931$ |
| | $\texttt{FIM-}\ell(w.n. = 2)$ | $6.94523 \pm 0.0$ | $7.03731 \pm 0.07836$ | $5.79548 \pm 0.14598$ | $5.80209 \pm 0.14687$ |
| | $\texttt{FIM-}\ell(w.n. = 4)$ | $5.15268 \pm 0.0$ | $5.16363 \pm 0.02147$ | $4.41386 \pm 0.0952$ | $4.4104 \pm 0.09807$ |
| | $\texttt{FIM-}\ell(w.n. = 8)$ | $2.8077 \pm 0.0$ | $2.79512 \pm 0.01428$ | $2.5547 \pm 0.05278$ | $2.57153 \pm 0.06213$ |
| | $\texttt{FIM-}\ell(w.n. = 16)$ | $0.86784 \pm 0.0$ | $0.92454 \pm 0.0055$ | $1.65999 \pm 0.13815$ | $1.72432 \pm 0.12407$ |
| 4 | ODEFormer Re-ev. | $0.03579 \pm 0.0$ | $0.03170 \pm 0.00393$ | $0.03305 \pm 0.00303$ | $0.05530 \pm 0.03747$ |
| | $\texttt{FIM-}\ell(w.n. = 2)$ | $0.02153 \pm 0.0$ | $0.0213 \pm 0.00116$ | $0.00883 \pm 0.00169$ | $0.01288 \pm 0.00156$ |
| | $\texttt{FIM-}\ell(w.n. = 4)$ | $0.00213 \pm 0.0$ | $\mathbf{0.00696 \pm 0.00064}$ | $0.00247 \pm 0.0009$ | $\mathbf{0.00786 \pm 0.00079}$ |
| | $\texttt{FIM-}\ell(w.n. = 8)$ | $0.00079 \pm 0.0$ | $0.00715 \pm 0.00041$ | $0.00147 \pm 0.00065$ | $0.00961 \pm 0.00046$ |
| | $\texttt{FIM-}\ell(w.n. = 16)$ | $0.00046 \pm 0.0$ | $0.00917 \pm 0.00033$ | $0.00121 \pm 0.00063$ | $0.01318 \pm 0.0007$ |

Table 11: ODEBench MAE of FIM−ℓ for different numbers of windows, split by dimensions. The standard deviation is calculated across 10 samplings of the corruption schemes.

| Dim. | Model | $\rho = 0$ $\gamma = 0$ | $\rho = 0$ $\gamma = 0.05$ | $\rho = 0.5$ $\gamma = 0$ | $\rho = 0.5$ $\gamma = 0.05$ |
|---|---|---|---|---|---|
| All | ODEFormer Re-ev. | $1.06575 \pm 0.0$ | $1.08829 \pm 0.15297$ | $1.17828 \pm 0.05353$ | $1.15633 \pm 0.04795$ |
| | FIM−ℓ($w.n. = 2$) | $1.13792 \pm 0.0$ | $1.16799 \pm 0.00904$ | $0.83951 \pm 0.01965$ | $0.88588 \pm 0.01983$ |
| | FIM−ℓ($w.n. = 4$) | $0.65753 \pm 0.0$ | $0.71483 \pm 0.00482$ | $0.57207 \pm 0.01368$ | $0.64132 \pm 0.01129$ |
| | FIM−ℓ($w.n. = 8$) | $0.33755 \pm 0.0$ | $0.41024 \pm 0.00456$ | $0.30721 \pm 0.00521$ | $0.41284 \pm 0.00723$ |
| | FIM−ℓ($w.n. = 16$) | $0.09628 \pm 0.0$ | $0.21383 \pm 0.00345$ | $0.17416 \pm 0.01029$ | $0.34229 \pm 0.01006$ |
| 1 | ODEFormer Re-ev. | $0.32923 \pm 0.0$ | $0.34722 \pm 0.08502$ | $0.39660 \pm 0.04165$ | $0.47208 \pm 0.04834$ |
| | FIM−ℓ($w.n. = 2$) | $0.38708 \pm 0.0$ | $0.43727 \pm 0.01308$ | $0.09335 \pm 0.00506$ | $0.20900 \pm 0.01057$ |
| | FIM−ℓ($w.n. = 4$) | $0.02499 \pm 0.0$ | $\mathbf{0.16663 \pm 0.01033}$ | $0.03213 \pm 0.00204$ | $\mathbf{0.20268 \pm 0.01535}$ |
| | FIM−ℓ($w.n. = 8$) | $0.02040 \pm 0.0$ | $0.20011 \pm 0.01021$ | $0.01403 \pm 0.00093$ | $0.26112 \pm 0.01003$ |
| | FIM−ℓ($w.n. = 16$) | $0.00801 \pm 0.0$ | $0.26645 \pm 0.00826$ | $0.00692 \pm 0.00025$ | $0.38727 \pm 0.00971$ |
| 2 | ODEFormer Re-ev. | $0.40630 \pm 0.0$ | $0.39648 \pm 0.03423$ | $0.60159 \pm 0.07357$ | $0.46471 \pm 0.09784$ |
| | FIM−ℓ($w.n. = 2$) | $0.33846 \pm 0.0$ | $0.34071 \pm 0.00229$ | $0.22632 \pm 0.01166$ | $0.23367 \pm 0.01281$ |
| | FIM−ℓ($w.n. = 4$) | $0.08653 \pm 0.0$ | $0.09716 \pm 0.00135$ | $0.07813 \pm 0.00575$ | $0.09487 \pm 0.00607$ |
| | FIM−ℓ($w.n. = 8$) | $0.02190 \pm 0.0$ | $0.03982 \pm 0.00074$ | $0.02557 \pm 0.00285$ | $0.05379 \pm 0.00249$ |
| | FIM−ℓ($w.n. = 16$) | $0.00485 \pm 0.0$ | $\mathbf{0.03821 \pm 0.00141}$ | $0.01548 \pm 0.00371$ | $0.06106 \pm 0.00213$ |
| 3 | ODEFormer Re-ev. | $4.81377 \pm 0.0$ | $4.94279 \pm 0.82914$ | $4.82151 \pm 0.18451$ | $4.88917 \pm 0.31551$ |
| | FIM−ℓ($w.n. = 2$) | $5.32823 \pm 0.0$ | $5.39580 \pm 0.05009$ | $4.43940 \pm 0.12939$ | $4.44416 \pm 0.12705$ |
| | FIM−ℓ($w.n. = 4$) | $3.84243 \pm 0.0$ | $3.84711 \pm 0.01762$ | $3.31108 \pm 0.08397$ | $3.30732 \pm 0.08331$ |
| | FIM−ℓ($w.n. = 8$) | $2.01825 \pm 0.0$ | $2.01172 \pm 0.01012$ | $1.83143 \pm 0.03501$ | $1.84830 \pm 0.04210$ |
| | FIM−ℓ($w.n. = 16$) | $0.57452 \pm 0.0$ | $0.62605 \pm 0.00421$ | $1.03782 \pm 0.07042$ | $1.09290 \pm 0.06818$ |
| 4 | ODEFormer Re-ev. | $0.02798 \pm 0.0$ | $0.02361 \pm 0.00258$ | $0.02508 \pm 0.00251$ | $0.04356 \pm 0.03045$ |
| | FIM−ℓ($w.n. = 2$) | $0.01355 \pm 0.0$ | $0.01418 \pm 0.00045$ | $0.00539 \pm 0.00113$ | $0.00960 \pm 0.00129$ |
| | FIM−ℓ($w.n. = 4$) | $0.00132 \pm 0.0$ | $0.00531 \pm 0.00054$ | $0.00131 \pm 0.00037$ | $\mathbf{0.00588 \pm 0.00061}$ |
| | FIM−ℓ($w.n. = 8$) | $0.00046 \pm 0.0$ | $0.00511 \pm 0.00030$ | $0.00066 \pm 0.00020$ | $0.00685 \pm 0.00037$ |
| | FIM−ℓ($w.n. = 16$) | $0.00027 \pm 0.0$ | $0.00650 \pm 0.00017$ | $0.00053 \pm 0.00017$ | $0.00933 \pm 0.00041$ |

Table 12: ODEBench RMSE of the inferred time derivative of FIM−ℓ for different number of window, split by dimensions. The standard deviation is calculated across 10 samplings of the corruption schemes.

| Dim. | Model | $\rho = 0$ $\gamma = 0$ | $\rho = 0$ $\gamma = 0.05$ | $\rho = 0.5$ $\gamma = 0$ | $\rho = 0.5$ $\gamma = 0.05$ |
|---|---|---|---|---|---|
| All | ODEFormer (g.t. traj.) | $30.49650 \pm 0.0$ | $71.30478 \pm 131.29482$ | $69.50647 \pm 50.34435$ | $93.24899 \pm 105.60553$ |
| | ODEFormer (inf. traj.) | $16.50882 \pm 0.0$ | $13.10672 \pm 2.30755$ | $19.82354 \pm 2.94507$ | $17.77896 \pm 4.61146$ |
| | FIM−ℓ($w.n. = 2$) | $10.09833 \pm 0.0$ | $10.07751 \pm 0.01023$ | $9.80062 \pm 0.00628$ | $9.85593 \pm 0.00853$ |
| | FIM−ℓ($w.n. = 4$) | $8.92481 \pm 0.0$ | $9.01273 \pm 0.01408$ | $8.74446 \pm 0.01624$ | $8.89065 \pm 0.01899$ |
| | FIM−ℓ($w.n. = 8$) | $6.84601 \pm 0.0$ | $7.177 \pm 0.03336$ | $6.91584 \pm 0.03868$ | $7.46845 \pm 0.04305$ |
| | FIM−ℓ($w.n. = 16$) | $3.49679 \pm 0.0$ | $4.79098 \pm 0.06319$ | $4.46558 \pm 0.13783$ | $6.48983 \pm 0.17314$ |
| | FIM−ℓ (Weighted Sum) | $3.49679 \pm 0.0$ | $\mathbf{3.78833 \pm 0.0204}$ | $\mathbf{4.46558 \pm 0.1321}$ | $\mathbf{4.78349 \pm 0.136708}$ |
| 1 | ODEFormer (g.t. traj.) | $8.58175 \pm 0.0$ | $6.98587 \pm 5.39939$ | $23.63005 \pm 18.09459$ | $20.6331 \pm 28.93572$ |
| | ODEFormer (inf. traj.) | $6.21711 \pm 0.0$ | $4.46383 \pm 2.4357$ | $11.4018 \pm 5.99679$ | $12.3632 \pm 9.90603$ |
| | FIM−ℓ($w.n. = 2$) | $0.57402 \pm 0.0$ | $0.53237 \pm 0.02785$ | $0.15242 \pm 0.00725$ | $\mathbf{0.29413 \pm 0.02427}$ |
| | FIM−ℓ($w.n. = 4$) | $0.08764 \pm 0.0$ | $\mathbf{0.34632 \pm 0.03219}$ | $0.08866 \pm 0.00263$ | $0.45892 \pm 0.04078$ |
| | FIM−ℓ($w.n. = 8$) | $0.06023 \pm 0.0$ | $0.87099 \pm 0.08969$ | $0.06783 \pm 0.00391$ | $1.34817 \pm 0.10076$ |
| | FIM−ℓ($w.n. = 16$) | $0.04306 \pm 0.0$ | $2.85851 \pm 0.14102$ | $0.05609 \pm 0.00569$ | $4.54185 \pm 0.25996$ |
| 2 | ODEFormer (g.t. traj.) | $9.48347 \pm 0.0$ | $115.23801 \pm 298.68335$ | $9.06534 \pm 3.45804$ | $100.57544 \pm 239.8902$ |
| | ODEFormer (inf. traj.) | $1.29766 \pm 0.0$ | $1.94693 \pm 2.33017$ | $3.10378 \pm 2.06017$ | $3.8734 \pm 3.27421$ |
| | FIM−ℓ($w.n. = 2$) | $1.43282 \pm 0.0$ | $1.41955 \pm 0.00186$ | $1.21969 \pm 0.01669$ | $1.22876 \pm 0.01812$ |
| | FIM−ℓ($w.n. = 4$) | $0.70673 \pm 0.0$ | $0.73517 \pm 0.00655$ | $0.69129 \pm 0.03759$ | $0.7396 \pm 0.03932$ |
| | FIM−ℓ($w.n. = 8$) | $0.21854 \pm 0.0$ | $\mathbf{0.32851 \pm 0.01379}$ | $0.30857 \pm 0.04604$ | $\mathbf{0.48065 \pm 0.05582}$ |
| | FIM−ℓ($w.n. = 16$) | $0.09232 \pm 0.0$ | $0.51666 \pm 0.03184$ | $0.18537 \pm 0.03789$ | $0.8219 \pm 0.06753$ |
| 3 | ODEFormer (g.t. traj.) | $145.80444 \pm 0.0$ | $110.45746 \pm 67.51671$ | $358.12564 \pm 284.26566$ | $258.31259 \pm 199.26659$ |
| | ODEFormer (inf. traj.) | $86.05997 \pm 0.0$ | $66.84346 \pm 9.40672$ | $89.96053 \pm 10.58968$ | $72.54148 \pm 24.72853$ |
| | FIM−ℓ($w.n. = 2$) | $58.27983 \pm 0.0$ | $58.28185 \pm 0.00338$ | $57.97466 \pm 0.0192$ | $57.97117 \pm 0.01773$ |
| | FIM−ℓ($w.n. = 4$) | $54.04507 \pm 0.0$ | $53.92284 \pm 0.01039$ | $52.94947 \pm 0.05369$ | $52.88163 \pm 0.0522$ |
| | FIM−ℓ($w.n. = 8$) | $42.37861 \pm 0.0$ | $42.28708 \pm 0.03208$ | $42.5487 \pm 0.21984$ | $42.59756 \pm 0.22269$ |
| | FIM−ℓ($w.n. = 16$) | $21.67166 \pm 0.0$ | $\mathbf{22.1478 \pm 0.09837}$ | $27.48439 \pm 0.82561$ | $\mathbf{28.11654 \pm 0.85931}$ |
| 4 | ODEFormer (g.t. traj.) | $0.15883 \pm 0.0$ | $0.14387 \pm 0.05189$ | $0.16521 \pm 0.04465$ | $0.44315 \pm 0.47787$ |
| | ODEFormer (inf. traj.) | $0.0639 \pm 0.0$ | $0.05313 \pm 0.01146$ | $0.06529 \pm 0.01458$ | $0.92522 \pm 2.12569$ |
| | FIM−ℓ($w.n. = 2$) | $0.03748 \pm 0.0$ | $0.03646 \pm 0.00119$ | $0.01764 \pm 0.00126$ | $0.02066 \pm 0.00155$ |
| | FIM−ℓ($w.n. = 4$) | $0.00423 \pm 0.0$ | $\mathbf{0.0116 \pm 0.00102}$ | $0.00546 \pm 0.00084$ | $\mathbf{0.01556 \pm 0.00131}$ |
| | FIM−ℓ($w.n. = 8$) | $0.00385 \pm 0.0$ | $0.0245 \pm 0.00234$ | $0.00529 \pm 0.00078$ | $0.03523 \pm 0.00243$ |
| | FIM−ℓ($w.n. = 16$) | $0.00288 \pm 0.0$ | $0.0707 \pm 0.00488$ | $0.00366 \pm 0.00053$ | $0.10899 \pm 0.00559$ |

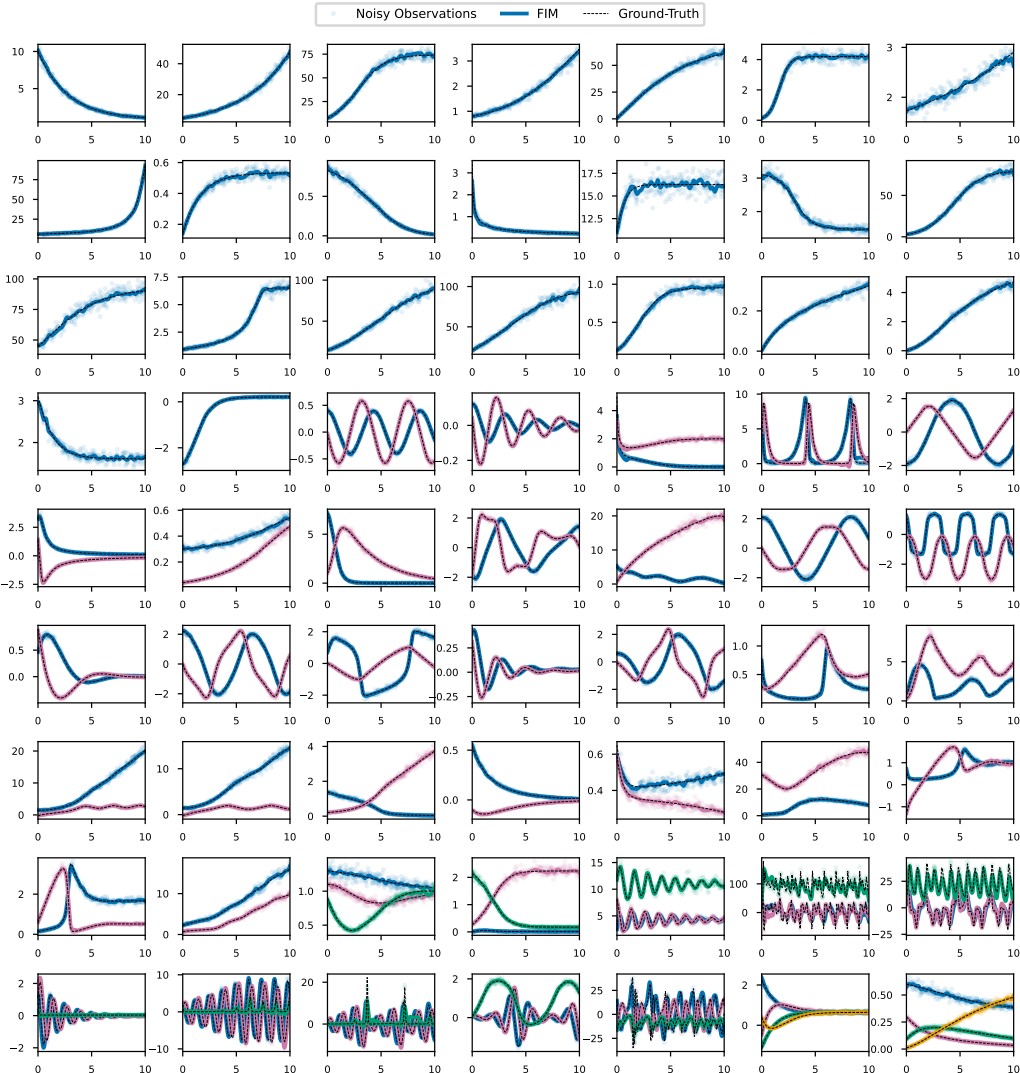

Figure 6: Reconstruction from noisy observations (dots) of all equations in the ODEBench dataset, using the first set of initial conditions, with corruptions $\rho = 0.5$ and $\gamma = 0.05$. FIM$-\ell$ (line) recovers the ground truth path (dashed line) closely for (almost) all equations.

Table 13: ODEBench MAE of the inferred time derivative of $\texttt{FIM-}\ell$ for different numbers of windows, split by dimensions. The standard deviation is calculated across 10 samplings of the corruption schemes.

| Dim. | Model | $\rho = 0$ $\gamma = 0$ | $\rho = 0$ $\gamma = 0.05$ | $\rho = 0.5$ $\gamma = 0$ | $\rho = 0.5$ $\gamma = 0.05$ |
|---|---|---|---|---|---|
| All | ODEFormer (g.t. traj.) | $9.98246 \pm 0.0$ | $10.3509 \pm 5.12400$ | $11.43592 \pm 3.89992$ | $14.56629 \pm 5.62873$ |
| | ODEFormer (inf. traj.) | $7.47623 \pm 0.0$ | $7.59557 \pm 0.80806$ | $8.03686 \pm 0.38837$ | $7.87698 \pm 0.61724$ |
| | $\texttt{FIM-}\ell(w.n. = 2)$ | $6.71538 \pm 0.0$ | $6.71455 \pm 0.00866$ | $6.47138 \pm 0.00666$ | $6.51303 \pm 0.00678$ |
| | $\texttt{FIM-}\ell(w.n. = 4)$ | $5.86143 \pm 0.0$ | $5.92169 \pm 0.00801$ | $5.64520 \pm 0.01431$ | $5.74910 \pm 0.01754$ |
| | $\texttt{FIM-}\ell(w.n. = 8)$ | $4.21719 \pm 0.0$ | $4.45152 \pm 0.01683$ | $4.21716 \pm 0.01543$ | $4.58770 \pm 0.02492$ |
| | $\texttt{FIM-}\ell(w.n. = 16)$ | $1.90556 \pm 0.0$ | $\mathbf{2.75833 \pm 0.03090}$ | $2.44375 \pm 0.05500$ | $\mathbf{3.79191 \pm 0.04925}$ |
| 1 | ODEFormer (g.t. traj.) | $1.02333 \pm 0.0$ | $0.85939 \pm 0.28399$ | $1.93692 \pm 0.79555$ | $1.69855 \pm 1.37894$ |
| | ODEFormer (inf. traj.) | $1.22521 \pm 0.0$ | $0.97085 \pm 0.44759$ | $1.79378 \pm 0.60934$ | $1.74534 \pm 1.01677$ |
| | $\texttt{FIM-}\ell(w.n. = 2)$ | $0.35084 \pm 0.0$ | $0.35811 \pm 0.02167$ | $0.06987 \pm 0.00370$ | $\mathbf{0.17388 \pm 0.01507}$ |
| | $\texttt{FIM-}\ell(w.n. = 4)$ | $0.03440 \pm 0.0$ | $\mathbf{0.21994 \pm 0.01723}$ | $0.02992 \pm 0.00100$ | $0.29445 \pm 0.02783$ |
| | $\texttt{FIM-}\ell(w.n. = 8)$ | $0.01847 \pm 0.0$ | $0.57124 \pm 0.04161$ | $0.02071 \pm 0.00083$ | $0.85952 \pm 0.04000$ |
| | $\texttt{FIM-}\ell(w.n. = 16)$ | $0.01120 \pm 0.0$ | $1.83930 \pm 0.07291$ | $0.01563 \pm 0.00059$ | $2.98943 \pm 0.12402$ |
| 2 | ODEFormer (g.t. traj.) | $1.11424 \pm 0.0$ | $4.60410 \pm 9.66290$ | $1.25932 \pm 0.11240$ | $4.67499 \pm 7.50841$ |
| | ODEFormer (inf. traj.) | $0.55929 \pm 0.0$ | $0.62367 \pm 0.29012$ | $0.73061 \pm 0.14608$ | $0.80667 \pm 0.31722$ |
| | $\texttt{FIM-}\ell(w.n. = 2)$ | $0.78924 \pm 0.0$ | $0.78135 \pm 0.00215$ | $0.59850 \pm 0.00569$ | $0.60812 \pm 0.00550$ |
| | $\texttt{FIM-}\ell(w.n. = 4)$ | $0.33064 \pm 0.0$ | $0.35679 \pm 0.00296$ | $0.29782 \pm 0.00601$ | $0.33816 \pm 0.00718$ |
| | $\texttt{FIM-}\ell(w.n. = 8)$ | $0.08201 \pm 0.0$ | $\mathbf{0.15581 \pm 0.00496}$ | $0.10640 \pm 0.00910$ | $\mathbf{0.21536 \pm 0.01454}$ |
| | $\texttt{FIM-}\ell(w.n. = 16)$ | $0.02903 \pm 0.0$ | $0.26067 \pm 0.01393$ | $0.05549 \pm 0.00629$ | $0.41416 \pm 0.01805$ |
| 3 | ODEFormer (g.t. traj.) | $57.40099 \pm 0.0$ | $50.32888 \pm 20.88564$ | $64.04977 \pm 24.41574$ | $74.73737 \pm 25.89933$ |
| | ODEFormer (inf. traj.) | $42.71272 \pm 0.0$ | $43.86978 \pm 4.69668$ | $44.45747 \pm 2.04799$ | $43.33972 \pm 3.66024$ |
| | $\texttt{FIM-}\ell(w.n. = 2)$ | $39.28646 \pm 0.0$ | $39.28669 \pm 0.00501$ | $38.93203 \pm 0.04304$ | $38.92776 \pm 0.04075$ |
| | $\texttt{FIM-}\ell(w.n. = 4)$ | $35.92175 \pm 0.0$ | $35.80043 \pm 0.01442$ | $34.66166 \pm 0.08952$ | $34.59348 \pm 0.08569$ |
| | $\texttt{FIM-}\ell(w.n. = 8)$ | $26.29593 \pm 0.0$ | $26.29148 \pm 0.01758$ | $26.22219 \pm 0.10493$ | $26.31823 \pm 0.09887$ |
| | $\texttt{FIM-}\ell(w.n. = 16)$ | $11.89775 \pm 0.0$ | $\mathbf{12.40848 \pm 0.03408}$ | $15.20400 \pm 0.34251$ | $15.83980 \pm 0.32635$ |
| 4 | ODEFormer (g.t. traj.) | $0.07487 \pm 0.0$ | $0.0685 \pm 0.02815$ | $0.07772 \pm 0.01916$ | $0.16812 \pm 0.1964$ |
| | ODEFormer (inf. traj.) | $0.01773 \pm 0.0$ | $0.01533 \pm 0.00195$ | $0.01658 \pm 0.00155$ | $0.06136 \pm 0.09291$ |
| | $\texttt{FIM-}\ell(w.n. = 2)$ | $0.01824 \pm 0.0$ | $0.01785 \pm 0.00047$ | $0.00565 \pm 0.0006$ | $0.00828 \pm 0.00107$ |
| | $\texttt{FIM-}\ell(w.n. = 4)$ | $0.00178 \pm 0.0$ | $\mathbf{0.00667 \pm 0.00026}$ | $0.00179 \pm 0.0001$ | $\mathbf{0.00892 \pm 0.00061}$ |
| | $\texttt{FIM-}\ell(w.n. = 8)$ | $0.00135 \pm 0.0$ | $0.01492 \pm 0.00074$ | $0.00168 \pm 0.00008$ | $0.02200 \pm 0.00134$ |
| | $\texttt{FIM-}\ell(w.n. = 16)$ | $0.00117 \pm 0.0$ | $0.04355 \pm 0.00189$ | $0.00143 \pm 0.00005$ | $0.06961 \pm 0.00279$ |

## F.4 LORENZ SYSTEM AND LATENTODE

### F.4.1 DATA DESCRIPTION AND PRE-PROCESSING

To compare our zero-shot model to a LatentODE, a specialised black-box model requiring training, we consider the chaotic Lorenz system

$$
\begin{aligned}
dx &= \sigma(y - x) \\
dy &= \rho x - y - xz \\
dz &= xy - \beta z
\end{aligned}
$$

with parameters $(\sigma, \beta, \rho) = (10, 28, \frac{8}{3})$. We simulate the system in the time interval $[0, 10]$, starting with initial values sampled from $\mathcal{N}([2.3, 8.1, 12.4], 1)$. Note that the Lorenz system with these parameters and the mean initial value is part of the ODEBench dataset.

To generate a time series, we sample an initial value, simulate the system and observe it at the regular grid on $[0, 1]$ with 512 points. The observations are subsampled with a survival probability of $0.5$ and corrupted with additive gaussian noise sampled from $\mathcal{N}(0, 0.05)$.

We generate 1024 time series for the training of LatentODE and additional 128 time series each for validation and test.

### F.4.2 TRAINING LATENTODE

We train LatentODE (Rubanova et al., 2019) on the standardised train set, selecting the model based on the lowest loss on the validation set. The encoder is an LSTM, the emission model a diagonal gaussian, fixing its standard deviation to 0.01. The target objective is the likelihood of the observations.

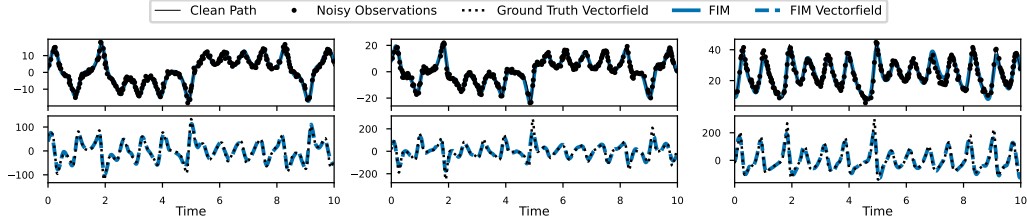

Figure 7: Inference of $\texttt{FIM-}\ell(w.n. = 16)$ on a Lorenz system time series, split int their individual dimensions. (**Top**) Noisy observations (black dots) of the system (black line) are interpolated by $\texttt{FIM-}\ell$ (blue line). (**Bottom**) Inferred values of the time derivative (blue dashed line) match the ground truth vector field values along the solution path (black dashed line) closely.

We train for 6000 epochs, with minibatches of size 32, using AdamW with learning rate $1e^{-3}$ and weight decay $1e^{-2}$. To help the model learn, we slowly anneal the input time series length over the initial 2000 epochs, starting at 25 observations.

We selected the model architecture by grid search. See Table 14 for the search grid and final parameters. The model trained roughly 9 hours on a A100 40GB GPU.

Table 14: Hyperparameters, including grid search range over some, for the LatentODE baseline on all datasets. Selected hyperparameters, determined by the loss on the validation set, are underlined. For simplicity, we set the dimension of the forward neural ODE to the hidden size of the encoder.

| Hyperparameter | Lorenz System | Motion Capture | Navier Stokes |
|---|---|---|---|
| Hidden size | $\underline{64}, 128, 256$ | $\underline{32}, 64, 128$ | $\underline{64}, 128, 256$ |
| NeuralODE hidden layers | $[64, 64], [128, 128]$ | $[64, 64]$ | $[128, 128]$ |
| Initial condition hidden layers | $[64, 64], [128, 128]$ | $[64, 64]$ | $[128, 128]$ |
| Emission model hidden layers | $[64, 64], [128, 128]$ | $[64, 64]$ | $[128, 128]$ |

### F.4.3 MODELLING AND RESULTS

Note that Latent ODE is notoriously difficult to train and we only managed to fit it to the data on a 64-dimensional hidden space (Dupont et al., 2019). The time derivative inference is therefore not directly feasible with our trained Latent ODE. We nevertheless compare $\texttt{FIM-}\ell(w.n. = 16)$ against said Latent ODE model on the reconstruction task, as well as against the Scipy[10] implementation of the cubic spline.

Table 15 displays our results in terms of RMSE calculated against the ground-truth, clean solution path on all 512 of the regular grid. They averaged over the 128 trajectories in the test set for described data corruption scenario. $\texttt{FIM-}\ell$ not only outperforms both baselines, but also perfectly infers the hidden vector field along the solution path, as can be seen in Figure 7.

Table 15: Reconstruction task on the Lorenz system data. RMSE is calculated on the whole trajectory.

| Model | Lorenz system |
|---|---|
| LatentODE | $3.25 \pm 0.99$ |
| Cubic spline | $3.97 \pm 5.8$ |
| $\texttt{FIM-}\ell$ | $2.01 \pm 0.33$ |

---

[10]https://scipy.org/

### F.4.4 Ablation Study: Number of Windows

Additionally, we perform an ablation study on the numbers of windows processed by FIM−ℓ on the Lorenz test set.

Figure 8 shows the RMSE as the number of windows increases from 1 to 32.

Initially, the RMSE is considerably higher than the LatentODE baseline, demonstrating that FIM−ℓ on it own can not handle arbitrarily complex systems, because it has not been trained on such complex data. With the addition of more windows, the RMSE decreases sharply. However, beyond approximately 14 windows, the RMSE reduction tapers off, suggesting diminishing returns with further increases in the number of windows.

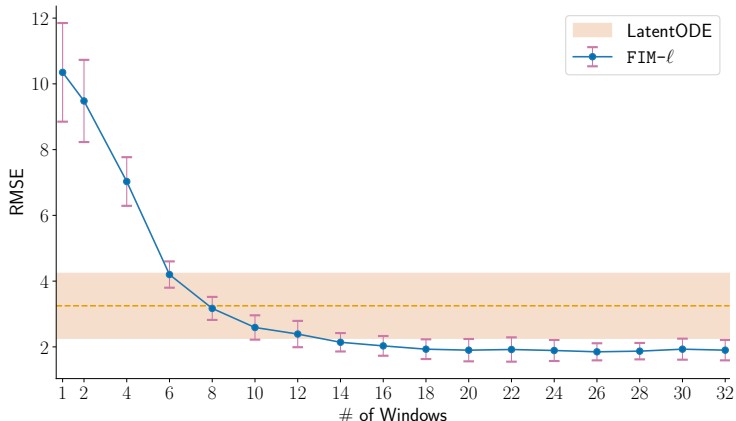

Figure 8: RMSE of LatentODE of FIM−ℓ(w.n. = m) for numbers of windows $m$ between 1 and 32 on the Lorenz test set. The standard deviation across 128 time series is represented by the shaded area for the LatentODE and by the whiskers for each FIM−ℓ(w.n. = m). The performance of FIM−ℓ increases, as the number of windows increase.

## F.5 Comparison against Neural ODE Processes

### F.5.1 Data Description and Pre-Processing

Norcliffe et al. (2021) introduce four 1D synthetic datasets of different functional type, including sines, exponentials, straight lines, and harmonic oscillators. Each task is defined by a parameterized function, where the parameters are sampled from predefined uniform distributions. An example trajectory is generated by sampling from these parameter distributions and then sampling from the function at evenly spaced timestamps, $t$, over a fixed range to produce 100 data points $(t, y)$. The equations for these tasks, along with the ranges for their defining parameters, are provided in Table 16.

10 random context points are selected out of all 100 points. They serve as the model inputs, where the task is to reconstruct the original function at all 100 points.

Table 16: Description of functions considered by Norcliffe et al. (2021), including the mathematical form, ranges for parameters $a$ and $b$, range for $t$, and the number of test samples.

| Task | Form | a | b | t | # test |
|------|------|---|---|---|--------|
| Sines | $y = a \sin(t - b)$ | $(-1, 1)$ | $(-1/2, 1/2)$ | $(-\pi, \pi)$ | 10 |
| Exponentials | $y = a/60 \times \exp(t - b)$ | $(-1, 1)$ | $(-1/2, 1/2)$ | $(-1, 4)$ | 10 |
| Straight lines | $y = at + b$ | $(-1, 1)$ | $(-1/2, 1/2)$ | $(0, 5)$ | 10 |
| Oscillators | $y = a \sin(t - b) \exp(-t/2)$ | $(-1, 1)$ | $(-1/2, 1/2)$ | $(0, 5)$ | 10 |

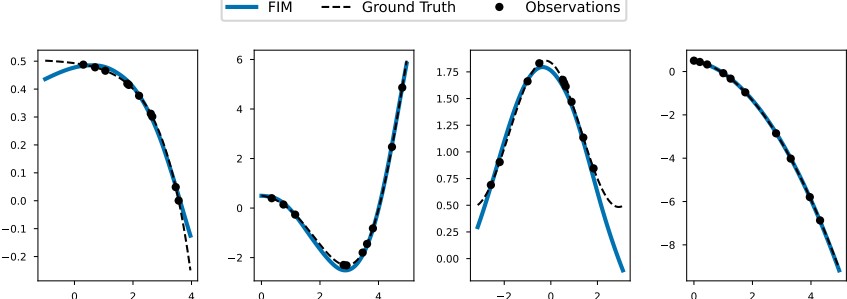

Figure 9: Samples from the one dimensional regression dataset of Norcliffe et al. (2021). The ground-truth path (black dashed lines) are observed at 10 irregular observation times (black dots). FIM−ℓ interpolates (blue line) well and recovers the ground-truth dynamics.

### F.5.2 MODELLING AND RESULTS

In Table 17, we present the performance results of our pre-trained model FIM−$\ell(w.n. = 1)$ on the 1D Regression Task proposed by Norcliffe et al. (2021). MSE, MAE $R^2$ are calculated over all the points (not just a subset of the target points) for 10 different samples from each function type. The metrics from 10 samples per function type are averaged, and we report the mean and standard deviation.

It is important to reiterate that our model has not been trained on this dataset. Still, our model outperforms the baseline from Norcliffe et al. (2021) in three out of the four tasks. The Oscillator set is the only one where our model underperforms. Its trajectories exhibit rapid changes at the edges of the considered interval, which FIM−$\ell$ can not identify based solely on the sparse observations.

Figure 9 shows four samples from the test set and the corresponding inferences of our model.

Table 17: Results on the 1D regression dataset from Norcliffe et al. (2021). We report the mean and standard deviation over 10 samples of each function class. The Neural ODE Processes baselines are selected as the top performing model from Table 1 of Norcliffe et al. (2021).

| Function | Model | $R^2\uparrow$ | MAE×$10^{-2}\downarrow$ | MSE×$10^{-2}\downarrow$ |
|---|---|---|---|---|
| Exponentials | Neural ODE Processes | - | - | $0.25 \pm 0.04$ |
| | FIM−$\ell(w.n. = 1)$ | $0.9656 \pm 0.0079$ | $1.11 \pm 0.11$ | $\mathbf{0.07 \pm 0.02}$ |
| Linear | Neural ODE Processes | - | - | $3.16 \pm 0.76$ |
| | FIM−$\ell(w.n. = 1)$ | $0.9992 \pm 0.0002$ | $3.85 \pm 0.47$ | $\mathbf{0.34 \pm 0.09}$ |
| Oscillators | Neural ODE Processes | - | - | $\mathbf{0.55 \pm 0.03}$ |
| | FIM−$\ell(w.n. = 1)$ | $0.8822 \pm 0.028$ | $20.48 \pm 2.54$ | $22.62 \pm 7.22$ |
| Sine | Neural ODE Processes | - | - | $2.09 \pm 0.12$ |
| | FIM−$\ell(w.n. = 1)$ | $0.9679 \pm 0.0042$ | $4.84 \pm 0.53$ | $\mathbf{0.67 \pm 0.16}$ |

## G POINT-WISE MISSING PATTERN IMPUTATION: ADDITIONAL RESULTS

### G.1 COMPARING AGAINST BAYOTIDE

#### G.1.1 DATA DESCRIPTION AND PRE-PROCESSING

For the *point-wise missing pattern imputation* model, we first consider two available datasets from Fang et al. (2024), namely the *Traffic-Guangzhou* and the *Solar* dataset.

*Traffic-Guangzhou* is a single time series with 500 observations and 214 components. *Solar* is a single time series with 52560 observations and 137 components.

For each of these datasets, the authors drop either 30% or 50% of the observed values. Importantly, components are dropped independent from each other, such that at any given observation time, components are (likely) only partially observed.

For our experiments, we use the pre-processed datasets provided by the authors[11], including the sampled mask indicating dropped values.

### G.1.2 MODELLING AND RESULTS

We apply $\text{FIM}-\ell(w.n. = m)$ with different number of windows $m$ to both datasets and both observation ratios and compare our results to the baselines extracted from Tables 2 and 3 from Fang et al. (2024).

Following the experimental setup of Fang et al. (2024), we compute the metrics RMSE and MAE only at the missing values. Table 18 contains the results with an observation ratio of 50% and Table 19 the results for an observation ratio of 70%.

The two datasets are of vastly different lengths. Still, with a suitable number of windows, $\text{FIM}-\ell$ can impute the missing values well in both datasets. It even outperforms all available baselines, even though they have been trained on the datasets.

The center plot of Figure 3 displays a partial time series of one component of the *Traffic-Guangzhou* dataset, including the missing values and imputation by $\text{FIM}-\ell$. We can see that $\text{FIM}-\ell$ captures the local patterns well and uses them effectively to impute the missing values.

Table 18: Performance on two datasets from Fang et al. (2024) with observed ratio 50%. Baselines have be extracted from Table 2 of Fang et al. (2024). We highlight the overall best model per dataset in bold and underline our best-performing model.

| Method | Traffic-GuangZhou | | Solar-Power | |
| --- | --- | --- | --- | --- |
| | RMSE | MAE | RMSE | MAE |
| SimpleMean | 9.852 | 7.791 | 3.213 | 2.212 |
| BRITS | 4.874 | 3.335 | 2.842 | 1.985 |
| NAOMI | 5.986 | 4.543 | 2.918 | 2.112 |
| SAITS | 4.839 | 3.391 | 2.791 | 1.827 |
| TIDER | 4.708 | 3.469 | 1.679 | 0.838 |
| Multi-Task GP | 4.887 | 3.530 | 2.847 | 1.706 |
| GP-VAE | 4.844 | 3.419 | 3.720 | 1.810 |
| CSDI | 4.813 | 3.202 | 2.276 | 0.804 |
| CSBI | 4.790 | 3.182 | 2.097 | 1.033 |
| BayOTIDE-fix weight | 11.032 | 9.294 | 5.245 | 2.153 |
| BayOTIDE-trend only | 4.188 | 2.875 | 1.789 | 0.791 |
| BayOTIDE | 3.820 | 2.687 | 1.699 | 0.734 |
| $\text{FIM}-\ell(w.n. = 1)$ | 10.614 | 8.113 | - | - |
| $\text{FIM}-\ell(w.n. = 2)$ | 7.690 | 5.648 | - | - |
| $\text{FIM}-\ell(w.n. = 4)$ | 5.377 | 3.861 | - | - |
| $\text{FIM}-\ell(w.n. = 8)$ | 4.440 | 3.090 | - | - |
| $\text{FIM}-\ell(w.n. = 16)$ | 3.741 | 2.562 | - | - |
| $\text{FIM}-\ell(w.n. = 32)$ | **3.600** | **2.427** | - | - |
| $\text{FIM}-\ell(w.n. = 800)$ | - | - | 2.044 | 1.188 |
| $\text{FIM}-\ell(w.n. = 1600)$ | - | - | 1.667 | 0.755 |
| $\text{FIM}-\ell(w.n. = 3200)$ | - | - | **1.550** | **0.595** |

## G.2 COMPARING AGAINST SAITS

### G.2.1 DATA DESCRIPTION AND PRE-PROCESSING

Du et al. (2023) collect four time series datasets (*PhysioNet-2012*, *Air-Quality*, *Electricity* and *ETT*) and benchmark a range of imputation models, including their own, on several imputation tasks. For detailed information about each of these datasets, including the sequence lengths and number of samples, we refer the reader to Table 1 of Du et al. (2023).

---

[11]https://github.com/xuangu-fang/BayOTIDE

Table 19: Performance on two datasets from Fang et al. (2024) with observed ratio 70%. Baselines have be extracted from Table 3 of Fang et al. (2024). We highlight the overall best model per dataset in bold and underline our best-performing model.

| Method | Traffic-GuangZhou | | Solar-Power | |
|---|---|---|---|---|
| | RMSE | MAE | RMSE | MAE |
| SimpleMean | 10.141 | 8.132 | 3.156 | 2.319 |
| BRITS | 4.416 | 3.003 | 2.617 | 1.861 |
| NAOMI | 5.173 | 4.013 | 2.702 | 2.003 |
| SAITS | 4.407 | 3.025 | 2.359 | 1.575 |
| TIDER | 4.168 | 3.098 | 1.676 | 0.874 |
| Multi-Task GP | 4.471 | 3.223 | 2.618 | 1.418 |
| GP-VAE | 4.373 | 3.156 | 3.561 | 1.723 |
| CSDI | 4.301 | 2.991 | 2.132 | 1.045 |
| CSBI | 4.201 | 2.955 | 1.987 | 0.926 |
| BayOTIDE-fix weight | 13.319 | 9.290 | 5.238 | 2.026 |
| BayOTIDE-trend only | 4.002 | 2.759 | 1.651 | 0.712 |
| BayOTIDE | 3.724 | 2.611 | 1.621 | 0.709 |
| $\texttt{FIM-}\ell(w.n. = 1)$ | 12.533 | 9.680 | - | - |
| $\texttt{FIM-}\ell(w.n. = 2)$ | 9.020 | 6.810 | - | - |
| $\texttt{FIM-}\ell(w.n. = 4)$ | 5.584 | 4.024 | - | - |
| $\texttt{FIM-}\ell(w.n. = 8)$ | 4.401 | 3.084 | - | - |
| $\texttt{FIM-}\ell(w.n. = 16)$ | 3.387 | 2.360 | - | - |
| $\texttt{FIM-}\ell(w.n. = 32)$ | **3.087** | **2.148** | - | - |
| $\texttt{FIM-}\ell(w.n. = 800)$ | - | - | 2.016 | 1.181 |
| $\texttt{FIM-}\ell(w.n. = 1600)$ | - | - | 1.470 | 0.655 |
| $\texttt{FIM-}\ell(w.n. = 3200)$ | - | - | **1.282** | **0.474** |

Here, we only consider one of their imputation tasks: point-wise imputation of 10% missing observations. The authors provide pre-processed datasets for this pattern[12], which we use for the following evaluation. Note that their pre-processed data includes pre-sampled masks for the missing values, to enable a fair comparison to their results.

### G.2.2 MODELLING AND RESULTS

Du et al. (2023) report MAE, RMSE and MRE for all datasets and a wide range of imputation models and methods in their Table 2, which we include here for completeness. We apply $\texttt{FIM-}\ell(w.n. = m)$ with different numbers of windows $m$ and $\texttt{FIM-}\ell(o.n. = n)$ for different window lengths $n$ to all datasets and report our results, together with the baselines, in Table 20.

With the right number or size of windows, $\texttt{FIM-}\ell$ is competitive on three out of four datasets. The *PhysioNet-2012* dataset is naturally sparse, where some components of a time series only contain a few observations. As a zero-shot approach that employs a channel independent strategy, $\texttt{FIM-}\ell$ does not perform well in such situations. In other words, *PhysioNet-2012* demonstrates the limitations of $\texttt{FIM-}\ell$ outlined in Appendix C.6.

### G.3 TSI-BENCH

### G.3.1 DATA DESCRIPTION AND PRE-PROCESSING

TSI-Bench is a collection of eight time series imputation datasets from different application domains: air-quality, traffic, energy and healthcare, assembled by Du et al. (2024). Its authors provide pre-processed splits of each dataset[13] for several imputation patterns.

Here, we consider their 10% and 50% point missing observation pattern. Note that the pre-processed datasets include sampled masks for missing values, enabling a fair comparison to their results.

---

[12] https://github.com/WenjieDu/SAITS
[13] https://github.com/WenjieDu/Awesome_Imputation

Table 20: Performance on datasets from Du et al. (2023) with 10% imputation data. Baselines have be extracted from Table 2 of Du et al. (2023). We highlight the overall best model per dataset in bold and underline our best-performing model.

| Method | PhysioNet-2012 MAE | RMSE | MRE | Air-Quality MAE | RMSE | MRE | Electricity MAE | RMSE | MRE | ETT MAE | RMSE | MRE |
|---|---|---|---|---|---|---|---|---|---|---|---|---|
| Median | 0.726 | 0.988 | 103.5% | 0.763 | 1.175 | 107.4% | 2.056 | 2.732 | 110.1% | 1.145 | 1.847 | 139.1% |
| Last | 0.862 | 1.207 | 123.0% | 0.967 | 1.408 | 136.3% | 1.006 | 1.533 | 53.9% | 1.007 | 1.365 | 96.4% |
| GRUI-GAN | 0.765 | 1.040 | 109.1% | 0.788 | 1.179 | 111.0% | - | - | - | 0.612 | 0.729 | 95.1% |
| $E^2$GAN | 0.702 | 0.964 | 100.1% | 0.750 | 1.126 | 105.6% | - | - | - | 0.584 | 0.703 | 89.0% |
| M-RNN | 0.533 | 0.776 | 76.0% | 0.294 | 0.643 | 41.4% | 1.244 | 1.867 | 66.6% | 0.376 | 0.428 | 31.6% |
| GP-VAE | 0.398 | 0.630 | 56.7% | 0.268 | 0.614 | 37.7% | 1.094 | 1.565 | 58.6% | 0.274 | 0.307 | 15.5% |
| BRITS | 0.256 | 0.767 | 36.5% | 0.153 | 0.525 | 21.6% | 0.847 | 1.322 | 45.3% | 0.130 | 0.259 | 12.5% |
| Transformer | 0.190 | 0.445 | 26.9% | 0.158 | 0.521 | 22.3% | 0.823 | 1.301 | 44.0% | 0.114 | 0.173 | 10.9% |
| SAITS-base | 0.192 | 0.439 | 27.3% | 0.146 | 0.521 | 20.6% | 0.822 | 1.221 | 44.0% | 0.121 | 0.197 | 11.6% |
| SAITS | **0.186** | **0.431** | **26.6%** | **0.137** | 0.518 | **19.3%** | 0.735 | 1.162 | 39.4% | **0.092** | **0.139** | **8.8%** |
| FIM$-\ell(w.n. = 1)$ | 0.443 | 0.899 | 63.3% | 0.167 | 0.433 | 23.6% | 0.251 | 0.251 | 6.3% | 0.108 | 0.178 | 10.5% |
| FIM$-\ell(w.n. = 2)$ | 0.414 | 0.888 | 59.1% | 0.146 | **0.414** | 20.7% | 0.091 | 0.199 | 4.9% | 0.105 | 0.176 | 10.2% |
| FIM$-\ell(w.n. = 4)$ | 0.406 | 0.849 | 58.0% | 0.143 | 0.461 | 20.2% | 0.078 | 0.174 | 4.2% | 0.108 | 0.186 | 10.5% |
| FIM$-\ell(w.n. = 8)$ | 0.409 | 0.751 | 58.4% | 0.186 | 0.541 | 26.0% | 0.071 | 0.164 | 3.8% | 0.134 | 0.252 | 13.0% |
| FIM$-\ell(w.n. = 16)$ | 0.479 | 0.842 | 68.5% | 0.394 | 0.747 | 55.7% | 0.070 | 0.162 | **3.7%** | 0.489 | 0.912 | 47.5% |
| FIM$-\ell(w.n. = 32)$ | - | - | - | - | - | - | 0.074 | 0.179 | 3.9% | - | - | - |
| FIM$-\ell(o.n. = 4)$ | 0.364 | 0.784 | 52.0% | 0.137 | 0.438 | 19.3% | 0.069 | 0.160 | 3.7% | 0.106 | 0.184 | 10.3% |
| FIM$-\ell(o.n. = 6)$ | 0.483 | 1.047 | 69.0% | 0.163 | 0.460 | 23.1% | 0.126 | 0.329 | 6.6% | 0.117 | 0.194 | 11.0% |
| FIM$-\ell(o.n. = 8)$ | 0.478 | 1.083 | 68.2% | 0.165 | 0.443 | 23.3% | 0.118 | 0.312 | 6.3% | 0.114 | 0.195 | 11.1% |
| FIM$-\ell(o.n. = 12)$ | 0.461 | 1.048 | 65.8% | 0.166 | 0.459 | 23.5% | 0.114 | 0.288 | 6.1% | 0.110 | 0.182 | 10.6% |
| FIM$-\ell(o.n. = 12)$ | 0.457 | 1.016 | 65.4% | 0.165 | 0.447 | 23.3% | 0.112 | 0.285 | 6.0% | 0.109 | 0.182 | 10.6% |
| FIM$-\ell(o.n. = 16)$ | 0.453 | 0.987 | 64.7% | 0.167 | 0.433 | 23.6% | 0.111 | 0.271 | 6.0% | 0.108 | 0.178 | 10.5% |

For details about each individual time series dataset, including the pre-processing, we refer the reader to Appendix A of (Du et al., 2024).

### G.3.2 MODELLING AND RESULTS

We apply FIM$-\ell(w.n. = m)$ with different numbers of windows $m$ and FIM$-\ell(o.n. = n)$ for different window lengths $n$ to all available datasets in both the 10% and 50% missingness patterns. In accordance with Du et al. (2024), we compute the performance metrics only at the missing values.

Table 21 contains the MAE of all baseline models (extracted from Table 2 of (Du et al., 2024)) and FIM$-\ell$ in the 10% missingness pattern. Table 22 contains the MAE and MSE of all baseline models (extracted from Table 11 of (Du et al., 2024)) and FIM$-\ell$ in the 50% missingness pattern.

Because of the short, and sometimes very sparse data (in particular in the 50% missingness pattern), we experimented with different application strategies for FIM$-\ell$ and report the corresponding results in both tables.

Let us now compare the two window specifying methods, FIM$-\ell(w.n. = m)$ and FIM$-\ell(o.n. = n)$ on the TSI-Bench datasets. Here, specifying the windows by their number of observations performs better overall, in particular in the 50% missingness pattern of Table 22. In the 10% missingness pattern, the (relative) difference between the two approaches is smaller, although still present.

With its windowing scheme, FIM$-\ell$ is not limited by the length of time series it can process. Some datasets in the TSI-Bench are based on a single long time series, that is split up into smaller chunks during pre-processing, to accommodate methods with such limitations. For these datasets, we experimented with *reassembling* the original time series via concatenation, before applying FIM$-\ell$. We denote these models by Long-FIM$-\ell(w.n. = m)$.

The reassembled time series could provide more context for our zero-shot method to better extract the (local) patterns. Table 22 reveals that such strategy can indeed improve the performance, in particular in terms of the MSE. However, it does not improve the performance in all datasets and for all metrics.

Table 21: MAE on datasets from Du et al. (2024) with 10% point missingness. Baselines have been extracted from Table 2 of Du et al. (2024). Parenthesis indicate the standard deviation of five training rounds of neural models. We highlight the overall best model per dataset in bold and underline our best-performing model.

| Method | BeijingAir | ItalyAir | PeMS | Pedestrian | ETT.h1 | Electricity | PhysioNet2012 | PhysioNet2019 |
|---|---|---|---|---|---|---|---|---|
| iTransformer | 0.123 (0.005) | 0.223 (0.014) | 0.226 (0.001) | 0.148 (0.005) | 0.263 (0.004) | 0.571 (0.178) | 0.379 (0.002) | 0.462 (0.006) |
| SAITS | 0.155 (0.004) | 0.185 (0.010) | 0.287 (0.001) | 0.131 (0.006) | **0.144** (0.006) | 1.377 (0.026) | 0.257 (0.019) | 0.352 (0.005) |
| Nonstationary | 0.209 (0.002) | 0.266 (0.007) | 0.331 (0.017) | 0.453 (0.024) | 0.359 (0.013) | 0.213 (0.014) | 0.410 (0.002) | 0.458 (0.001) |
| ETSformer | 0.187 (0.002) | 0.259 (0.004) | 0.347 (0.006) | 0.207 (0.011) | 0.227 (0.007) | 0.412 (0.005) | 0.373 (0.003) | 0.451 (0.005) |
| PatchTST | 0.198 (0.011) | 0.274 (0.026) | 0.330 (0.013) | 0.126 (0.003) | 0.240 (0.013) | 0.550 (0.039) | 0.301 (0.011) | 0.420 (0.007) |
| Crossformer | 0.184 (0.004) | 0.246 (0.011) | 0.337 (0.007) | **0.119** (0.005) | 0.232 (0.008) | 0.540 (0.034) | 0.525 (0.202) | 0.378 (0.007) |
| Informer | 0.148 (0.002) | 0.205 (0.008) | 0.302 (0.003) | 0.154 (0.010) | 0.167 (0.006) | 1.291 (0.031) | 0.297 (0.003) | 0.403 (0.002) |
| Autoformer | 0.257 (0.012) | 0.295 (0.008) | 0.598 (0.074) | 0.197 (0.008) | 0.267 (0.008) | 0.748 (0.027) | 0.417 (0.009) | 0.476 (0.002) |
| Pyraformer | 0.178 (0.004) | 0.217 (0.006) | 0.285 (0.003) | 0.153 (0.012) | 0.182 (0.008) | 1.096 (0.033) | 0.294 (0.002) | 0.387 (0.004) |
| Transformer | 0.142 (0.001) | 0.191 (0.010) | 0.294 (0.002) | 0.136 (0.009) | 0.178 (0.015) | 1.316 (0.036) | 0.259 (0.006) | **0.341** (0.002) |
| BRITS | 0.127 (0.001) | 0.235 (0.007) | 0.271 (0.000) | 0.149 (0.005) | 0.145 (0.002) | 0.971 (0.016) | 0.297 (0.001) | 0.355 (0.001) |
| MRNN | 0.568 (0.002) | 0.638 (0.003) | 0.624 (0.000) | 0.735 (0.001) | 0.789 (0.019) | 1.824 (0.005) | 0.708 (0.029) | 0.778 (0.015) |
| GRUD | 0.233 (0.002) | 0.368 (0.012) | 0.355 (0.002) | 0.204 (0.008) | 0.325 (0.004) | 0.976 (0.015) | 0.450 (0.004) | 0.471 (0.001) |
| TimesNet | 0.230 (0.010) | 0.280 (0.004) | 0.312 (0.001) | 0.157 (0.008) | 0.254 (0.008) | 1.011 (0.016) | 0.353 (0.003) | 0.394 (0.003) |
| MICN | 0.203 (0.001) | 0.283 (0.004) | 0.281 (0.003) | - | 0.267 (0.010) | 0.392 (0.006) | 0.378 (0.013) | 0.461 (0.007) |
| SCINet | 0.191 (0.011) | 0.288 (0.010) | 0.487 (0.101) | 0.149 (0.012) | 0.246 (0.015) | 0.581 (0.015) | 0.341 (0.005) | 0.427 (0.002) |
| StemGNN | 0.161 (0.002) | 0.260 (0.008) | 0.493 (0.079) | 0.127 (0.006) | 0.248 (0.012) | 1.360 (0.078) | 0.331 (0.001) | 0.416 (0.002) |
| FreTS | 0.211 (0.008) | 0.273 (0.008) | 0.396 (0.027) | 0.138 (0.004) | 0.262 (0.029) | 0.718 (0.043) | 0.315 (0.008) | 0.406 (0.017) |
| Koopa | 0.363 (0.108) | 0.307 (0.041) | 0.532 (0.122) | 0.173 (0.020) | 0.435 (0.132) | 1.309 (0.531) | 0.413 (0.007) | 0.451 (0.019) |
| DLinear | 0.215 (0.016) | 0.242 (0.009) | 0.362 (0.009) | 0.179 (0.004) | 0.227 (0.006) | 0.519 (0.008) | 0.370 (0.000) | 0.432 (0.001) |
| FiLM | 0.318 (0.010) | 0.340 (0.011) | 0.784 (0.064) | 0.413 (0.010) | 0.583 (0.008) | 0.834 (0.031) | 0.458 (0.001) | 0.494 (0.003) |
| CSDI | **0.102** (0.010) | 0.539 (0.418) | 0.238 (0.047) | 0.231 (0.064) | 0.151 (0.008) | 1.483 (0.459) | **0.252** (0.002) | 0.408 (0.019) |
| US-GAN | 0.137 (0.002) | 0.264 (0.012) | 0.296 (0.001) | 0.151 (0.016) | 0.458 (0.590) | 0.938 (0.009) | 0.310 (0.003) | 0.358 (0.002) |
| GP-VAE | 0.240 (0.006) | 0.369 (0.012) | 0.341 (0.007) | 0.319 (0.010) | 0.329 (0.017) | 1.152 (0.074) | 0.445 (0.006) | 0.562 (0.004) |
| Mean | 0.721 | 0.574 | 0.798 | 0.728 | 0.737 | 0.422 | 0.708 | 0.762 |
| Median | 0.681 | 0.518 | 0.778 | 0.667 | 0.710 | 0.408 | 0.690 | 0.747 |
| LOCF | 0.188 | 0.233 | 0.375 | 0.257 | 0.315 | 0.104 | 0.449 | 0.478 |
| Linear | 0.112 | **0.135** | 0.211 | 0.167 | 0.197 | **0.065** | 0.366 | 0.387 |
| FIM-$\ell(w.n. = 1)$ | 0.148 | 0.162 | 0.317 | 0.234 | 0.344 | 0.105 | 0.489 | 0.448 |
| FIM-$\ell(w.n. = 2)$ | 0.131 | 0.150 | 0.247 | 0.194 | 0.263 | 0.088 | 0.460 | 0.434 |
| FIM-$\ell(w.n. = 4)$ | 0.127 | 0.196 | 0.227 | 0.170 | 0.234 | 0.078 | 0.444 | 0.436 |
| FIM-$\ell(w.n. = 8)$ | 0.165 | 0.416 | 0.311 | 0.245 | 0.234 | 0.072 | 0.445 | 0.464 |
| FIM-$\ell(w.n. = 16)$ | 0.401 | - | 0.484 | 0.449 | 0.291 | 0.071 | 0.521 | 0.538 |
| FIM-$\ell(w.n. = 32)$ | - | - | - | - | 0.501 | 0.105 | 0.580 | 0.612 |
| FIM-$\ell(o.n. = 4)$ | 0.122 | 0.144 | **0.208** | 0.171 | 0.220 | 0.069 | 0.402 | 0.418 |
| FIM-$\ell(o.n. = 6)$ | 0.145 | 0.154 | 0.277 | 0.177 | 0.295 | 0.119 | 0.536 | 0.499 |
| FIM-$\ell(o.n. = 8)$ | 0.145 | 0.162 | 0.301 | 0.184 | 0.302 | 0.117 | 0.532 | 0.487 |
| FIM-$\ell(o.n. = 8)$ | 0.144 | 0.162 | 0.307 | 0.186 | 0.308 | 0.114 | 0.523 | 0.481 |
| FIM-$\ell(o.n. = 12)$ | 0.147 | 0.162 | 0.272 | 0.200 | 0.295 | 0.111 | 0.512 | 0.469 |
| FIM-$\ell(o.n. = 12)$ | 0.146 | 0.162 | 0.274 | 0.200 | 0.295 | 0.110 | 0.507 | 0.463 |
| FIM-$\ell(o.n. = 16)$ | 0.148 | 0.162 | 0.317 | 0.234 | 0.318 | 0.110 | 0.502 | 0.461 |
| Long-FIM-$\ell(w.n. = 16)$ | - | 0.425 | - | - | - | - | - | - |
| Long-FIM-$\ell(w.n. = 32)$ | - | 0.353 | 0.945 | - | 0.733 | - | - | - |
| Long-FIM-$\ell(w.n. = 64)$ | 0.397 | 0.235 | 0.705 | - | 0.495 | - | - | - |
| Long-FIM-$\ell(w.n. = 128)$ | 0.258 | 0.174 | 0.427 | - | 0.291 | - | - | - |
| Long-FIM-$\ell(w.n. = 256)$ | 0.167 | 0.145 | 0.273 | - | 0.240 | 0.172 | - | - |
| Long-FIM-$\ell(w.n. = 512)$ | 0.133 | - | 0.217 | - | 0.225 | 0.101 | - | - |
| Long-FIM-$\ell(w.n. = 1024)$ | 0.120 | - | - | - | - | 0.081 | - | - |
| Long-FIM-$\ell(w.n. = 2048)$ | - | - | - | - | - | 0.072 | - | - |
| Long-FIM-$\ell(w.n. = 4096)$ | - | - | - | - | - | 0.070 | - | - |

Table 22: MAE and MSE on datasets from Du et al. (2024) with 50% point missingness. Baselines have been extracted from Table 11 of Du et al. (2024). Parenthesis indicate the standard deviation of five training rounds of neural models. We highlight the overall best model per dataset in bold and underline our best-performing model.

| Method | BeijingAir MAE | MSE | ItalyAir MAE | MSE | PeMS MAE | MSE | ETT_h1 MAE | MSE | Electricity MAE | MSE | Pedestrian MAE | MSE |
|---|---|---|---|---|---|---|---|---|---|---|---|---|
| iTransformer | 0.163 (0.003) | 0.233 (0.004) | 0.321 (0.007) | 0.327 (0.011) | 0.295 (0.007) | **0.539 (0.016)** | 0.348 (0.002) | 0.233 (0.003) | 0.893 (0.085) | 1.884 (0.160) | 0.200 (0.006) | 0.343 (0.006) |
| SAITS | 0.194 (0.003) | 0.193 (0.007) | 0.285 (0.010) | 0.236 (0.014) | 0.302 (0.001) | 0.595 (0.003) | **0.223 (0.007)** | **0.107 (0.005)** | 1.399 (0.069) | 3.837 (0.316) | 0.205 (0.011) | 0.392 (0.027) |
| Nonstationary | 0.231 (0.001) | 0.271 (0.007) | 0.314 (0.005) | 0.361 (0.010) | 0.394 (0.013) | 0.688 (0.016) | 0.382 (0.004) | 0.292 (0.006) | 0.217 (0.031) | 0.191 (0.048) | 0.487 (0.033) | 0.859 (0.098) |
| ETSformer | 0.249 (0.004) | 0.261 (0.009) | 0.401 (0.007) | 0.421 (0.011) | 0.386 (0.007) | 0.586 (0.009) | 0.364 (0.013) | 0.269 (0.022) | 0.878 (0.008) | 1.687 (0.024) | 0.320 (0.004) | 0.519 (0.010) |
| PatchTST | 0.210 (0.009) | 0.206 (0.007) | 0.345 (0.011) | 0.313 (0.010) | 0.348 (0.006) | 0.609 (0.008) | 0.275 (0.023) | 0.149 (0.017) | 0.856 (0.044) | 1.573 (0.141) | 0.198 (0.003) | 0.351 (0.005) |
| Crossformer | 0.215 (0.007) | 0.224 (0.004) | 0.325 (0.009) | 0.293 (0.009) | 0.357 (0.003) | 0.607 (0.008) | 0.270 (0.021) | 0.146 (0.017) | 0.980 (0.042) | 2.255 (1.656) | **0.191 (0.008)** | 0.356 (0.014) |
| Informer | 0.184 (0.005) | 0.213 (0.003) | 0.304 (0.007) | 0.247 (0.015) | 0.330 (0.005) | 0.600 (0.009) | 0.279 (0.008) | 0.162 (0.007) | 1.277 (0.028) | 3.239 (0.080) | 0.210 (0.006) | 0.378 (0.021) |
| Autoformer | 0.898 (0.001) | 1.554 (0.003) | 0.833 (0.017) | 1.880 (0.044) | 0.602 (0.068) | 1.242 (0.173) | 0.984 (0.008) | 1.553 (0.025) | 2.164 (0.001) | 8.092 (0.010) | 1.033 (0.015) | 2.273 (0.082) |
| Pyraformer | 0.198 (0.005) | 0.223 (0.011) | 0.312 (0.012) | 0.254 (0.018) | 0.305 (0.002) | 0.580 (0.004) | 0.291 (0.026) | 0.167 (0.021) | 1.131 (0.036) | 2.711 (0.079) | 0.202 (0.006) | 0.381 (0.007) |
| Transformer | 0.185 (0.003) | 0.192 (0.005) | 0.279 (0.011) | **0.230 (0.017)** | 0.316 (0.004) | 0.588 (0.005) | 0.274 (0.012) | 0.162 (0.017) | 1.365 (0.034) | 3.554 (0.085) | 0.194 (0.014) | 0.342 (0.033) |
| BRITS | 0.169 (0.001) | 0.194 (0.003) | 0.321 (0.005) | 0.283 (0.007) | **0.287 (0.001)** | 0.561 (0.002) | 0.238 (0.006) | 0.127 (0.004) | 1.124 (0.010) | 2.828 (0.023) | 0.259 (0.017) | 0.433 (0.021) |
| MRNN | 0.603 (0.006) | 0.775 (0.006) | 0.724 (0.001) | 1.391 (0.006) | 0.645 (0.001) | 1.072 (0.003) | 0.816 (0.006) | 1.219 (0.004) | 1.810 (0.004) | 5.793 (0.011) | 0.773 (0.001) | 1.258 (0.003) |
| GRUD | 0.279 (0.001) | 0.303 (0.002) | 0.476 (0.009) | 0.539 (0.011) | 0.372 (0.002) | 0.619 (0.002) | 0.417 (0.008) | 0.337 (0.005) | 1.087 (0.011) | 2.458 (0.034) | 0.307 (0.005) | 0.507 (0.007) |
| TimesNet | 0.265 (0.005) | 0.233 (0.007) | 0.370 (0.010) | 0.323 (0.012) | 0.348 (0.002) | 0.567 (0.001) | 0.339 (0.004) | 0.210 (0.004) | 1.131 (0.017) | 2.644 (0.077) | 0.269 (0.016) | 0.392 (0.017) |
| MICN | 0.456 (0.006) | 0.553 (0.013) | 0.548 (0.003) | 0.852 (0.012) | 0.392 (0.006) | 0.608 (0.010) | 0.606 (0.073) | 0.688 (0.152) | 0.965 (0.008) | 2.018 (0.032) | - | - |
| SCINet | 0.222 (0.012) | 0.230 (0.036) | 0.337 (0.008) | 0.319 (0.006) | 0.500 (0.093) | 0.849 (0.193) | 0.326 (0.014) | 0.194 (0.013) | 0.778 (0.023) | 1.162 (0.115) | 0.251 (0.005) | 0.391 (0.015) |
| StemGNN | 0.186 (0.004) | 0.263 (0.005) | 0.307 (0.014) | 0.280 (0.019) | 0.446 (0.021) | 0.862 (0.064) | 0.325 (0.019) | 0.200 (0.025) | 1.362 (0.187) | 3.803 (0.920) | 0.200 (0.009) | 0.343 (0.014) |
| FreTS | 0.235 (0.015) | 0.246 (0.010) | 0.349 (0.015) | 0.345 (0.053) | 0.422 (0.019) | 0.686 (0.027) | 0.319 (0.025) | 0.195 (0.030) | 0.871 (0.084) | 1.320 (0.275) | 0.224 (0.004) | 0.314 (0.016) |
| Koopa | 0.373 (0.079) | 0.445 (0.119) | 0.345 (0.032) | 0.359 (0.056) | 0.506 (0.114) | 0.855 (0.184) | 0.515 (0.159) | 0.577 (0.351) | 1.755 (0.250) | 7.390 (1.677) | 0.246 (0.017) | 0.330 (0.030) |
| DLinear | 0.245 (0.005) | 0.242 (0.006) | 0.340 (0.004) | 0.337 (0.005) | 0.389 (0.013) | 0.604 (0.020) | 0.311 (0.003) | 0.186 (0.003) | 0.734 (0.011) | 0.988 (0.038) | 0.310 (0.002) | 0.455 (0.006) |
| FiLM | 0.331 (0.008) | 0.409 (0.009) | 0.402 (0.018) | 0.468 (0.052) | 0.781 (0.059) | 1.499 (0.124) | 0.589 (0.005) | 0.793 (0.003) | 0.907 (0.024) | 1.434 (0.078) | 0.453 (0.007) | 0.664 (0.008) |
| CSDI | **0.144 (0.007)** | 0.472 (0.155) | 0.958 (0.551) | 29.266 (31.183) | 0.288 (0.040) | 0.651 (0.090) | 0.318 (0.016) | 0.207 (0.011) | 0.798 (0.455) | 21.850 (22.140) | 0.351 (0.074) | 1.117 (0.220) |
| US-GAN | 0.192 (0.001) | **0.187 (0.005)** | 0.357 (0.009) | 0.278 (0.011) | 0.330 (0.001) | 0.566 (0.001) | 0.755 (0.973) | 2.119 (3.955) | 1.119 (0.007) | 2.610 (0.018) | 0.233 (0.005) | 0.328 (0.007) |
| GP-VAE | 0.258 (0.004) | 0.234 (0.008) | 0.453 (0.014) | 0.495 (0.022) | 0.346 (0.015) | 0.617 (0.015) | 0.414 (0.013) | 0.301 (0.011) | 1.099 (0.032) | 2.973 (0.040) | 0.451 (0.022) | 0.677 (0.031) |
| Mean | 0.708 | 1.078 | 0.588 | 1.096 | 0.799 | 1.416 | 0.738 | 0.971 | 0.423 | 0.581 | 0.763 | 1.258 |
| Median | 0.677 | 1.143 | 0.533 | 1.116 | 0.777 | 1.476 | 0.708 | 1.022 | 0.408 | 0.627 | 0.705 | 1.386 |
| LOCF | 0.264 | 0.429 | 0.346 | 0.511 | 0.547 | 1.094 | 0.425 | 0.491 | 0.140 | 0.181 | 0.365 | 0.636 |
| Linear | 0.165 | 0.231 | **0.214** | 0.252 | 0.343 | **0.539** | 0.267 | 0.178 | **0.078** | **0.035** | 0.247 | 0.279 |
| FIM-$\ell$($w.n.$ = 1) | 0.198 | 0.325 | 0.254 | 0.287 | 0.496 | 0.960 | 0.411 | 0.370 | 0.110 | 0.057 | 0.343 | 0.366 |
| FIM-$\ell$($w.n.$ = 2) | 0.185 | 0.330 | 0.236 | 0.271 | 0.434 | 0.840 | 0.337 | 0.275 | 0.096 | 0.047 | 0.303 | 0.305 |
| FIM-$\ell$($w.n.$ = 4) | 0.190 | 0.371 | 0.352 | 0.609 | 0.384 | 0.714 | 0.317 | 0.251 | 0.093 | 0.045 | 0.273 | 0.251 |
| FIM-$\ell$($w.n.$ = 8) | 0.318 | 0.520 | 0.563 | 1.049 | 0.473 | 0.866 | 0.326 | 0.305 | 0.091 | 0.046 | 0.388 | 0.450 |
| FIM-$\ell$($w.n.$ = 16) | 0.559 | 0.886 | - | - | 0.626 | 1.136 | 0.445 | 0.508 | 0.122 | 0.140 | 0.574 | 0.763 |
| FIM-$\ell$($w.n.$ = 32) | - | - | - | - | - | - | 0.650 | 0.911 | 0.533 | 1.499 | - | - |
| FIM-$\ell$($o.n.$ = 4) | 0.166 | 0.298 | 0.225 | 0.249 | 0.375 | 0.708 | 0.279 | 0.198 | 0.083 | 0.038 | 0.274 | 0.272 |
| FIM-$\ell$($o.n.$ = 6) | 0.240 | 0.506 | 0.250 | 0.282 | 0.583 | 1.295 | 0.469 | 0.662 | 0.210 | 0.336 | 0.294 | 0.287 |
| FIM-$\ell$($o.n.$ = 8) | 0.234 | 0.422 | 0.254 | 0.287 | 0.583 | 1.327 | 0.467 | 0.661 | 0.199 | 0.294 | 0.306 | 0.314 |
| FIM-$\ell$($o.n.$ = 8) | 0.233 | 0.401 | 0.254 | 0.287 | 0.568 | 1.235 | 0.467 | 0.649 | 0.191 | 0.257 | 0.302 | 0.293 |
| FIM-$\ell$($o.n.$ = 12) | 0.208 | 0.370 | 0.254 | 0.287 | 0.533 | 1.098 | 0.420 | 0.453 | 0.187 | 0.227 | 0.327 | 0.334 |
| FIM-$\ell$($o.n.$ = 12) | 0.212 | 0.410 | 0.254 | 0.287 | 0.546 | 1.136 | 0.420 | 0.448 | 0.182 | 0.211 | 0.324 | 0.325 |
| FIM-$\ell$($o.n.$ = 16) | 0.198 | 0.325 | 0.254 | 0.287 | 0.496 | 0.960 | 0.417 | 0.500 | 0.198 | 0.240 | 0.343 | 0.366 |
| Long-FIM-$\ell$($w.n.$ = 16) | - | - | 0.414 | 0.458 | - | - | - | - | - | - | - | - |
| Long-FIM-$\ell$($w.n.$ = 32) | - | - | 0.358 | 0.383 | 0.905 | 1.692 | 0.693 | 1.084 | - | - | - | - |
| Long-FIM-$\ell$($w.n.$ = 64) | 0.360 | 0.512 | 0.283 | 0.282 | 0.709 | 1.193 | 0.495 | 0.576 | - | - | - | - |
| Long-FIM-$\ell$($w.n.$ = 128) | 0.266 | 0.397 | 0.238 | 0.245 | 0.516 | 0.848 | 0.362 | 0.303 | - | - | - | - |
| Long-FIM-$\ell$($w.n.$ = 256) | 0.201 | 0.279 | 0.215 | 0.231 | 0.417 | 0.721 | 0.328 | 0.281 | 0.157 | 0.103 | - | - |
| Long-FIM-$\ell$($w.n.$ = 512) | 0.177 | 0.265 | - | - | 0.365 | 0.651 | 0.310 | 0.265 | 0.106 | 0.052 | - | - |
| Long-FIM-$\ell$($w.n.$ = 1024) | 0.171 | 0.279 | - | - | - | - | - | - | 0.094 | 0.044 | - | - |
| Long-FIM-$\ell$($w.n.$ = 2048) | - | - | - | - | - | - | - | - | 0.089 | 0.043 | - | - |
| Long-FIM-$\ell$($w.n.$ = 4096) | - | - | - | - | - | - | - | - | 0.097 | 0.066 | - | - |

## H  TEMPORAL MISSING PATTERN IMPUTATION: ADDITIONAL RESULTS

### H.1  MOTION CAPTURE

#### H.1.1  DATA DESCRIPTION AND PRE-PROCESSING

Let us now consider the imputation problem setup proposed by Heinonen et al. (2018) on a human motion capture dataset, consisting of 50-dimensional pose measurements of walking subjects. We take the data provided by Yildiz et al. (2019), which was pre-processed according to previous work of Wang et al. (2007).

The dataset contains 43 trajectories of a maximal length of 125. Following Heinonen et al. (2018), we remove 20% out of the center of each trajectory. To compare to Heinonen et al. (2018), we also apply the provided PCA projection and consider the first 3 PCA components. In the following, we call this the *PCA* setup.

Because `FIM` can be applied to arbitrary dimensional data, due to our channel independent strategy (see Appendix D.7), we also consider the 50 dimensional data directly. In the following, we call this the *No PCA* setup.

#### H.1.2  TRAINING LATENTODE

To obtain another baseline model, we train LatentODE (Rubanova et al., 2019) on the first 3 PCA components on the likelihood *outside* of the interpolation window, i.e. it has never seen and is not trained on the (missing) data *inside* the imputation window. We train and test on all 43 trajectories, selecting the model based on the performance on all trajectories. This approach is similar to the approach of models trained in Heinonen et al. (2018).

The training data is standardised for training. We use a LSTM as as the encoder and a diagonal gaussian emission model, fixing its standard deviation to 0.01. We train for 120.000 gradient descent iterations over the whole dataset, using AdamW with learning rate $1e^{-3}$ and weight decay $1e^{-2}$. To help the model learn, we slowly anneal the input time series length over the initial 60.000 epochs, starting at 25 observations.

We did ablation over the hidden size of the model. See Table 14 for the final hyperparameters. The model trained roughly 3.5 hours on a A100 40GB GPU.

#### H.1.3  MODELLING AND RESULTS

In accordance with Heinonen et al. (2018), we compute the performance metrics in 50 dimensional space and only on the missing points inside the imputation gap. We evaluate `FIM`, LatentODE and also a cubic spline composed with Savitzky–Golay filters (Savitzky & Golay, 1964) in the *PCA* setup. Table 23 reports the RMSE, including baselines extracted from Heinonen et al. (2018), while Table 24 reports the MAE.

`FIM` performs almost as well as the specialized LatentODE model, especially when considering the large standard deviation for both approaches. These results indicate that a zero-shot imputation approach is indeed viable.

PCA dimensionality reduction induces some level of intrinsic error. In contrast to our baseline models, `FIM` is not restricted by dimensionality of the data. By applying our model to the 50 dimensional data *without PCA projection*, we can, in principle, avoid this error.

We report the results of this *No PCA* setup in Tables 23 and 24. Indeed, our model is performing better without the PCA projection, showcasing the strengths of our flexible, zero-shot methodology

Table 23: RMSE of imputation in the Motion Capture and Navier Stokes datasets. The RMSE is calculated only in the imputation window, in accordance with Heinonen et al. (2018).

| Model | Filter | Motion Capture | | Navier Stokes | |
|---|---|---|---|---|---|
| | | PCA | No PCA | PCA | No PCA |
| LatentODE | - | $3.066 \pm 1.767$ | - | $0.133 \pm 0.053$ | - |
| Cubic spline | - | $7.333 \pm 2.49$ | $8.95 \pm 2.78$ | $0.174 \pm 0.003$ | $0.186 \pm 0.004$ |
| Cubic spline | Savgol(15, 3) | $6.317 \pm 1.813$ | $6.752 \pm 1.676$ | $0.151 \pm 0.003$ | $0.143 \pm 0.002$ |
| Cubic spline | Savgol( 8, 3) | $7.078 \pm 2.147$ | $8.271 \pm 2.282$ | $0.148 \pm 0.003$ | $0.147 \pm 0.003$ |
| Cubic spline | Savgol( 4, 3) | $7.251 \pm 2.184$ | $8.783 \pm 2.508$ | $0.171 \pm 0.003$ | $0.18 \pm 0.004$ |
| npODE | - | $3.94 \pm 3.50$ | - | - | - |
| GPDM | - | $5.31 \pm 3.39$ | - | - | - |
| VGPLVM | - | $3.91 \pm 1.89$ | - | - | - |
| FIM | - | $3.271 \pm 1.22$ | $\mathbf{2.977 \pm 0.96}$ | $\mathbf{0.103 \pm 0.004}$ | $\mathbf{0.971 \pm 0.005}$ |

Table 24: MAE of imputation in the Motion Capture and Navier Stokes datasets. The MAE is calculated only in the imputation window, in accordance with Heinonen et al. (2018).

| Model | Filter | Motion Capture | | Navier Stokes | |
|---|---|---|---|---|---|
| | | PCA | No PCA | PCA | No PCA |
| LatentODE | - | $1.658 \pm 0.989$ | - | $0.076 \pm 0.03$ | - |
| Cubic spline | - | $3.362 \pm 1.175$ | $4.209 \pm 1.436$ | $0.085 \pm 0.003$ | $0.083 \pm 0.003$ |
| Cubic spline | Savgol(15, 3) | $2.897 \pm 0.871$ | $2.998 \pm 0.881$ | $0.084 \pm 0.00$ | $0.075 \pm 0.002$ |
| Cubic spline | Savgol( 8, 3) | $3.229 \pm 1.029$ | $3.695 \pm 1.22$ | $0.081 \pm 0.003$ | $0.072 \pm 0.002$ |
| Cubic spline | Savgol( 4, 3) | $3.298 \pm 1.035$ | $4.061 \pm 1.354$ | $0.085 \pm 0.003$ | $0.082 \pm 0.003$ |
| FIM | - | $1.765 \pm 0.627$ | $\mathbf{1.611 \pm 0.453}$ | $\mathbf{0.062 \pm 0.003}$ | $\mathbf{0.051 \pm 0.002}$ |

## H.2 NAVIER STOKES

### H.2.1 DATA DESCRIPTION AND PRE-PROCESSING

As an application of FIM to high-dimensional data, we consider the simulation of a two-dimensional, incompressible Navier-Stokes equation from (Course & Nair, 2023)[14]. The equation is simulated on a two-dimensional grid of size $199 \times 1499$ for a total of $596,602$ states. Following (Course & Nair, 2023), we remove the first $20\%$ of the trajectory for warmup and are left with $2441$ simulation steps.

We use this preprocessed simulation to create another imputation dataset, with a similar setup as in the motion capture dataset. We drop the last observation and cut the remaining trajectory into $61$ time series of length $40$. Then we remove the central $20\%$ of each time series, creating a temporal missing pattern imputation task.

While our model can handle this high-dimensional data, as we will show below, we need to apply PCA dimensionality reduction to train a (specialized) baseline model. Following (Course & Nair, 2023), we project the data to 38 dimensional space with randomised PCA, which already captures the high dimensional dynamics well.

In the following, we will again refer to these two setups as *PCA* and *No PCA* respectively.

### H.2.2 TRAINING LATENTODE

As a baseline, we train LatentODE (Rubanova et al., 2019) in the *PCA* setup on the likelihood *outside* of the imputation window, with the same reasoning as for the motion capture imputation problem in Section H.1.2.

LatentODE is trained on all $61$ (standardised) time series, uses a LSTM as the encoder and a diagonal gaussian emission model with fixed standard deviation of 0.01. We train for 300.000 gradient descent iterations over the whole dataset, using AdamW with learning rate $1e^{-3}$ and weight decay

---

[14]https://github.com/coursekevin/svise

Table 25: Performance comparison of different models across window counts.

| Window count | Pre-trained | Fine-tuned on noisy obs | Fine-tuned on clean obs |
|:---:|:---:|:---:|:---:|
| 1 | $10.32 \pm 1.478$ | $8.983 \pm 0.932$ | $9.001 \pm 0.945$ |
| 4 | $7.136 \pm 0.697$ | $5.936 \pm 0.432$ | $5.947 \pm 0.433$ |
| 8 | $2.860 \pm 0.237$ | $2.708 \pm 0.176$ | $2.704 \pm 0.175$ |
| 12 | $1.956 \pm 0.161$ | $1.913 \pm 0.144$ | $1.911 \pm 0.143$ |
| 16 | $1.652 \pm 0.126$ | $1.702 \pm 0.127$ | $1.703 \pm 0.126$ |
| 20 | $1.526 \pm 0.125$ | $1.586 \pm 0.112$ | $1.588 \pm 0.112$ |
| 24 | $1.504 \pm 0.138$ | $1.532 \pm 0.108$ | $1.534 \pm 0.108$ |
| 28 | $1.516 \pm 0.133$ | $1.505 \pm 0.098$ | $1.506 \pm 0.098$ |
| 32 | $1.561 \pm 0.142$ | $1.500 \pm 0.118$ | $1.501 \pm 0.118$ |

$1e^{-2}$. We slowly anneal the input time series length over the initial $60.000$ epochs, starting at just 5 observations.

We studied some ablation over the hidden size. The search grid, including the final hyperparameters, is shown in Table Table 14. The model trained roughly 7 hours on a A100 40GB GPU.

### H.2.3 MODELLING AND RESULTS

Let us first consider the *PCA* setup where we trained the LatentODE baseline. For both `FIM` and LatentODE we report the RMSE, in Table 23, and MAE, in Table 24, inside the imputation gap. In this particular task, `FIM` was again able to match the performance of LatentODE.

As `FIM` can be applied to data of any dimensionality, we also experimented with the *No PCA* setup, imputing the missing data in $596, 602$ dimensional space directly. As in the Motion Capture dataset, `FIM` performs even better without the errors induced by the PCA projection.

Finally, we report the computational load of inference in dataset, as it is considerably larger than in all other experiments, because of its high dimensionality. The application of `FIM` took roughly 9 hours on a A100 40GB GPU. As a comparison, the application of a cubic spline on the same data took roughly $1.1$ hours on 32 CPU cores.

## I ON FINE-TUNING FIM

Although outside the scope of the present work, fine-tuning FIM to specific applications would be of great interest and value.

In practice, one does not have access to any ground-truth initial condition, nor to any time derivative (of the hidden interpolating function). However, the model can still be optimized to reconstruct the available data. To illustrate the plausibility of this approach in a controlled setting, we have fine-tuned `FIM-`$\ell$ to reconstruct the Lorentz system data we studied in Appendix F.4. The details of this dataset can be found in that Appendix.

Setting up the fine-tuning:

1. `FIM-`$\ell$ takes as input the noisy observations along one path and returns (its best guess of) the interpolation function, which can be evaluated at any time. We fine-tune `FIM-`$\ell$ in two settings, namely

   (a) to reconstruct the noisy input data, and
   (b) to reconstruct the ground-truth, clean values of the Lorentz system's solution, at the input observation times.

   Clearly, case (a) is the realistic setting. We consider case (b) as a control case, because `FIM-`$\ell$ was pretrained to reconstruct the clean path.

2. As we described in Subsection 3.2.2 of the main text, `FIM-`$\ell$ can be used to process either sets of up to $L_{\max}$ observations simultaneously, or subsets (or windows) with fewer observations, followed by a combination of the local window estimates. This trick is what allows us to interpolate

time series of any length and handle interpolation functions which lie outside our distribution of "simple" functions.

The number of windows into which we split the target dataset is a **hyperparameter** of the model, as it is clearly problem dependent. Indeed, Figure 8 of Appendix F.4.4 demonstrates the effect of modifying it on the Lorentz system data. To fine-tune $\text{FIM}-\ell$, we therefore need to specify the number of windows hyperparameter. That is, we need to specify the temporal scale at which we want to fine-tune the model. In our simple experiment below we chose to fine-tune $\text{FIM}-\ell$ on four (4) windows.

3. Other details: We fine-tuned $\text{FIM}-\ell$ with a learning rate of 1e-6, which was the learning rate at which we stopped pretraining. We fine-tuned it for two epochs only, on all components of the Lorenz systems (i.e. channel independence strategy).

Table 25 (which expands on Fig. 8) contains our results.

The first thing we notice is that the performance of $\text{FIM}-\ell$ clearly improves at the time scale (i.e. at the window count) on which it was fine-tuned. Indeed, we fine-tuned it on 4 windows. Very importantly, the fine-tuning process does not negatively perturb the interpolation performance at different time scales (i.e. at fewer or larger window counts). Therefore, there is no catastrophic forgetting. This observation alone demonstrates the plausibility of fine-tuning with $\text{FIM}-\ell$.

The second thing we notice is that fine-tuning on the noisy and clean data had a similar effect. The continuous nature of the interpolation function appears to make the fine-tuning robust to noise in the target data. However, we have also observed that if one fine-tunes $\text{FIM}-\ell$ on the noisy data for many more epochs, or simply with a larger learning rate, its performance at different time scales does deteriorate. This occurs because the model starts fitting the noise. Therefore, to properly address fine-tuning within our framework, we would need to define an emission model that handles the noise.

One could picture an emission model, (pre-)trained on our synthetic dataset, which would take as input both the interpolation function $x(t)$ — represented on some sensor grid, like in DeepONets — and the noisy data $y_1, \ldots, y_l$. The emission model would then output the probability of observing the noisy value $y_i$ at any desire time $\tau_i$, that is $p(y_i|x(\tau_i))$.

Coupling $\text{FIM}-\ell$ together with this emission model would allow to finetune $\text{FIM}-\ell$ (or the emission model, or both) to any target dataset. We plan to pursue this general direction in the near future.

