# OpenReview forum: "Zero-shot Imputation with Foundation Inference Models for Dynamical Systems"
_ICLR.cc/2025/Conference — ICLR 2025 Poster_

### Official Review · Reviewer_6Fr3 · 2024-11-02

**Soundness:** 4
**Presentation:** 4
**Contribution:** 3
**Rating:** 6
**Confidence:** 3

**Summary:**

This paper addresses the challenges of imputing missing time-series data. A framework for zero-shot time-series imputation is proposed with 1) a synthetic data generation model for sampling a set of ODE solutions and 2) a neural recognition model mapping the time series data onto parametric functions. Empirical results present improved performance of imputation compared to baseline models.

**Strengths:**

1. The problem of imputing time-series data is important and the idea of linking amortized inference and neural operator is interesting.
2. The methodology of the foundation inference model is well-structured.
3. Experiments were performed on the imputation of point-wise and temporal missing patterns to present the effectiveness of the proposed method.

**Weaknesses:**

1. I think the idea of zero-shot amortized inference framework is somewhat incremental. There have been works on amortized inference for few-shot time-series forecasting (e.g. [1] [2] [3][4]) in terms of lacking sufficient observation and learning single dynamics, where the idea of amortized inference learning the prior knowledge of dynamics is similar to the proposed method. Could the authors add a discussion about these works and the benefits of the proposed method? Also, it would be good if the authors could make a comparison of some of these methods in experiments.
2. In Equation 3, 4, and 5, do $\phi^{\theta}_i$ and $\psi^{\theta}$ share the same parameters $\theta$? Similar problem to the parameter $\psi$ in FIM model in Equation 7 and 8.

**Questions:**

Please find the questions in the Weaknesses section above.

---

> ### Author Response · Authors · 2024-11-19
>
> We would like to thank the reviewer for the helpful comments and questions, which we know will improve the presentation of our work. Below we address each of them.
>
> **@W1**: Could the reviewer please specify the references they refer to? As soon as we know what references the reviewer means, we will address all the proposed questions.
>
> **@W2**: We use $\theta$ to denote the complete set of trainable parameters inside FIM-l (see line 259). Specifically, the set of MLPs $\phi_1$, …, $\phi_5$ and the sequence processing network $\psi_1$ all have different trainable parameters. We denote the set of all these parameters with $\theta$.
>
> Similarly, we use $\varphi$ to denote the complete set of trainable parameters inside FIM **that do not belong to** FIM-l (see lines 345, 346). We make the distinction between the trainable parameters in these two models because FIM leverages the *pretrained* components of FIM-l.
>
> Please note that this distinction is necessary, because the *pretrained* parameters of FIM-l (i.e. $\theta$) are **not** updated during the (pre)training of FIM (see lines 352, 353).
>
> **Proposed modification:** we will change the sentence *"Let us also denote the trainable network parameters with $\theta$"* in line 259 into: *Let us also denote the set of all trainable parameters in these networks with $\theta$*.
>
> Does this modification address the weakness pointed out by the reviewer?

---

> ### Comment · Reviewer_6Fr3 · 2024-11-19
>
> Thank you for pointing out the missing reference and the clarification of the notation. Please find the detailed work below. Given the limited time, the authors could focus on the discussion of the few-shot learning method vs your proposed method. If time allows, the authors could include some comparison experiments.
>
> [1] Hewitt, Luke B., et al. "The variational homoencoder: Learning to learn high capacity generative models from few examples." arXiv preprint arXiv:1807.08919 (2018).
>
> [2] Singh, Gautam, et al. "Sequential neural processes." Advances in Neural Information Processing Systems 32 (2019).
>
> [3] Jiang, Xiajun, et al. "Sequential latent variable models for few-shot high-dimensional time-series forecasting." The Eleventh International Conference on Learning Representations. 2023.
>
> [4] Wang, Rui, Robin Walters, and Rose Yu. "Meta-learning dynamics forecasting using task inference." Advances in Neural Information Processing Systems 35 (2022): 21640-21653.

---

> > ### Author Response · Authors · 2024-11-23
> >
> > We thank the reviewer for providing the references above. As we discuss below, there are a few fundamental differences between these (meta-learning) references and our proposal. In our opinion, what follows demonstrates that, while our proposal can be understood as related to meta-learning, it can hardly be considered as incremental, specially with respect to the references above. Let us then begin by briefly stating the key features of the four references above, as to better explain the differences between their and our proposals.
> >
> > All four works consider the problem of (meta)training a single model on a set of different, albeit related datasets $\{D_1, \dots, D_M\}$, each of which is characterized by a corresponding (that is, shared) latent variable $\{c_1, \dots, c_M\}$. For Jiang et al. (2023), for example, each dataset $D_m$ can correspond to a turbulent flow dynamic simulation, each with a different buoyant force ($c_m$) acting on the fluid. All four works also rely on a trainable decoder model $p_\theta(D_m|c_m)$, which is conditioned on the corresponding variable $c_m$, and a trainable encoder $q_\varphi(c_m|D_m)$, which infers $c_m$ from data (see below, for details). Note that the conditioning on $c_m$ is precisely what allows for meta (or transfer) learning in these works.
> >
> > 1. The first and most important difference between all these approaches and ours is that *they need to be trained on datasets from their target domains*, which makes both their inferred representations and optimized weights **problem specific**. Indeed, every one of these works focuses, by construction, on meta-learning among *similar systems (or datasets) only*. To illustrate, let us consider again the work of Jiang et al. (2023). They first forecast bouncing ball simulations under different gravity conditions. For them, each gravity corresponds to a different dataset, and their inferred representations should encode them (or at least some aspects of them). Later, they consider forecasting turbulent flow dynamics under different buoyant forces and *train their model anew*, in order to encode the forces characterizing each dataset. Thus, they do not attempt to perform meta-learning between e.g. the bouncing ball and the turbulent flow simulations.
> >
> > - *In sharp contrast* our proposal **is not problem specific**, because it entirely relies on the two general assumptions (inductive biases, or prior knowledge) we put forward in lines 165-178 of our manuscript. These simple assumptions pertain to the nature of interpolating functions only, and are therefore independent of the actual data generation mechanisms underlying our target datasets. Given that our model is only trained on a synthetic dataset encoding these two assumptions, it can only recognize patterns that help it reconstruct the best interpolating function given the available (context) data, regardless of the underlying data generation mechanism.
> >
> > 2. A second important difference lies in how these works leverage their prior knowledge for transfer. Specifically, they assume that their target domains allow for the collection of their different yet related datasets $\{D_1, \dots, D_M\}$, on which to train their (meta)models.
> >
> > - We instead fully rely on our two assumptions of lines 165-178, and therefore on our synthetic dataset which encodes them.
> >
> > 3. A third difference is that all these works assumed there is a **shared latent variable** $c_m$ encoding the main features of the $m$th dataset $D_m$ in their collection. The first three works above rely on neural variational inference to optimize their encoder-decoder pairs, and infer their representations in an unsupervised manner. What is more, all these three works leverage encoder models that process sets of observations simultaneously (instead of single observations, like e.g. variational autoencoders [VAE]) in order to infer the shared latent variable $c$. Note that this feature makes them *few-shot encoders*. In contrast, Wang et al. (2022) is more akin to VAE, in that they obtain one representation per observation, and only later regularize their representations to be the shared via their training objective.
> >
> > - In our proposal **our latent variables are not shared**, for each time series has associated to it a single interpolating function. Transfer learning takes place because of the general assumptions we encode into our synthetic dataset.

---

> > ### Author Response · Authors · 2024-11-23
> >
> > 4. A fourth difference pertains to amortization. All four works above, our proposal, and any inference method that makes use of neural networks (like e.g. any VAE-based model) amortize the inference procedure, because they use a single neural network model to map the observations onto the parameters of their latent distributions.
> >
> > - In lines 88-99 of our manuscript we instead refer to amortization in the sense first introduced by Stuhlmueller, et al. (2013). We direct the reviewer to these lines, where we elaborate on the concept of amortization as used throughout our paper.
> >
> > *For all the reasons outlined above, we do not view our work as incremental*, relative to the four references we are discussing --- apart from the shared goal of transferring knowledge from a training set to distinct target sets.
> >
> > **The benefit of our proposal** (compared to the references above) is that, as we demonstrated in our experimental sections, it allows us to automatically impute missing data from widely different systems of any dimensionality, without retraining nor fine-tuning our recognition model.
> >
> > *To conclude*, it is worth noting that none of the references discussed above addresses time series imputation. Specifically, Hewitt et al. (2018) do not study dynamical systems, while the other three focus on forecasting. Consequently, we find no clear or fair basis for directly comparing our proposal with any of these works.
> >
> > We hope the discussion above answers the questions and remarks of the reviewer.
> >
> > **Proposed modification**: We will include a new appendix titled *"Related Work on Meta-learning"* into our manuscript, which will contain and expand upon the discussion above. It will also make reference to other representative references on meta-learning and time series modeling. Finally, we will reference this new appendix in our related work section.
> >
> > **References**
> >
> > - Stuhlmueller, et al: Learning stochastic inverses (Neurips, 2013)

---

> > > ### Comment · Reviewer_6Fr3 · 2024-11-26
> > >
> > > Thanks for the authors' comments on the list of related work. I partially agree with the authors' point, although the first comment "meta-learning is problem-specific" is not accurate. Conceptually, meta-learning is the process of learning to learn on heterogeneous tasks, which is not problem-specific. The statement "perform meta-learning between e.g. the bouncing ball and the turbulent flow simulations" does not make sense if the authors have a clear understanding of heterogeneity. In fact, the synthetic data generation model is problem-specific, as the ODE integrates the domain-specific knowledge. And I would encourage the authors to conduct experiments from Van der Pol oscillator to Air quality if the proposed model is really "non-problem-specific".
> > >
> > > Apart from the above points, the authors' response helps to make the presentation better. I will maintain my scores as they are.

---

> ### Author Response · Authors · 2024-11-28
>
> We thank the reviewer for engaging with us in the discussion. We hope the reviewer will now allow us to further comment on their reply, for we think there *might be* a fundamental point that we have not yet succeeded in conveying to the reviewer, namely that *we use one and the same pre-trained model to impute the data* **in all experiments**, *without any parameter finetuning*.
>
> We have mentioned this throughout the paper and, in fact, the weights of the pre-trained model are attached to our submission (we will also make them accessible on Hugging Face). Yet the last sentence of the reviewer’s comment seems to indicate that this feature was overlooked. *We apologize if we are misreading this comment*. If not, let us continue:
>
> **@** *I would encourage the authors to conduct experiments from Van der Pol oscillator to Air quality*: The prepositions “from” and “to” indicate transfer. Keeping the concept of transfer in mind —  within the context of our work —  one could say that our methodology transfers the knowledge encoded into our synthetic dataset (that is, our two assumptions in lines 165-178) from the pretrained weights onto the Van der Pol oscillator and Air quality imputation tasks.
>
> By this we mean that *after (pre)training on the synthetic dataset, we do not modify the (numerical values of the) parameters of the neural networks building FIM-l nor FIM*. We directly apply our pretrained models — without modification — **throughout all our experiments, in all experimental sections**. In particular, we use the same pre-trained FIM-l model when processing both the Van der Pol oscillator (Section 4.1) *and* the Air quality dataset (Section 4.2) , without modification.
>
> We hope this clarifies why we referred to our proposal as "not problem specific", as compared to the four references cited by the reviewer. Let us continue by adding that we never said meta-learning is problem specific. In fact, all our remarks and discussion focused only on the works referenced by the reviewer. Quoting *word-by-word* our remark above:
>
> *“They [The four works referenced by the reviewer] need to be trained on datasets from their target domains, which makes both their inferred representations and optimized weights problem specific.”*
>
> By this we meant that — keeping with the example of Jiang et al. (2023) – the parameters of the neural network building of the model of Jiang et al. (2023) are **not the same** for the bouncing ball experiment and the turbulent flow experiment. Similarly, the semantic content of the shared representations inferred by the model of Jiang et al. (2023) is **not the same** for the bouncing ball experiment and the turbulent flow experiment. In the former case it corresponds to gravity, whereas in the latter it corresponds to buoyant force.
>
> *In contrast*, our method **maintains the same network parameters and representation semantics throughout all experiments**. The representations consistently correspond to the time derivative and initial conditions of the hidden interpolating function, regardless of the target dataset.
>
> All that being said, let us conclude by acknowledging that “problem specific” and “not problem specific” were perhaps not the best adjectives to make our points.

---

> > ### Comment · Reviewer_6Fr3 · 2024-11-28
> >
> > Thanks again for the authors' clarification on their proposed model. However, the main point in the initial comment was the idea of integrating prior knowledge into downstream tasks (such as the dynamic modeling in the four works), while it was confounded with the comparison between the objective of building a foundation model and different model designs in the four papers.
> >
> > All these works, including the proposed model and the four meta-learning methods, share the same idea that prior knowledge is learned and integrated into downstream tasks. In the proposed model, the prior knowledge is represented as the pre-trained model. In the four meta-learning methods, it could be feature embeddings, physics parameters, or statistical distributions. Depending on the goal of the task, the prior knowledge could be either sufficiently elaborated (in foundation models) or task-specific (in the four works).
> >
> > I agree that the authors' comments clarify that the proposed work is novel in terms of how to implement the prior knowledge and how to use the knowledge for imputation. The high-level idea behind the assumption of inductive bias or prior knowledge in lines 165-178 is not new but has been practiced in various applications.

---

> > > ### Author Response · Authors · 2024-12-03
> > >
> > > We thank the reviewer for the engaging discussion. Adding this discussion to our related work and appendix will enhance the presentation and completeness of our paper.

---

### Official Review · Reviewer_8Y1g · 2024-11-03

**Soundness:** 3
**Presentation:** 3
**Contribution:** 3
**Rating:** 6
**Confidence:** 4

**Summary:**

The paper develops a ODE-based foundation model for time series imputation. The idea is to simulate a large collection of incomplete time series including both point-wise and temporal missing patterns. A recognition model is trained to map such time series to the initial condition, and time derivatives at each time step, with both mean and log-variance prediction. The design of the recognition model is inspired by deep ONet. There is no need to fine tune the model on application-specific data. The experiments show the improvement over many sota methods specifically trained on application data.

**Strengths:**

1. very interesting work on time series imputation, potentially pointing out a new direction
2. design of neural recognition model is novel and interesting, albeit inspired by deepOnet. Such "borrowing" is still interesting.
3. zero-shot performance is surprisingly good

**Weaknesses:**

1. as admitted by the authors, one limitation is that when the actual application does not well match the assumption of the simulation data, the performance can deteriorate. Though the paper focuses on zero-shot prediction, it will be great to discuss  possible methods for fine tuning or adaptation to specific applications. Since your training assumes the ground-truth of the initial and time derivatives of each sampled trajectories are known, and the neural recognition models directly learn to fit them, such training cannot directly apply to real dataset where these ground-truth is unknown.
2.  The design of the neural recognition model seems not well justified, especially for the branch-net component. why do you use RNN? Why not using attention mechanism instead? Given RNN/LSTM's are replaced by attention  nearly every where, the authors should explain their rationale of such choices, either theoretically or empirically.
3. How are the hyperparameters selected? Why using 1024 dimensional embeddings? Are this done by a rigorous validation process? If so, how?  It will be good for the authors to specify the details of hyperparmeter selection and validation process.

**Questions:**

see above.

---

> ### Author Response · Authors · 2024-11-20
>
> Let us start by thanking the reviewer for both the detailed review and the kind words about our work. Below we address each of the weaknesses, comments and questions.
>
> **@W1**: we thank the reviewer for this very interesting suggestion. Although outside the scope of the present work, we agree that fine-tuning FIM to specific applications would be of great interest and value.
>
> As pointed out by the reviewer, in practice one does not have access to any ground-truth initial condition, nor to any time derivative (of the hidden interpolating function). However, the model can still be optimised  to **reconstruct** the available data. To illustrate the plausibility of this approach in a controlled setting, we have fine-tuned FIM-l to reconstruct the Lorentz system data we studied in Appendix E. The details of this dataset can be found in lines 1821-1829 of that Appendix.
>
> **Setting up the fine-tuning**:
>
> 1. FIM-l takes as input the noisy observations along one path and returns (its best guess of) the interpolation function, which can be evaluated *at any time*. We fine-tuned FIM-l in two settings, namely
>
> (a) to reconstruct the noisy input data, and
>
> (b) to reconstruct the ground-truth, clean values of the Lorentz system’s solution, at the input observation times.
>
> Clearly, case (a) is the realistic setting. We consider case (b) as a control case, because FIM-l was pretrained to reconstruct the clean path (for details, check out Appendix B.3 where we introduced the training objective).
>
> 2. As we described in the Subsection titled *Processing Data of Any Length and Dimensionality with FIM-l* in lines 295-310 of the main text, FIM-l can be used to process either sets of up to $L_{max}$ observations simultaneously, or subsets (or windows) with fewer observations, followed by a combination of the local window estimates. This trick is what allows us to interpolate time series of any length and handle interpolation functions which lie outside our distribution of “simple” functions.
>
> The **number of windows** into which we split the target dataset **is a hyperparameter** of the model, as it is clearly problem dependent.  Indeed, Figure 8 of Appendix E demonstrates the effect of modifying it on the Lorentz system data. To fine-tune FIM-l, we therefore need to specify the number of windows hyperparameter. That is, we need to specify *the temporal scale at which we want to fine-tune the model*. In our simple experiment below we chose to fine-tune FIM-l on **four (4) windows**.
>
> 3. Other details: We fine-tuned FIM-l with a learning rate of 1e-6, which was the learning rate at which we stopped pretraining. We fine-tuned it for two epochs only, on all components of the Lorenz systems (i.e. channel independence strategy)
>
> **Results**: The Table below (which expands on Fig 8) contains our results.
>
> | Window count  |  Pre-trained       | Fine-tuned on noisy obs  | Fine-tuned on clean obs  |
> |:--------------:|:--------------------:|:-------------------------:|:-------------------------:|
> |            1  | 10.32 $\pm$ 1.478 |     8.983 $\pm$ 0.932    |      9.001 $\pm$ 0.945   |
> |            4  |  7.136 $\pm$ 0.697 |     5.936 $\pm$ 0.432    |      5.947 $\pm$ 0.433   |
> |            8  |  2.860 $\pm$ 0.237 |     2.708 $\pm$ 0.176    |      2.704 $\pm$ 0.175   |
> |           12  |  1.956 $\pm$ 0.161 |     1.913 $\pm$ 0.144    |      1.911 $\pm$ 0.143   |
> |           16  |  1.652 $\pm$ 0.126 |     1.702 $\pm$ 0.127    |      1.703 $\pm$ 0.126   |
> |           20  |  1.526 $\pm$ 0.125 |     1.586 $\pm$ 0.112    |      1.588 $\pm$ 0.112   |
> |           24  |  1.504 $\pm$ 0.138 |     1.532 $\pm$ 0.108    |      1.534 $\pm$ 0.108   |
> |           28  |  1.516 $\pm$ 0.133 |     1.505 $\pm$ 0.098    |      1.506 $\pm$ 0.098   |
> |           32  |  1.561 $\pm$ 0.142 |     1.500 $\pm$ 0.118    |      1.501 $\pm$ 0.118   |
>
>
> - The first thing we notice is that the performance of FIM-l clearly improves at the time scale (i.e. at the window count) on which it was fine-tuned. Indeed, we fine-tuned it on 4 windows. Very importantly, the fine-tuning process does not negatively perturb the interpolation performance at different time scales (i.e. at fewer or larger window counts). Therefore, *there is no catastrophic forgetting*. This observation alone demonstrates the plausibility of fine-tuning with FIM-l.
>
> - The second thing we notice is that fine-tuning on the noisy and clean data had a similar effect. The continuous nature of the interpolation function appears to make the fine-tuning robust to noise in the target data. However, we have also observed that if one fine-tunes FIM-l on the noisy data for many more epochs, or simply with a larger learning rate, its performance at different time scales does deteriorate. This occurs because the model starts fitting the noise. Therefore, to properly address fine-tuning within our framework, we would need to define an *emission model* that handles the noise.

---

> ### Author Response · Authors · 2024-11-20
>
> *(Continuation from above)*: One could picture an emission model, (pre-)trained on our synthetic dataset, which would take as input both the interpolation function $x(t)$ — represented on some sensor grid, like in DeepONets — and the noisy data $y_1, \dots, y_l$. The emission model would then output the probability of observing the noisy value $y_i$ *at any desire time $\tau_i$*, that is $p(y_i|x(\tau_i))$.
>
> Coupling FIM together with this emission model would allow to finetune FIM (or the emission model, or both) to any target dataset. We plan to pursue this general direction in the near future.
>
> **Proposed modification**: We will include the discussion above as a new Appendix titled: *On fine-tuning FIM* into the manuscript.
>
> **@W2**: We purposely tried to make the definition of FIM-l in Section 3.2.1 agnostic to the architecture of the sequence processing network $\psi_1$.
>
> That being said, in our ablation studies of Appendix B.5 (lines 1069-1112) we did experiment with the architecture and found that both BiLSTM and Transformer networks yielded similar performance (Table 5). There is no specific reason why we reported our results with BiLSTM instead of with Transformer in the main text.
>
> Finally, we noticed that we did not include the specifics for these two networks in our manuscript. These follow below:
>
> - Transformer: MLP: 4x256, N-of-layers: 4, QKV-dim: 256, N-of-heads 8, output_dim: 256, total-parameter-count: ~20M
> - BiLSTM: MLP: 4x1024, hidden-dim: 512 (i.e. 256 per each directions), total-parameter-count: ~20M
>
> We will include these details back into our manuscript.
>
> **@W3**: In Table 4 of Appendix B.5 (lines 1069-1112), we report our studies on the effect of changing the size of the training dataset and the parameter count of the BiLSTM-based FIM-l. As can be seen in this Table, going beyond the 20M parameter count did not improve the performance of the model.
>
> Unfortunately, here again we did not include the specifics of the architectures. These follow:
>
> - 2M: MLP: 4x256, hidden-dim: 256
> - 20M: MLP: 4x1024, hidden-dim: 512
> - 50M: MLP: 3x2048, hidden-dim: 1024
>
> These results justify our hyperparameter selection.
>
> We hope the comments and arguments above answer the questions and weaknesses proposed by the reviewer.

---

> > ### Comment · Reviewer_8Y1g · 2024-12-02
> >
> > Thanks for the author's response and clarification. Hope these results and discussion will be integrated into the final version.

---

> > > ### Author Response · Authors · 2024-12-03
> > >
> > > Let us thank the reviewer again for the interesting suggestions and questions. As we wrote in our first general reply (*Summary of proposed modifications*):
> > >
> > > - We will create a new appendix titled “*On fine-tuning FIM*”, which will include and expand upon our answer to W1;
> > > - We will include the missing architecture details pertinent to our ablation studies of Appendix B.5, which justify the specifics of the model architecture of our final model.

---

### Official Review · Reviewer_nBHN · 2024-11-04

**Soundness:** 4
**Presentation:** 4
**Contribution:** 3
**Rating:** 8
**Confidence:** 3

**Summary:**

This paper introduces foundation model for time-series imputation, using several parametric functions of ODEs for training the model. The authors generate many synthetic noisy, irregular observations to train recognition model in offline manner.  They use combination of objective function with respect to initial value and data points to train this model. This trained model can conduct zero-shot imputation on unseen time series dataset with different dimensions. They validate their method on various different benchmarks in several domains.

**Strengths:**

- This paper is well written and organized. I enjoy reading this paper.
- To the best of my knowledge, this paper is novel in that this firstly introduces the zero-shot imputation of time-series with dimension free inference.
- I think the training objective and the way they give supervision to the model is also novel.
- The experimental results are impressive, covering a wide range of datasets and scenarios enough to prove the zero-shot capability of this method. Additionally, the paper faithfully includes details for reproducing the experiments which is helpful

**Weaknesses:**

- Although this model impute the data in zero-shot manner, experimental results are not that impressive in Table 2. Also, the authors use reported value from the other papers which can question whether comparison is really fair.
-  I think it would be better to include ODE based methods for the baseline considering concept of this paper.

**Questions:**

- I think some ablation studies for measuring the generalization capability of this model will be beneficial.

---

> ### Author Response · Authors · 2024-11-19
>
> We thank the reviewer for their encouraging feedback, as well as for the proposed questions and weaknesses. We’re very grateful for their recognition of our work.
>
> **@W1**:  Regarding fairness, our personal experience suggests that one rarely gets baselines to perform as well as their authors do, simply because the authors of said baselines are intimately familiar with the nuances of their proposal. We thus think that by reporting scores from other papers, we are being as fair as possible to the baselines.
>
> That said, we are also sure we evaluate our models on *exactly the same setting as the authors do*, for we obtained all target datasets we used in Table 2, as preprocessed by the authors, from their public repositories. We thus hope their published datasets are indeed the same ones they used for computing their scores in their publications. If this is the case, we believe our evaluations are fair with respect to the baselines.
>
> Does the reviewer disagree? Or are we misunderstanding the reviewer’s comment?
>
> **@W2**: We did include ODE baselines. Indeed, every baseline we compare against in Section 4.1 is ODE-based. To wit: ODEFormer, LatentODE (details in Appendix E.4, lines 1810-1921) and NeuralODEProcesses (details in Appendix E.5, lines 1921 - 1985) are all ODE-based.
>
> In Section 4.2, we compared against BRITS, which leverages a linear ODE to model the values of the times series between observations.
>
> Finally, in Section 4.3, we compared against both LatentODE and npODE (details in Appendix G.1, lines 2226-2305), which are ODE-based.
>
> We however noticed we did not make explicit reference to the comparison against npODE in the main text. We will add this reference to the manuscript.
>
> **@Q1**: We did include ablation studies in Appendix B.5 (lines 1069-1112). Specifically, we studied changing:
>
> - the parameter count in the models;
> - the training dataset size;
> - the architecture of the sequence processing network $\psi_1$ (i.e. LSTM vs Transformer); and
> - the maximum and minimum number of context points seen by the model during training.
>
> While our experiments in Section 4 of the main text demonstrated that our models perform well on many different synthetic and real-world applications and therefore that they *generalise* beyond the (synthetic) training dataset, we are unsure about how to set up an ablation study measuring their generalisation capabilities.
>
> Could the reviewer please be more precise about their expectations of the setup of such a study?

---

> > ### Comment · Reviewer_nBHN · 2024-11-27
> >
> > For W2, I actually mean table2 (real world imputation experiment.). For W1, I partially agree with your idea, but complexity of experimental setup can cause difference even with the best effort of replicating the environment. So I think at least trying to re-implementing baseline is important for readers who are interested in the performance of this model.

---

> > > ### Author Response · Authors · 2024-11-28
> > >
> > > We thank the reviewer for their clarifying remarks and their continued engagement with our work. Let us briefly extend our previous reply with the following two points.
> > >
> > > @W1: We understand the reviewer's concern about the inherent complexity of experimental setups typical in machine learning. And we fully agree that there is always a possibility of systematic error.
> > > To this we can only add that Du et al. (2024) introduces a benchmark which implements and compares most baselines of Table 2, 21 and 22. We extract the scores in these tables from their reports, and we obtained the preprocessed datasets and metric implementations from their public repository.
> > >
> > > We will now try to clone their public repository, run it and evaluate our pretrained models with their evaluation scripts. This way the readers will be confident about the fairness of our comparisons.
> > >
> > > @W2: Let us simply reiterate here that our goal with this section was to compare our zero-shot approach against the most common SOTA models in the time series imputation literature. Of these models, only BRITS is an ODE-based method.

---

### Official Review · Reviewer_6Udz · 2024-11-04

**Soundness:** 2
**Presentation:** 3
**Contribution:** 2
**Rating:** 5
**Confidence:** 3

**Summary:**

This paper proposes a supervised learning framework for zero-shot time series imputation in dynamical systems of any dimensionality. The framework uses a synthetic data generation model based on two key assumptions:
(1) time series with point-wise missing data have simple ODE-based interpolation solutions, and
(2) time series with temporal missing patterns are locally simple.
The authors introduce two neural interpolation models, FIM and FIM-l, designed for point-wise and temporal missing patterns, respectively.
The approach is evaulated over 8 datasets

**Strengths:**

— The proposed FIM has great zero-shot imputation performance and achieves SOTA results on multiple datasets.
— Unlike prior work, this paper proposes a zero-shot approach that can be used on processes of any dimensionality.
— The synthetic data generation model offers a general method for creating meaningful synthetic time series, which could support future research.
- The authors are honest about the limitation of the approach, especially those related to the synthetic distributions used in the analysis and setting (ODEs) as opposed to real-world data which can exhibit any (or no ) distrubution.

**Weaknesses:**

- Fig 2 is difficult to read, the colors are too  similar
- The paper is too "buzzwordy", I would have  preferred more of a proper technical discussion
- The results of the proposed method are not always close to SOTA or better than SOTA

**Questions:**

— The FIM-l model only works for time series that follow a ‘simple’ distribution. Since existing imputation models already perform well on real-world and synthetic data, the practical value of the proposed model might be limited.
— The authors did not explain clearly how the FIM could handle processes of any dimensionality, and how the performance of the model varies with the dimensionality in practice.

---

> ### Author Response · Authors · 2024-11-19
>
> Before we start, let us thank the reviewer for their review, as well as for the proposed questions and weaknesses. We attempt to answer them in what follows.
>
> **@W1**: We will modify the colour to make the difference between the ground-truth trajectory and our estimates more apparent.
>
> **@W3**: Let us simply comment here that all baseline’s scores in Table 2 correspond to the scores of SOTA models that are trained on the target datasets. In contrast, FIM-l is neither trained nor finetuned on the target datasets.
>
> It is not a priori obvious that a pre-trained model can outperform, or even perform on par, with models that are actually trained on the target datasets. Indeed, most foundation models for time series, which mostly focus on forecasting, often perform *at best* on par with models that are trained or fine-tuned on the target datasets. See e.g. [1, 2, 3]. Our findings of Table 2 are in agreement with this.
>
> **@Q1**: We thank the reviewer for this question, because it highlights a very important feature of our methodology, namely that it can perform zero-shot imputation of point-wise missing values in complex time series, *via their decomposition into simpler time series*. It seems however that we did not stress this feature enough.
>
> FIM-l is indeed trained on what we define as “simple” functions. However, in the subsection titled *“Processing data of any length and dimensionality with FIM-l”* of the main text, in lines 295-310, we explain how to deal with data whose interpolating function is **not** represented by our ensemble of “simple” functions. In short, our key idea is that the domain of any complex function can be decomposed into subdomains or regions on which the function is locally “simple”. We therefore first make use of FIM-l to process the data on each of these regions (or subdomains), and then combine the local estimates into a global one. We provided the specifics on how we implemented these operations in Appendix B.6.2 (lines 1138-1195).
>
> Note that this simple approach is what allowed us to impute point-wise missing data sampled from complex analytic functions, like those in Figure 2. It also allowed us to impute point-wise missing data in real-world scenarios, as shown on the left and centre panel of Figure 3, and as demonstrated in Table 2. We invite the reader to also check Figure 6 in the Appendix, which contains the reconstructed data for all 63 ODE solutions in ODEBench, many of which feature complex functions. In sum, all these cases feature data whose interpolating functions do not belong to our ensemble of “simple” functions, and yet we are able to impute them with our windowing approach of section B.6.2 and FIM-l.
>
> **Proposed modification:** In order to make this feature of our methodology more visible to readers, we will modify the title *“Processing data of any length and dimensionality with FIM-l”* to *“Beyond “simple” functions: Processing data of any length and dimensionality with FIM-l”*.  Does the reviewer think this modification makes our point clearer?
>
> We hope the clarifications above have helped convey to the reviewer how we use FIM-l to process data that extends beyond our ensemble of “simple” functions. **We also hope they elucidate the practical value of  FIM-l**. To wit, as demonstrated in all our experiments, the practical value of FIM-l lies in its ability to be used off the shelf across very different cases, without the need for any fine-tuning. It also works as the backbone of FIM which, as we demonstrated in 4.3, can perform zero-shot imputation of  the significantly harder temporal missing patterns, and outperform models trained on the target datasets.
>
> **@Q2**: In lines 189-193 we first explain that our methodology takes a channel independent strategy. That is, it treats each component of every target dataset as independent, and therefore processes them independently with FIM-l. Later, in the aforementioned subsection titled *“Processing data of any length and dimensionality with FIM-l”* of lines 295-310, we write: we adopt a channel independent strategy and process each component of any target, D-dimensional process independently with FIM-l. There, we also refer the reader to Appendix B.6 (lines 1115-1195), where we further expatiate on this treatment.
>
> We believe these sentences explain how FIM handles processes of any dimensionality. If these sentences are insufficient, could the reviewer please specify which aspects remain unclear?

---

> ### Author Response · Authors · 2024-11-19
>
> **On performance vs dimensionality**: The performance of FIM — or of any other method that takes a channel independent strategy — does not directly nor necessarily depend on the dimensionality of the target process. Rather, it depends on the complexity of the dynamics characteristic of each of the components in the target process, and whether this complexity is represented by the training distribution of the model in question. Indeed, a channel independent strategy simply does not rely on the correlations present among the components of the target process. Interestingly enough, it has recently been argued (see [4]) that models trained using this strategy outperform models which are trained to exploit correlations across the components.
>
> Regarding FIM-l specifically, let us kindly direct the attention of the reviewer to e.g. Table 11 of the Appendix, where we report the MAEs of FIM evaluated on ODEBench, separated by the dimensionality of the target process. There we see that it’s easier for FIM-l to fit the four dimensional processes than, say, the one dimensional ones. The three dimensional processes, which include many chaotic systems, are in contrast the hardest to fit. Figure 6 in the Appendix paints a similar picture.
>
> We hope these arguments answer the questions of the reviewer.
>
> **References:**
>
> 1.  Liu et al. (KDD 2024): Generative Pretrained Hierarchical Transformer for Time Series Forecasting;
> 2.  Darlow et al. (ICLR 2024): DAM: Towards a Foundation Model for Time Series Forecasting;
> 3.  Das et al. (ICML 2024): A decoder-only foundation model for time-series forecasting.
> 4.  Han et al. (2024): The Capacity and Robustness Trade-off: Revisiting the Channel Independent Strategy for Multivariate Time Series Forecasting

---

> ### Comment · Reviewer_6Udz · 2024-11-26
> **Reply to authors**
>
> The authors have improved the depth of their manuscript, however, they only partially address my comments. I agree that their approach is working well on simple functions defined by ODEs. However, real world data are either not governed by ODEs or are non-stationary.
> For these reasons, I intend to keep my scores as they are.

---

> > ### Author Response · Authors · 2024-11-26
> >
> > **@they [the authors] only partially address my comments**: We have thoroughly commented on every question and concern raised by the reviewer. Could the reviewer please elaborate on what we seem to have missed?
> >
> > **@I agree that their approach is working well on simple functions defined by ODEs. However, real world data are either not governed by ODEs or are non-stationary**: Let us reiterate that we have shown in our experimental sections that our approach works well on complex, real-world datasets. In fact, and as we wrote in our answer to **Q1** above:
> >
> > - In section 4.2, we evaluated FIM-l on **8 real-world, high-dimensional datasets** featuring point-wise missing data. Indeed, we considered traffic speed, road occupancy, pedestrian activity and solar records, as well as two popular air quality datasets. All these datasets are real-world datasets which are commonly studied by the time series community; especially those who focus on imputation and forecasting.
> >
> > - In section 4.3, we evaluated FIM on the well-known motion capture dataset, as well as on Navier-Stokes simulations, both featuring (the more complex) temporal missing data patterns. Again, **these datasets are real-world, high-dimensional datasets**.
> >
> > Please note that we are not the first — *nor will we be the last* — to model empirical data with ODEs. Indeed, many of the SOTA baselines we compared against, which considered similar or the same target datasets as we did, are also ODE-based.

---

### Author Response · Authors · 2024-11-26
**General reply to reviewers**

We would like to thank all reviewers for their valuable comments and questions. We have carefully read and addressed all of them, for each reviewer separately, in the discussions below.

We are happy to provide any additional information the reviewers might need during the discussion period, in order to make an informed decision. For now, let us summarise the main updates we propose to do to the manuscript, which we believe address all questions and concerns of the reviewers.

**Summary of proposed modifications**

1. To make the feature of our methodology that allows us to process time series of any length and large complexity more visible to readers, we will modify the title *“Processing data of any length and dimensionality with FIM-l”* of the subsection in lines 295-310 into *“Beyond “simple” functions: Processing data of any length and dimensionality with FIM-l”.* This modification addresses **Question 1 of reviewer 6Udz**.

2. To address **weakness 2 from reviewer nBHN** we will specify, near lines 373-377, which of our baselines are ODE-based and to which subsection (i.e. 4.1, 4.2 or 4.3) they are pertinent.

3. We will create a new appendix titled *“On fine-tuning FIM”*, which will include and expand upon our answer to **weakness 1 from reviewer 8Y1g**. Our answer demonstrates that fine-tuning FIM is plausible, interesting and of practical value. It also puts forward the way we plan to pursue this idea in future work.

4. To address **weakness 2 and 3 from reviewer 8Y1g**, we will include the missing architecture details pertinent to our ablation studies of Appendix B.5. These details justify the specifics of the model architecture of our final model.

5. We will create a new appendix titled *“Related Work on Meta-learning”*, which will contain and expand upon our answer to **weakness 1 proposed by reviewer 6Fr3**. It will also make reference to other representative references on meta-learning and time series modeling. Our answer explains the key differences between our zero-shot approach and recent meta-learning proposals for dynamical systems, thereby elucidating the novelty of our methodology.

6. We will change the sentence *"Let us also denote the trainable network parameters with $\theta$"* in line 259 into *“Let us also denote the set of all trainable parameters in these networks with $\theta$”* to address **weakness 2 from reviewer 6Fr3**.

---

> ### Author Response · Authors · 2024-12-03
>
> The discussion period is about to end. We therefore would like to thank once again the reviewers for their valuable input and summarize the output of the discussions.
>
> As we described in our first general reply to all reviewers, many of the questions and concerns raised by the reviewers were resolved via small modifications to either the appendix or the main text. Other questions opened up interesting discussions on the possibility of fine-tuning within our framework, and the connection of our framework with recent meta-learning approaches for dynamical systems. These discussions will be incorporated and expanded upon in our manuscript as two new appendices.
>
> Beyond these points, Reviewer nBHN raised concerns about fairness in our comparison with the baselines in Table 2. We clarified that the baseline scores, the code for their computation, and the preprocessed datasets were obtained from a recent public benchmark. To further ensure fairness, we propose cloning the benchmark's public repository, running it, and evaluating our pretrained models using their evaluation scripts. Details of these additional experiments will be included into a new appendix
>
> Finally, Reviewer 6Udz considered the results in Table 2, where our proposal does not consistently achieve state-of-the-art performance, to be a weakness. In response, we note that all baseline scores correspond to state-of-the-art models trained directly on the target datasets, whereas our proposal is neither trained nor fine-tuned on these datasets. It is not immediately evident that a zero-shot model should outperform or match models specifically trained on the target datasets. Indeed, most foundation models for time series, which primarily focus on forecasting, typically perform at best on par with models trained or fine-tuned on the target datasets (see, e.g., [1, 2, 3]). Our findings in Table 2 align with this trend.
>
> **References**
>
> 1. Liu et al. (KDD 2024): Generative Pretrained Hierarchical Transformer for Time Series Forecasting;
> 2. Darlow et al. (ICLR 2024): DAM: Towards a Foundation Model for Time Series Forecasting;
> 3. Das et al. (ICML 2024): A decoder-only foundation model for time-series forecasting.

---

### Meta-Review · Area_Chair_priT · 2024-12-23

**Metareview:**

The paper addresses the problem of imputing missing values in time series whose dynamics are assumed to follow unknown ODEs. The objective is zero-shot imputation, i.e. performing imputation without re-training from a series of new observations. The authors propose a “foundation models” that is pretrained on a large number of synthetic series to generate ODE features. The methodology proceeds in two steps. First generate a large set of noisy ODE solutions corresponding to sparse and noisy time series, second learn to map the time series data to parametric functions consisting of the initial condition and the evolution function of the ODE that generated the data. The later is the model that best interpolates the data. At inference, using this interpolation property, the pretrained model  is used for imputation in two settings: pointwise and temporal imputation. The proposed approach is evaluated on diverse time series datasets.

The reviewers acknowledge the originality of the approach targeting zero shot imputation and of the methodology which implements a novel method for amortized inference. They all stress that the proposed method is limited to relatively simple time series defined by ODEs. However since these limitations are clearly outlined in the paper and considering the novelty of the approach that opens new perspectives and the SOTA performance in the considered evaluation setting, I propose an acceptation.

**Additional Comments On Reviewer Discussion:**

The reviewers consider the approach novel and believe it opens new perspectives, even though it is limited to simple settings. The authors have added new experiments and addressed most of the concerns.

---

### Decision · Program_Chairs · 2025-01-22

Accept (Poster)